# *SNUPN* deficiency causes a recessive muscular dystrophy due to RNA mis-splicing and ECM dysregulation

SNURPORTIN-1, encoded by *SNUPN*, plays a central role in the nuclear import of spliceosomal small nuclear ribonucleoproteins. However, its physiological function remains unexplored. In this study, we investigate 18 children from 15 unrelated families who present with atypical muscular dystrophy and neurological defects. Nine hypomorphic *SNUPN* biallelic variants, predominantly clustered in the last coding exon, are ascertained to segregate with the disease. We demonstrate that mutant SPN1 failed to oligomerize leading to cytoplasmic aggregation in patients' primary fibroblasts and CRISPR/Cas9-mediated mutant cell lines. Additionally, mutant nuclei exhibit defective spliceosomal maturation and breakdown of Cajal bodies. Transcriptome analyses reveal splicing and mRNA expression dysregulation, particularly in sarcolemmal components, causing disruption of cytoskeletal organization in mutant cells and patient muscle tissues. Our findings establish *SNUPN* deficiency as the genetic etiology of a previously unrecognized subtype of muscular dystrophy and provide robust evidence of the role of SPN1 for muscle homeostasis.

Muscular dystrophies (MDs) are a group of inherited disorders that encompasses distinct yet interlocking subtypes, each displaying a heterogeneous spectrum of phenotypes and genetic causes[1]. These disorders share basic features, including progressive muscular weakness resulting from myofiber degeneration and their replacement with fibrous and fatty tissues. Dystrophic muscles are characterized by an accumulation of growth factors and cytokines that mediate the progression of fibrosis[2]. The overexpansion of the muscle fibrosis is considered the endpoint of severe dystrophies.

The complexity and overlap of clinical features between MDs have led to the development of a comprehensive scheme for identifying new cases including thorough clinical phenotyping, mode of inheritance, and the use of electrophysiological, histopathological, biochemical, and molecular tests. While this strategy has greatly improved the diagnosis of MDs, half of the patients still lack a definitive molecular signature[3].

To date, pathogenic variants in more than 60 different genes have been associated with various forms of MDs[4]. Duchenne muscular dystrophy (DMD, MIM310200), a lethal X-linked recessive disease caused by deficiency in dystrophin, a sarcoplasmic protein, was the first characterized and the most common inherited muscular dystrophy[5,6]. Other types of MD were later reported such as myotonic dystrophy type 1 (DM1, MIM160900), facioscapulohumeral dystrophy (FSHD, MIM158900), Emery-Dreifuss muscular dystrophy (EDM2, MIM181350), numerous types of limb-girdle muscular dystrophies (LGMDs) and congenital muscular dystrophies (CMDs)[7–11]. LGMDs and CMDs show more commonly autosomal recessive inheritance pattern and a lower incidence compared to DMD[12]. In many cases, the boundary between LGMDs and CMDs remains blurred. For instance, patients carrying loss-of-function pathogenic variants in *LAMA2* encoding for the extracellular protein Laminin alpha 2, present with the most severe form of CMD (MDC1A, MIM607855), whereas missense pathogenic variants lead to a milder, later-onset LGMD (LGMDR23, MIM618138)[2].

Recent advances in molecular diagnostic tools and greater pathophysiological insight initiated a trend toward functional classification of MDs. Genetic subtypes of CMDs and LGMDs have been split into functional clusters mostly related to the cellular localization of the altered proteins[9,13]. MDs pathogenesis is associated with dysfunction of a large variety of proteins found in the extracellular matrix (ECM),

e-mail: nbeillard@ku.edu.tr

basal lamina, sarcolemma, or in different organelles such as endo-plasmic reticulum, nuclear envelope, mitochondria, and lysosomes[9,14]. They are frequently triggered by defects in the dystrophin-glycoprotein complex (DGC). The main function of this complex, localized in the sarcolemma, is to maintain the muscle fiber integrity by connecting the cytoskeleton actin fibers to laminin of the ECM. In addition to pathogenic variants in structural genes, alterations in pre-messenger RNA splicing regulators can also contribute to the devel-opment of muscle disorders. For example, splicing regulator dys-function is associated with FSHD or DM1, while defects in survival of motor neuron 1 (SMN1), which plays a central role in the biogenesis of small nuclear ribonucleoproteins (snRNPs), can lead to spinal mus-cular atrophy (SMA1, MIM253300)[15,16].

$SNUPN$, which codes for SNURPORTIN-1, is a key adapter protein for nuclear import of snRNPs[17]. SPN1 binds directly to the tri-methylguanosine cap ($m_3G$ cap) and Importin-β1 (Imp-β1) to form the snRNP import complex[18,19]. In addition, oligomers of SMN bridge the gap between the import machinery and a ring-shaped complex of seven Sm proteins that assembles around the Sm core domain of snRNAs[20]. Once the snRNP assembling complex is imported into the nucleus, snRNPs and SMN are released and directed to Cajal bodies (CBs) where they contribute to spliceosome assembly whereas free SPN1 and Imp-β1 are exported back to the cytoplasm and recycled[18,21]. Structural and biochemical studies have extensively characterized SPN1 binding to $m_3G$ cap, Imp-β1, and Exportin-1/CRM1[17,22–25]. However, only a few studies have demonstrated a role for SPN1 in the nucleo-cytoplasmic trafficking exclusively using transiently overexpressed protein[19,26]. Importantly, SPN1 has never been associated with a human disease and its endogenous function in model organisms remains to be characterized.

Here, we present a cohort of 18 children from 15 unrelated families across three continents, all carrying recessive hypomorphic $SNUPN$ disease-causing variants. These variants manifest in varying degrees of muscle weakness, elevated serum creatinine kinase levels, and are often accompanied by extramuscular phenotypes such as neurode-generation and cataracts. Through functional studies on primary fibroblasts, CRISPR/Cas9-mediated mutant cells, and muscle tissues from patients, we discover a previously unknown biological function of SPN1 in muscle homeostasis. Our findings demonstrate that SPN1 mutants fail to form homomers, leading to SPN1 cytoplasmic aggre-gation, defects in spliceosomal maturation ultimately resulting in mRNA mis-splicing. Specifically, we observe dysregulation of mRNA transcripts related to essential components of the DGC and ECM functions. Additionally, we characterize the disorganization of the cytoskeleton through protein filament aggregation, which further underscores the role of SPN1 in ECM-cytoskeleton crosstalk. These findings offer promising avenues for potential therapeutic interven-tions in muscular dystrophies.

## Results

### Identification of 18 patients with muscular dystrophy carrying biallelic $SNUPN$ germline variants

The index patient (II:1), is a female born to a non-consanguineous Kosovar family (F1) who developed normally until early childhood (Fig. 1a). At the age of five, she began experiencing frequent falls and developed an abnormal waddling gait with mild anterior pelvic tilt. Her symptoms progressively worsened over time. Examination of her upper and lower limb muscles revealed generalized muscle weakness slightly more pronounced on the left side (Fig. 1a). Radiological investigations showed a stable thoracic scoliosis and the electromyography (EMG) study revealed a myopathic pattern. Serum creatinine kinase (CK) levels measured at five and seven years old were elevated. Taken together, these clinical observations were consistent with a diagnosis of MD. Although an exhaustive molecular screening was performed, no patho-genic variant could be identified (Supplementary Notes). We carried out

a whole exome sequencing (WES) and uncovered a private germline homozygous disease-causing variant in $SNUPN$ (MIM607902) ($m1$, Chr15:c.926T>G, p.Ile309Ser), which had not been reported as homo-zygous in gnomAD, BRAVO/TOPmed and Exome Variant Server public databases. To date, with the exception of one de novo variant (p.Asp255Glu) associated with a disorder but of uncertain pathogenicity, no homozygous variants in $SNUPN$ have been ever associated with a disease[27]. Sanger sequencing on unaffected parents (F1-I:1 and F1-I:2) and brother (F1-II:2) confirmed that this private variant segregated with the disease, suggesting an autosomal recessive mode of inheritance. Importantly, p.Ile309Ser was predicted to be deleterious by most of the pathogenicity classifiers that were tested (CADD, DANN, M-CAP, FATHMM-MKL, SIFT, PolyPhen-2, and MutationTaster). In addition, Ile309 was highly conserved across vertebrate SPN1 orthologues which further supported its possible pathogenicity (Fig. 2c, Supplementary Data 2 and Supplementary Notes).

Subsequently, 17 additional cases from 14 unrelated families were identified through GeneMatcher[28] or Centogene (Rostock, Germany) in-house bio/databank[29] (Fig. 1c and Supplementary Data 1). The patient cohort in this study comprises a total of 10 males and 8 females with onset ages ranging from birth to 10 years old. 12 out of 18 cases presented symptoms before the age of two (67%), although the pro-gression of muscle weakness varied among them. All the patients presented with proximal muscular weakness in upper and lower limbs whereas distal and axial muscles were not always affected. Ten affected individuals experienced loss of independent ambulation over the years (Fig. 1a and Supplementary Fig. 1a, c), and one had never achieved walking (F3-II:4). CK levels were elevated in most cases, ranging between 1500 U/L to 8000 U/L in 16 patients (94%). Consistently, all patients who underwent EMG tests were diagnosed with myopathic pattern. Abnormal muscle fibers, particularly a predominance of type I fibers, were observed in all 10 skeletal muscle biopsies available for analysis. Hematoxylin & Eosin (H&E) and modified Gomori Trichrome (mGT) stainings showed a few hypotrophic fibers that were angulated with areas of cytoplasmic eosinophilia or intense trichrome staining in probands from F1, F2, and F12 families (Fig. 1d and Supplementary Notes). Observation of decreased Succinate Dehydrogenase (SDH) intensity in numerous muscle fibers of F1-II:1, F2-II:1, and F12-II:2 mutants suggested a reduction in the oxidative activity (Fig. 1d and Supplementary Notes). Furthermore, all assessed skeletal muscles demonstrated a significant increase in endomysium thickness, indi-cative of fibrosis. In addition, muscular ultrasound performed on two patients (F12-II:2 and F15-II:1) showed severe general muscle atrophy with homogeneously increased echogenicity indicative of fibrous remodeling (Supplementary Fig. 1d and Supplementary Notes). These data supported the clinical diagnosis and confirmed the typical mor-phological alterations of myofibers, and fibrosis observed in MDs. However, the overall histopathological features were not sufficient to delineate the specific subtype of MD.

Additionally, all patients displayed at least one extramuscular clinical manifestation with a high prevalence of central nervous system (78%), ocular (33%), skeletal (75%), respiratory system (61%) involve-ment (Fig. 1a, b and Supplementary Fig. 1a–c). Central nervous system imaging performed on nine patients revealed seven with either cere-bellar atrophy or thin corpus callosum. Six affected patients were diagnosed with congenital bilateral cataracts and underwent surgery. Most of the patients with skeletal involvement had joint contractures (78%) and vertebral column anomalies (75%) including scoliosis, rigid spine, and lordosis. Disease progression and severity varied among patients, 11 patients developed respiratory insufficiency requiring respiratory assistance; two (F3-II:4 and F7-II:1) died due to respiratory failure before the age of 15 years old. Targeted molecular screening failed to identify the genetic etiology in all patients of the cohort. Hence, trio or single WES was carried out, revealing unique biallelic potentially disease-causing variants in $SNUPN$, thereby confirming

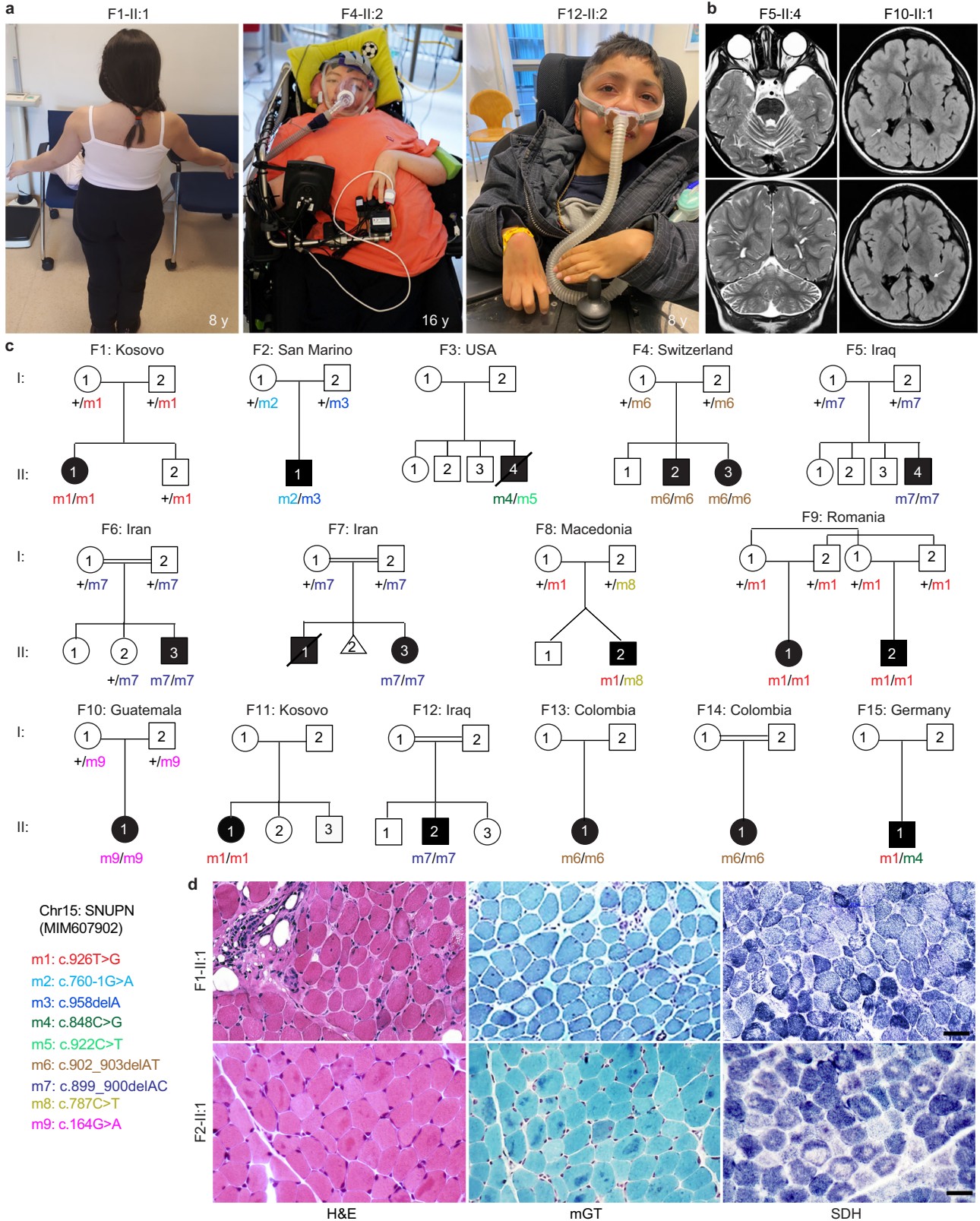

Mendelian recessive inheritance. In total, we identified four germline homozygous variants including the one in patient F1 (*m1*: c.926T>G; p.Ile309Ser) and three novel variants (*m6*: c.902_903delAT; p.Tyr301-Cysfs*29; *m7*: c.899_900delAC; p.Asp300Valfs*30; *m9*: c.164G>A; p.Arg55Gln). Additionally, we found four compound heterozygous variants (*m2*: c.760-1G>A (disrupted acceptor site) and *m3*: c.958delA; p.Met320ter; *m4*: c.848C>G; p.Ser283ter; and *m5*: c.922C>T; p.Gln308ter; *m8*: c.787C>T; p.Gln263ter and *m1*: c.926T>G; p.Ile309-Ser; *m1*: c.926T>G; p.Ile309Ser and *m4*: c.848C>G; p.Ser283ter) (Fig. 1c, d). Except for two missense variants (*m1* and *m9*) affecting highly conserved residues, all the others were either nonsense (*m3*, *m4*, *m5*, and *m8*), frameshift (*m6* and *m7*), or splice site (*m2*) variants. Thus, they were highly anticipated to result in protein alteration or truncation. Notably, patients with homozygous nonsense or frameshift variants

**Fig. 1 | Identification of 18 patients diagnosed with muscular dystrophy carrying biallelic *SNUPN* variants. a** Pictures of affected individuals: An 8-year-old affected girl from family F1 (F1-II:1) displaying an abnormal posture with left dissymmetry and difficulties in raising her arms. Affected sibling from family 4 (F4-II:2; 16-year-old) and family 12 (F12-II:2; 8-year-old) are permanently bound to wheelchair and artificial respiratory devices exhibiting severe disability. **b** Brain magnetic resonance imaging (MRI) of affected individuals from family 5 (F5-II:4; 22-month-old) and family 10 (F10-II:1, 3-year-old) revealed cerebellar atrophy and white matter hyperintensities (white arrows), respectively. **c** Pedigree of fifteen families segregating autosomal recessive muscular dystrophy. Double lines indicate a consanguineous marriage. Filled black symbols and crossed symbols indicate affected and deceased individuals, respectively while triangle indicates miscarriage. Compound heterozygous variants are presented based on their parental origin and *SNUPN* variant coordinates are provided. **d** Immunohistochemistry of skeletal muscle from patient F1-II:1 and F2-II:1: Hematoxylin and eosin (H&E) and modified Gomori Trichrome (mGT) images revealed muscle fibers with an atrophic appearance, displaying heterogeneity in size and shape. Additionally, some fibers exhibited a centralized nucleus and intense cytoplasmic staining. Succinate dehydrogenase (SDH) staining demonstrated a reduction in oxidative enzymatic activity, as indicated by reduced staining, in numerous muscle fibers in both patients. Scale bar, 100 μm. The images shown are representative of stainings on single muscle biopsy in each indicated patient. No independent replicates were performed.

displayed more severe muscle weakness whereas those with missense variant in C-terminus had milder symptoms. In this cohort, frameshift variant (*m7*) was found in four independent families. Another frameshift variant (*m6*) was annotated in Exome Variant Server in twenty-two individuals (minor allele frequency ≤10⁻⁴) at the homozygous state, but its clinical significance was unknown, suggesting the existence of additional undiagnosed cases. Of these nine *SNUPN* variants, four (*m2*, *m3*, *m7*, and *m8*) have not been previously annotated in public databases (gnomAD, BRAVO/TOPmed, and Exome Variant Server). Runs of homozygosity (ROH) analysis revealed founder haplotypes in families harboring *SNUPN* pathogenic variants *m1* (F1 and F9), *m6* (F13 and F14), and *m7* (F6, F7, and F12) (Supplementary Fig. 1e). Exhaustive clinical and genetic findings for all 18 patients are provided in Fig. 1, Table 1, Supplementary Data 1, and Supplementary Notes.

In summary, we have discovered nine distinct recessive disease-causing *SNUPN* variants among 18 affected individuals exhibiting a broad range of muscular weakness, that seem directly correlated with the impact of the pathogenic variants. Taken together, these findings provide strong clinical, histological, and genetic evidence to support the causal association between *SNUPN* variants and a previously uncharacterized subtype of muscular dystrophy disorder.

## The C-terminus of SPN1 is essential for its cellular localization

*SNUPN* consists of 9 exons (Fig. 2a), and codes for SPN1, a 360-amino acid protein that contains two characterized domains, the Importin β binding (IBB) and the TMG binding regions (Fig. 2b). Remarkably, eight out of nine variants cluster in the last coding exon of *SNUPN* encompassing only 198 nucleotides. These variants result in missense, frameshift, truncation, or splicing disruption, which alter the C-terminus of the protein (Fig. 2a, b). Arg55Gln from F10-II:1 is the only mutated residue located in the N-terminus. Notably, there is a high degree of sequence similarity in SPN1 protein, particularly within the C-terminus region, among multiple organisms. In addition, Arg55 and Ile309, the two missense residues are highly conserved between vertebrate species (Fig. 2c). These data indicated that the alterations of the C-terminus of SPN1 may lead to pathogenic effects.

According to the Human Protein Atlas (HPA), skeletal muscle and cerebral cortex, which are the main affected organs in patients, express high levels of *SNUPN* RNA (Supplementary Fig. 2a) and SPN1 protein (Supplementary Fig. 2b), suggesting a fundamental role of *SNUPN* in these tissues.

To investigate the potential pathogenicity of *SNUPN* variants, and their consequences on the level and localization of endogenous SPN1, we performed quantitative PCR (RT-qPCR), western blotting, and immunofluorescence staining on human primary mutant and wild-type (WT) fibroblasts. Primary dermal cutaneous fibroblasts were isolated from affected patients (F1^{m1/m1}, F2^{m2/m3}, F4-II:2^{m6/m6}, F4-II:3^{m6/m6}, and F10^{m9/m9}) and three wildtype (WT) individuals. RT-qPCR analysis revealed that levels of *SNUPN* transcripts were not significantly changed in F1^{m1/m1}, F2^{m2/m3}, F4-II:2^{m6/m6}, F4-II:3^{m6/m6}, and F10^{m9/m9} relative to WT fibroblasts (Supplementary Fig. 2c). The level of SPN1 in the whole protein lysate was evaluated by western blot using a pan anti-SPN1 antibody. As expected, two shorter and less intense bands corresponding to the truncated SPN1 mutants were observed in the F2^{m2/m3} patient (Fig. 2d, lane 5). Overall, a significant reduction in the total amount of SPN1 protein in mutants was observed except for F10^{m9/m9} (Fig. 2d, e). In the cytoplasm of WT cells, SPN1 staining displayed a more organized and evenly-distributed pattern whereas the staining appeared to be more intense and localized closer to the nuclei of the patients' fibroblasts (Fig. 2f and Supplementary Fig. 2d). The analyses of SPN1 perinuclear fluorescence intensity confirmed significant augmentation between F1^{m1/m1}, F2^{m2/m3}, F4-II:3^{m6/m6}, F10^{m9/m9}, and three WT cells (Supplementary Fig. 2e).

To investigate further the discrepancy between SPN1 western blot and immunofluorescence results, we created two HeLa CRISPR/Cas9-mediated SPN1 mutant lines. The first named, SPN1^{sgEx2}, harbors two mutations at the beginning of the gene (Exon 2). The second line, named SPN1^{sgEx9}, carries three types of deletions in the final exon (Exon 9) that closely resemble the variants found in our patients, notably *m6* and *m7*. In western blot, both SPN1^{sgEx2} and SPN1^{sgEx9} HeLa mutant cells

## Table 1 | Clinical summary of patients with *SNUPN* pathogenic variants

| Clinical synopsis | HPO terms | Ratio | Percentage |
|---|---|---|---|
| Female patient | | 8/18 | 44% |
| Male patient | | 10/18 | 56% |
| Consanguinity | | 4/18 | 22% |
| Onset before ambulation | | 12/18 | 67% |
| Deceased | | 2/18 | 12% |
| Muscular phenotypes | | | |
| Proximal upper limb weakness | HP:0008997 | 18/18 | 100% |
| Distal upper limb weakness | HP:0008959 | 12/16 | 75% |
| Proximal lower limbs weakness | HP:0008994 | 18/18 | 100% |
| Distal lower limb weakness | HP:0009053 | 16/18 | 89% |
| Axial weakness | HP:0003327 | 16/18 | 89% |
| Non-ambulatory (on last examination) | HP:0002540 | 11/18 | 61% |
| CK high (>500) | HP:0030234 | 16/17 | 94% |
| Myopathy (EMG) | HP:0003198 | 11/11 | 100% |
| Muscular histology | | | |
| Abnormal muscle fiber | HP:0004303 | 10/10 | 100% |
| Endomysial fibrosis | HP:0100297 | 9/9 | 100% |
| Neurological defects | | | |
| Cerebellar atrophy | HP:0001272 | 5/9 | 56% |
| Thin corpus callosum | HP:0200012 | 3/7 | 43% |
| Other features | | | |
| Cataract | HP:0000518 | 6/18 | 33% |
| Respiratory insufficiency | HP:0002093 | 11/18 | 61% |
| Abnormal vertebral column | HP:0000925 | 12/16 | 75% |
| Limb joint contracture | HP:0003121 | 14/18 | 78% |

*CK* creatinine kinase, *EMG* electromyography, *HPO* Human Phenotype Ontology.

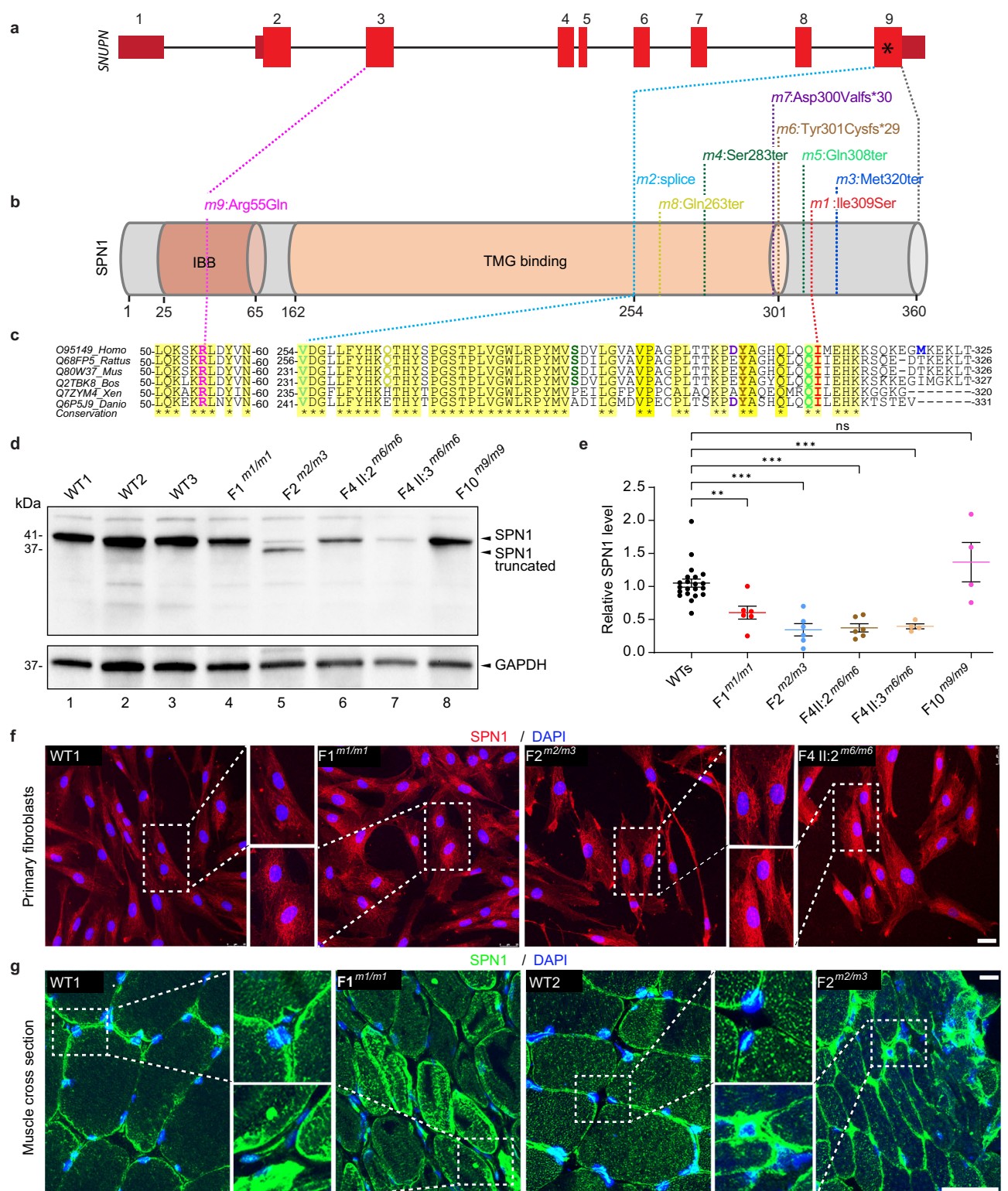

displayed a significant reduction of total endogenous SPN1 protein level (Supplementary Fig. 2f). However, we observed a notable increase in SPN1 signal in SPN1[sgEx9] but not in SPN1[sgEx2], in their cytoplasm when using immunofluorescence (Supplementary Fig. 2g). Together, these findings corroborate our results with patients' cells carrying mutations in the C-terminal region. Thus, we suspect that SPN1 mutant proteins with C-terminal alterations may be particularly unstable in solution, despite our stringent measures to prevent degradation. Our observation aligns with prior findings of SPN1

degradation in solution as opposed to its higher stability when it is in a complex state[22].

SPN1 expression was also examined in human skeletal muscle tissue sections by immunofluorescence (Fig. 2g). While we observed a mild punctuated staining inside the muscle fiber in WT samples, the staining appeared stronger, with numerous subsarcolemmal densities in F1[m1/m1] and F2[m2/m3] mutant muscle tissue sections. Hence, the SPN1 mutant distribution pattern in the cytoskeleton and its perinuclear accumulation further supported our findings in mutant fibroblasts.

**Fig. 2 | The C-terminus of SPN1 is essential for cellular localization. a** Schematic diagram depicting the genomic structure of *SNUPN* in humans which consists of nine exons. **b** SPN1 protein contains two conserved domains: Importin β binding domain (IBB) (dark brown) and trimethylguanosine ($m_3$G)-cap-binding domain (TMG binding) (light brown). Position of the nine pathogenic germline variants is indicated in both *SNUPN* DNA and SPN1 protein. Nearly all pathogenic variants (*m1-m8*) are clustered in the C-terminus region between residues 254–320 and only the *m9* pathogenic variant is located in the N-terminus. **c** Multiple sequence alignment of SPN1 protein from different species. Yellow shading indicates fully conserved regions where the pathogenic variants are located. **d** Immunoblot analysis showing significant decrease of endogenous SPN1 in patients F1$^{m1/m1}$, F2$^{m2/m3}$, F4-II:2$^{m6/m6}$, and F4-II:3$^{m6/m6}$ compared to WTs ($n = 3$) fibroblasts. Note that the germline compound heterozygous pathogenic variants F2$^{m2/m3}$ created two truncated forms of SPN1. F10$^{m9/m9}$ shows a mild but not significant increase of SPN1. GAPDH served as loading control. **e** Scatter plot showing quantification analysis of SPN1 protein level in five patients compared to three WTs fibroblasts lines using independent immunoblots. $n = 18$ (WTs), 6 (F1$^{m1/m1}$, F2$^{m2/m3}$, F4-II:2$^{m6/m6}$), 4 (F4-II:3$^{m6/m6}$, F10$^{m9/m9}$) immunoblots. Each dot represents one quantification. Fold change relative to WTs is plotted as mean ± SEM. ns nonsignificant; **$P = 0.0076$; ***$P = 0.0007$ (Ordinary one-way ANOVA and two-tailed unpaired *t*-test). Source data are provided as a Source Data file. **f** Images of immunofluorescence showing endogenous SPN1 (red) accumulation around the nucleus in three patients' fibroblast lines (F1$^{m1/m1}$, F2$^{m2/m3}$, and F4-II:2$^{m6/m6}$) compared to WT. Nuclei were labeled with DAPI (blue). $n = 3$ independent stainings. Scale bar, 10 µm. **g** Images of immunofluorescence on muscle sections showing aggregation of SPN1 along the sarcolemma and within the sarcoplasm in mutant muscle sections (F1$^{m1/m1}$ and F2$^{m2/m3}$) compared to healthy WTs. $n = 2$ independent stainings. Scale bar, 50 µm.

Taken together, these data lead us to hypothesize that *SNUPN* plays a central role in maintaining muscle integrity, and that alterations in the C-terminus region of SPN1 are likely to be a key determinant in the pathogenesis of the disease.

## Intact SPN1 C-terminus is required for its oligomerization

Pathogenic variants (*m1-m8*) located between SPN1 residues 254 and 320 are expected to alter either the TMG binding domain or a C-terminus domain which has not yet been characterized (Fig. 2b). Therefore, we utilized predicted structural models and computational tools to further understand how these pathogenic variants would impact the structure of SPN1 (Fig. 3a).

The structural prediction of WT SPN1 shows a well-defined alpha helix in the C-terminal domain between Gly303 and Glu322, a region that spans all our SPN1 pathogenic variants. Remarkably, this alpha-helix appears to be partially or entirely disrupted in all the mutant models (Fig. 3a). Additionally for *m9*:SPN1$^{Arg55Gln}$, the only mutant with a N-terminus pathogenic variant, the voluminous and positively charged arginine residue is substituted with glutamine, a smaller and neutral residue, which is expected to alter the structure around the N-terminus.

Since alpha helices are well known to play a significant role in structural stability of multiple protein subunits, we anticipated that alterations in the C-termini of SPN1 mutants could impair their ability to form oligomeric complexes. Our speculation was supported by previous data demonstrating an intramolecular interaction between SPN1 N- and C-termini[26]. To further test this hypothesis, Myc- and Flag-SPN1 constructs were co-expressed in HEK293T cells, and their protein extracts were subjected to Flag immunoprecipitation (Fig. 3b and Supplementary Fig. 3a, b). We first observed that Flag-SPN1$^{WT}$ was able to pull down specifically Myc-SPN1$^{WT}$ (Fig. 3b, lane 2), confirming that SPN1 molecules can associate with each other. Interestingly, the most severe mutants, Flag-SPN1$^{Tyr301Cysfs*29}$ and Flag-SPN1$^{Asp300Valfs*30}$ were unable to immunoprecipitate their Myc-tagged counterparts (Fig. 3b, lanes 4 and 5) whereas Flag-SPN1$^{Ile309Ser}$ pulled down Myc-SPN1$^{Ile309Ser}$ less efficiently (Fig. 3b, lane 3 and Supplementary Fig. 3a, lane 2) compared to SPN1$^{WT}$ (Fig. 3b, lane 2 and Supplementary Fig. 3a, lane 1) and SPN1$^{Arg55Gln}$ (Supplementary Fig. 3a, lane 3). Moreover, Myc-SPN1 frameshift mutants *m6* and *m7* were unable to interact with Flag-SPN1$^{WT}$, unlike Myc-SPN1 WT and *m1* and *m9* (Supplementary Fig. 3b). These data demonstrate that an intact C-terminal region is required for SPN1 self-interaction.

To further ascertain this finding, Myc-tagged SPN1$^{WT}$, SPN1$^{Ile309Ser}$, SPN1$^{Tyr301Cysfs*29}$, and SPN1$^{Asp300Valfs*30}$ mutant constructs were overexpressed in HeLa cells. We observed the presence of additional bands at 100 and 150 kDa in the cytoplasmic extract of overexpressed SPN1$^{WT}$ cells without treatment with dithiothreitol (DTT), a commonly used thiol-reducing reagent (Fig. 3c, lane 1 and Supplementary Fig. 3c, lane 2). Remarkably, the strongest band at 150 kDa band appeared slightly reduced in SPN1$^{Ile309Ser}$ (Fig. 3c, lane 2) and was completely absent in samples overexpressing either SPN1$^{Tyr301Cysfs*29}$ and SPN1$^{Asp300Valfs*30}$, the two most severe mutants (Fig. 3c, lanes 3 and 4 and Supplementary Fig. 3c, lanes 4 and 5). Treatment of overexpressed SPN1$^{WT}$ and SPN1$^{Ile309Ser}$ samples with DTT resulted in the complete disruption of oligomeric complexes (Fig. 3c, lanes 5 and 6). Other mutant samples treated with DTT did not show any significant changes compared to untreated ones (Fig. 3c, lanes 7 and 8). Thus, SPN1 oligomers appeared to be stable structures resistant to dissociation by SDS. However, treatment with DTT, which is known to break disulfide bonds, was able to dissociate SPN1 oligomers efficiently, suggesting that the molecules were covalently linked.

Next, we assessed the oligomeric state of endogenous SPN1 using an anti-SPN1 pan antibody by examining cytoplasmic fractions from WT, F1$^{m1/m1}$, F2$^{m2/m3}$, F4-II:3$^{m6/m6}$, and F10$^{m9/m9}$ mutant fibroblasts without reducing agents. As expected, the monomer around 41 kDa was detected in all the samples. However, we observed that the signal of the higher band at around 150 kDa was systematically decreased the most for the SPN1 mutant with the strongest C-terminal disruption, whereas the band at around 100 kDa was not significantly reduced (Fig. 3d). In addition, no significant difference was noted for the N-terminal mutant F10$^{m9/m9}$. Remarkably, these two upper bands were never observed in WT nuclear fractions.

Taken together, these data demonstrate that SPN1 exists as a higher complex in the cytoplasm and strongly suggest the formation of homo-dimers or tetramers, as indicated by the size of the observed oligomers (100 and 150 kDa). However, it appears that the C-terminal region is more essential for tetramer than for dimer formation. Interestingly, the prediction of coiled-coil SPN1 oligomeric state with LOGICOIL pointed out a region in the C-terminus spanning amino acids Gly303 to Leu339 which exhibited a higher probability of forming a tetramer rather than a dimer. Next, we used this C-terminal region to generate a predicted tetramer on AlphaFold2-Colab using the model with the best Predicted Aligned Error (PAE) where two antiparallel helices interact with each other in a parallel orientation. It appeared that the Ile309 residue was predicted to be directly involved in these hydrophobic interactions (Fig. 3e and Supplementary Fig. 3e).

Consequently, we decided to further examine the interaction between the WT and C-terminal mutant regions by generating several constructs with only the C-terminal region, encompassing all our pathogenic variants, and tagged with either Myc or Flag (Myc-SPN1$^{254-360}$ and Flag-SPN1$^{254-360}$). Firstly, the WT Flag-SPN1$^{254-360}$ was able to interact with its counterpart Myc-SPN1$^{254-360}$, confirming that this region is sufficient to form an oligomer (Supplementary Fig. 3d, lane 1). Secondly, the *m1* SPN1$^{254-360}$ was also able to interact efficiently with WT SPN1$^{254-360}$ fragment (Fig. 3f, lane 1), whereas *m6* and *m7* SPN1$^{254-328}$ constructs could not interact with the WT C-terminal fragment (Fig. 3f, lanes 2 and 3). Notably, the binding of the *m1* SPN1$^{254-360}$ fragment with itself was significantly less efficient than its binding to the WT C-terminal fragment (Fig. 3f, lane 4), confirming the likely involvement of the Ile309 residue in oligomerization.

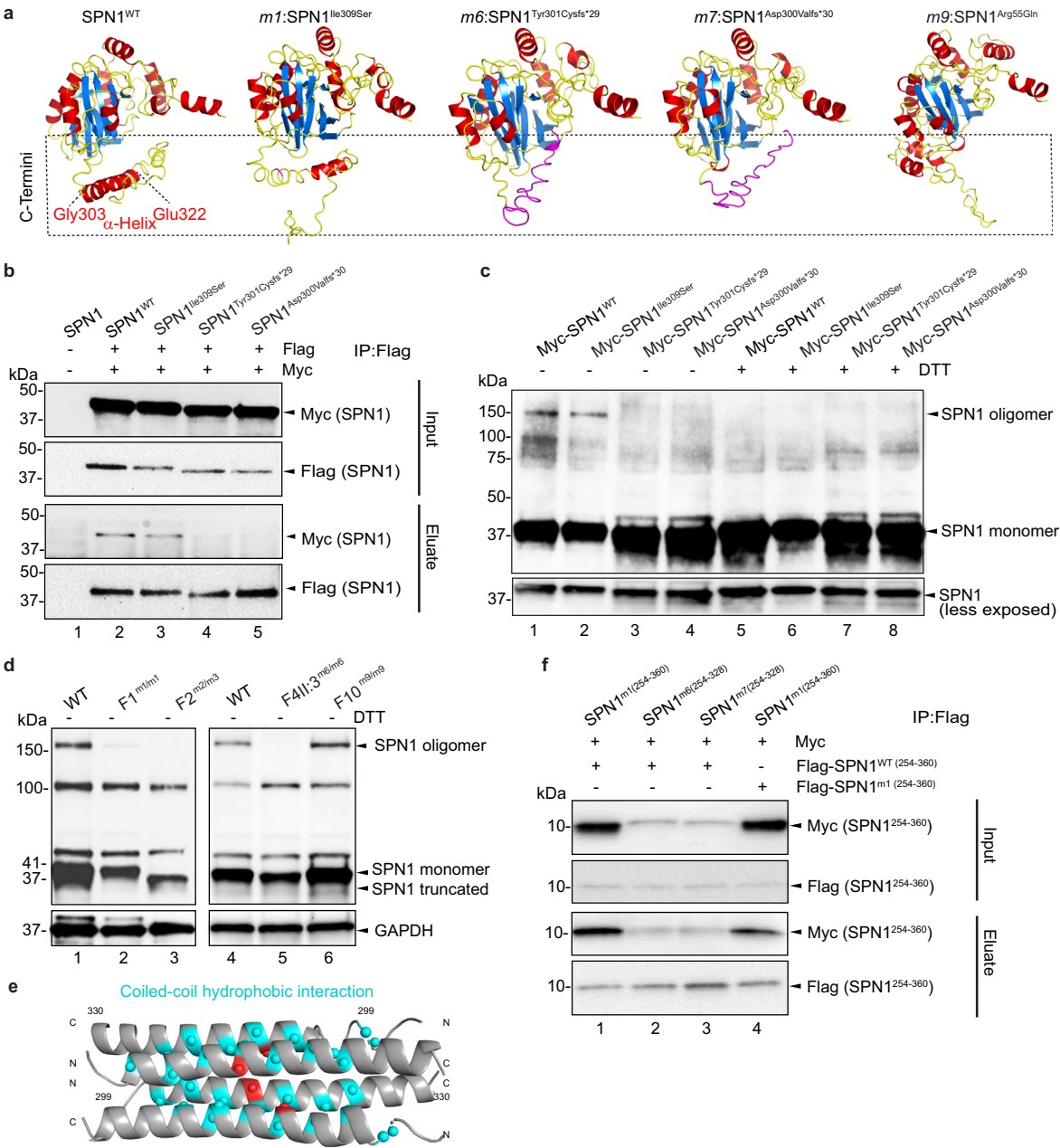

**Fig. 3 | Intact SPN1 C-terminus is required for its oligomerization. a** Comparison between the 3D structure predictions of SPN1 WT and *m1, m6, m7, m9* pathogenic variants. Protein is shown colored based on the secondary structure, with alpha helices in red, beta sheets in blue, and loops in yellow. Mutated amino acids in *m6* and *m7* are shown in magenta. C-termini are indicated in the dotted box. **b** Co-immunoprecipitation (co-IP) assay performed in HEK293T cells co-transfected with Flag- and Myc-tagged SPN1 using Flag beads. Input and eluate samples blotted with anti-Myc or anti-Flag antibodies reveal SPN1 interaction between SPN1^WT^ (Lane 2) and SPN1^Ile309Ser^ (Lane 3) but not with Flag-SPN1^Tyr301Cysfs*29^ and Flag-SPN1^Asp300Valfs*30^ (Lanes 4 and 5). *n* = 3 independent experiments. **c** Immunoblot analysis of HeLa cells cytoplasmic fractions transfected with WT or mutant Myc-tagged SPN1 constructs. In absence of DTT, two bands at 41 and 150 kDa were observed in samples transfected with Myc-SPN1^WT^ and Myc-SPN1^Ile309Ser^ whereas bands at 150 kDa disappeared totally in lysate transfected with Myc-SPN1^Tyr301Cysfs*29^, and Myc-SPN1^Asp300Valfs*30^ constructs. Upon treatment with DTT, higher bands were not detected in either samples. *n* = 2 independent experiments. **d** Immunoblot analysis of endogenous SPN1 in cytoplasmic fractions of fibroblast showing SPN1 specific bands at 41 kDa and 150 kDa. The upper band was unchanged in F10^m9/m9^, very faint in F1^m1/m1^, and completely disappeared in F2^m2/m3^ and F4-II-2^m6/m6^ samples. GAPDH served as loading control. *n* = 3 independent experiments. **e** Model of tetramer formation with C-terminal region (residues 299-330) based on AlphaFold2 Colab and Socket2 coiled-coil predictions. Ile309 is highlighted in red and residues predicted to form hydrophobic interactions in blue. **f** Co-IP assay using Flag beads and performed with HEK293T cells co-transfected with Flag- and Myc-tagged SPN1 C-termini fragments. Input and eluate samples blotted with anti-Myc or anti-Flag antibodies reveal interaction between Myc-SPN1^WT(254-360)^ and Flag-SPN1^m1(254-360)^ (Lane 1) but not with Flag-SPN1^m6(254-328)^ and Flag-SPN1^m7(254-328)^ (Lanes 2 and 3). Self-interaction of SPN1^m1(254-360)^ is reduced (Lane 4). *n* = 3 independent experiments. Source data are provided as a Source Data file.

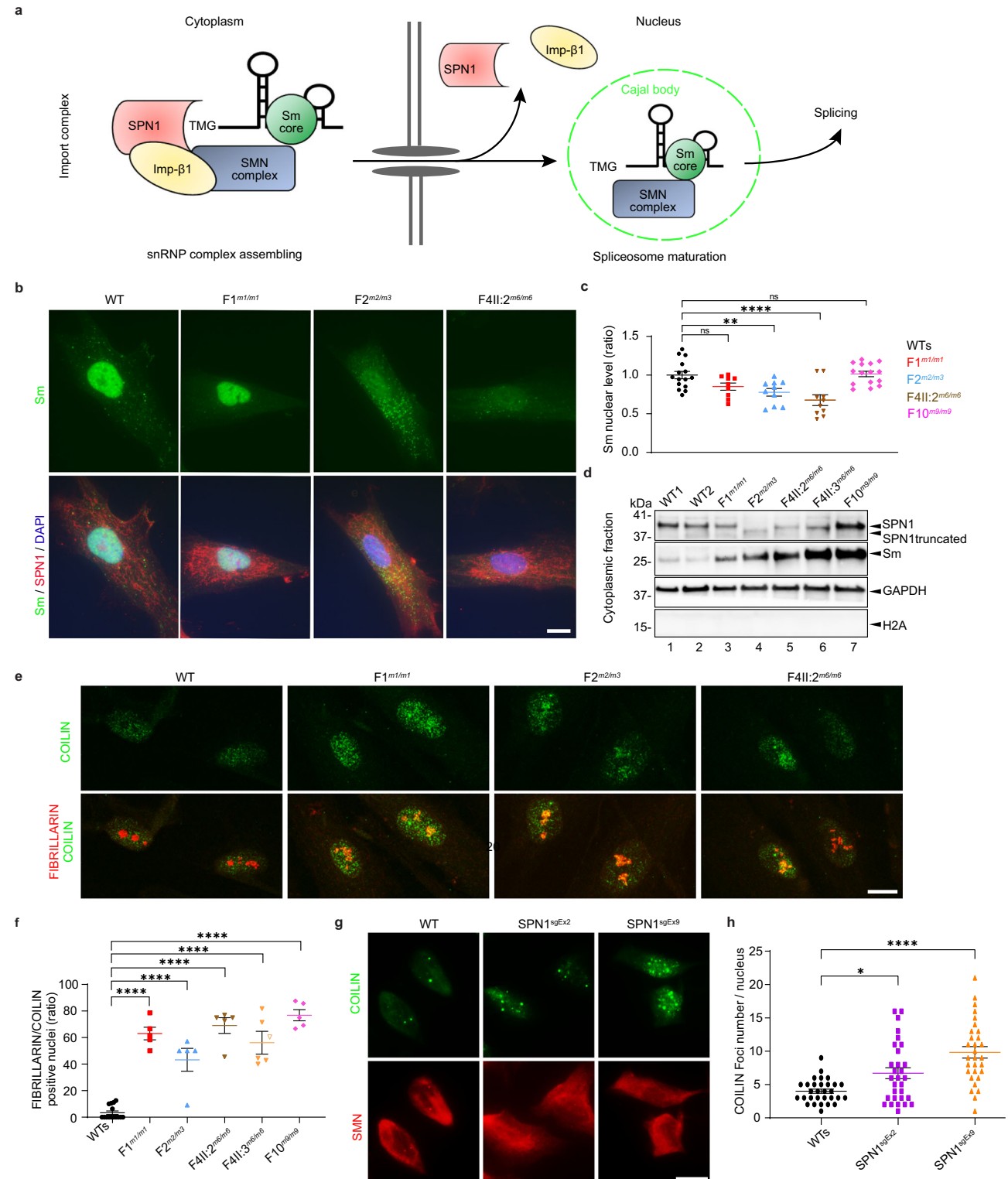

In summary, our data demonstrate that SPN1 molecules are able to form homomers which are disrupted when pathogenic variants occur in the SPN1 C-terminus domain. This finding lends further credence to the central role of the C-terminus region in proper SPN1 stability and function.

## SPN1 is necessary for proper spliceosomal maturation

The nuclear import of snRNPs is facilitated by SPN1's direct binding to the m₃G cap, Imp-β1 via its IBB domain, and the SMN complex, as demonstrated in numerous studies[19] (Fig. 4a). Given this established

role, we anticipated that the inability of SPN1 to form homomers would hinder the assembly of the snRNP import complex and subsequently prevent its nuclear translocation. To investigate this hypothesis, we employed two different approaches to assess the alteration of Sm proteins, core components of the snRNPs complex, in its nucleo-cytoplasmic transport.

First, we observed a significant reduction in Sm proteins levels in the nuclei of F2$^{m2/m3}$ and F4-II:2$^{m6/m6}$ mutant fibroblasts compared to WT cells, as demonstrated by immunofluorescence staining (Fig. 4b, c). The same observation was confirmed in the nuclei of both *SNUPN*

**Fig. 4 | SPN1 C-terminus is necessary for proper spliceosomal maturation.**
**a** Schematic of snRNPs biogenesis and maturation pathways. In cytoplasm, snRNP occurs in three steps: (1) snRNP core formation, (2) trimethylguanosine $m_3G$ cap (TMG) addition, (3) association of SPN1 with TMG, Imp-β1, and SMN complex. In nucleus, the complex releases the snRNP-SMN complex to Cajal bodies to complete spliceosome maturation. **b** Immunofluorescence images of fibroblasts showing reduced nuclear Sm (green) in $F2^{m2/m3}$ and $F4\text{-}II:2^{m6/m6}$. SPN1 (red) aggregation around the nucleus is evident in all the mutants compared to WT. DAPI labeled nuclei (blue). Scale bar, 10 μm. **c** Quantitative analysis showing significant decrease of nuclear Sm in $F2^{m2/m3}$, $F4^{m6/m6}$ compared to two WT (WTs) fibroblast lines. Note that mild reduction in $F1^{m1/m1}$ is not significant. $n = 15$ (WTs), 9 ($F1^{m1/m1}$), 10 ($F2^{m2/m3}$), 10 ($F4\text{-}II:2^{m6/m6}$), 15 ($F10^{m9/m9}$) cells. Fold change relative to WTs is plotted as mean ± SEM. ns nonsignificant; **$P = 0.0088$; ****$P = 0.0001$ (two-tailed unpaired $t$-test). **d** Immunoblot analysis of fibroblast cytoplasmic fractions showing increased Sm in all patients (Lanes 3-7) compared to WTs samples (Lanes 1–2). GAPDH and

H2A served as cytoplasmic and nuclear markers, respectively. $n = 3$ independent stainings. **e** Immunofluorescence images showing co-localization of COILIN (green) and FIBRILLARIN (red) in $F1^{m1/m1}$, $F2^{m2/m3}$, $F4\text{-}II:2^{m6/m6}$ compared to WT fibroblasts. Scale bar, 10 μm. **f** Scatter plot showing ratio of positive nuclei stained with COILIN and FIBRILLARIN in fibroblasts from five patients compared to two WT. $n = 59$ (WTs), 53 ($F1^{m1/m1}$), 36 ($F2^{m2/m3}$), 55 ($F4\text{-}II:2^{m6/m6}$), 44 ($F4\text{-}II:3^{m6/m6}$), 42 ($F10^{m9/m9}$) total nuclei. Each dot represents the average from one image ($n = 5$). Fold change relative to WTs plotted as mean ± SEM. ****$P < 0.0001$ (two-tailed unpaired $t$-test). **g** Immunofluorescence images showing increased number of COILIN (green) foci in SPN1 HeLa mutant lines compared to WT. **h** Scatter plot representing quantification of COILIN foci in SPN1 HeLa mutant lines compared to WT. Each dot represents a cell ($n = 30$). Fold change relative to WTs plotted as mean ± SEM. *$P = 0.0162$; ****$P < 0.0001$ (two-tailed unpaired $t$-test). Source data are provided as a Source Data file.

mutant HeLa lines generated via CRISPR-Cas9 (Supplementary Fig. 4a). These data are further supported by a significant increase of Sm protein levels in the cytoplasmic fraction of all five available mutant fibroblast, suggesting cytoplasmic retention (Fig. 4d). Collectively, these data demonstrate the cytoplasmic retention of Sm proteins, which results from a defective assembly of the pre-import snRNP complex in all patients, with a significant reduction in the nuclear import of Sm proteins in the $F2^{m2/m3}$ and $F4\text{-}II:2^{m6/m6}$ mutants.

Since the last step of spliceosome maturation occurs in the Cajal bodies (CBs), we next used an anti-COILIN antibody, a well-established component of CBs, to further investigate CB formation by immunofluorescence[30,31]. While it is known that CBs are not distinguishable in WT fibroblasts, we noticed that in all five mutant fibroblasts, the COILIN staining appeared very bright and dispersed throughout the nucleoplasm (Fig. 4e and Supplementary Fig. 4b). Indeed, by using FIBRILLARIN, a dense fibrillar marker of nucleolus, we demonstrated a significant relocalization of COILIN to the nucleoli of all five mutants (Fig. 4e, f and Supplementary Fig. 4b). We further confirmed these results in the two *SNUPN* HeLa mutant cell lines (Supplementary Fig. 4c). Interestingly, it has been previously demonstrated that RNAi targeting SPN1 in HeLa cells resulted in the same phenotype[31]. Finally, we utilized the *SNUPN* HeLa mutant cell lines to assess the integrity of CBs. As expected, COILIN staining revealed a significant increase in COILIN foci localized throughout the nucleolus in both mutant lines (Fig. 4g, h).

Taken together, our results on primary fibroblasts and HeLa mutant cell lines demonstrated that endogenous SPN1 is required for the import of snRNPs into nuclei, assembly of CBs, and consequently for proper spliceosomal maturation.

### SPN1 deficiency dysregulates ECM-associated key components
Collectively, our previous data prompted us to hypothesize that SPN1 deficiency would result in an aberrant nuclear alternative splicing. Thus, we conducted a deep transcriptome sequencing followed by comprehensive analyses to identify potential changes in mRNA targets.

Importantly, our clustering analysis on RNA-seq data using the three SPN1 available mutant ($F1^{m1/m1}$, $F2^{m2/m3}$, and $F4\text{-}II:2^{m6/m6}$) and two WT fibroblasts showed clear separation between mutant and WT groups (Supplementary Fig. 5a). A bioinformatic differential splicing analysis revealed a significant number of mRNA mis-splicing (AS) events including 519 skipped exons (ES) and 88 retained introns (RI) in mutants (Supplementary Fig. 5b). To further support the validity of the collected data, our splicing analysis revealed an intron retention between exons 8 and 9 for $F2^{m2/m3}$ *SNUPN* mutant, confirming that the c.760-1G>A heterozygous splice site variant detected in the WES resulted in an impaired mRNA splicing (Supplementary Fig. 5c). Next, we focused our attention on the exon skipping events according to the level of exon inclusion as illustrated in the volcano plot (Fig. 5a). Remarkably, a significant number

of mis-spliced genes in the three mutants were related to muscle or extracellular matrix (ECM) ontologies. One of these genes, *SGCA* (Alpha-Sarcoglycan), is a central component of the sarcolemma and is known to ensure the stability of the muscle fibers by linking the cytoskeleton to the ECM. Deep visualization of the splice junctions of this gene with Sashimi plots showed consistent skipping of exon 3 in all three mutants compared to the two WTs (Fig. 5b). Other ECM-related genes, *POSTN* (Periostin) and *COL6A2* (Collagen, type VI, alpha), are also appealing candidates as they have been directly linked to myogenesis and muscular dystrophy, respectively[32,33]. Other members of the collagen family (*COL6A1*, *COL6A2*, and *COL12A1*) and the myosin light chain 9 (*MYL9*) related to muscle contraction were also found to be significantly misspliced in the three mutants. Moreover, we noted that *DUSP18*, an atypical dual specificity phosphatase, was among the most significantly misspliced genes and showed exon 2 skipping in all three mutants (Supplementary Fig. 5d). While little is known about this gene, *DUSP18* was recently reported to be involved in regulating the aggregation of Ataxin-1 (ATXN1), a protein recurrently associated with neurodegenerative diseases[34].

Our analysis of differentially expressed genes (DEGs) uncovered a total of 88 significantly dysregulated genes (Log2 (fold change) > 1.5) in all three mutants (Supplementary Fig. 5e). Notably, gene ontology enrichment analyses revealed that for both down- and upregulated genes, the main affected biological pathways were linked to ECM organization, cell adhesion and differentiation (Fig. 5c and Supplementary Fig. 5f). Among the most strongly downregulated genes, we noticed several attractive candidates related to the sarcolemma and basal lamina. The most significantly downregulated gene, *LAMA5* (Laminin subunit alpha 5) is a constituent of the basal lamina which plays a major role in the ECM maintenance in mammalian tissues. Remarkably, *SGCA*, a component of DGC, which was found to be differentially mis-spliced, was also significantly downregulated in mutant cells (Fig. 5d). RT-qPCR assays on independent samples for orthogonal validation confirmed a significant downregulation of *SGCA* and *LAMA5* in all five mutant fibroblasts compared to the WTs (Fig. 5e). Conversely, integrin beta 2 (*ITGB2*) and secreted frizzled-related protein 2 (*SFRP2*), components associated with ECM remodeling[35], were significantly upregulated (Fig. 5d, e). Interestingly, the inhibition of ITGB1 was recently shown to improve the CMD phenotype in a zebrafish model[36]. We also noted a reduction of *COL4A5*, an isoform of type IV collagen, the major collagenous component of the basal lamina directly connected to laminin at the interface of the sarcolemma (Fig. 5d). Importantly, this result was corroborated by a marked reduction of endogenous COL IV protein in the *SNUPN* mutant fibroblasts (Supplementary Fig. 5g), as well as in the two *SNUPN* mutant HeLa lines (Supplementary Fig. 5h). Immunofluorescence was used to explore SGCA and COL IV markers in muscle cryosections from two available patients, $F1^{m1/m1}$ and $F2^{m2/m3}$. The sarcolemma from both patients showed significantly reduced levels for both markers around many

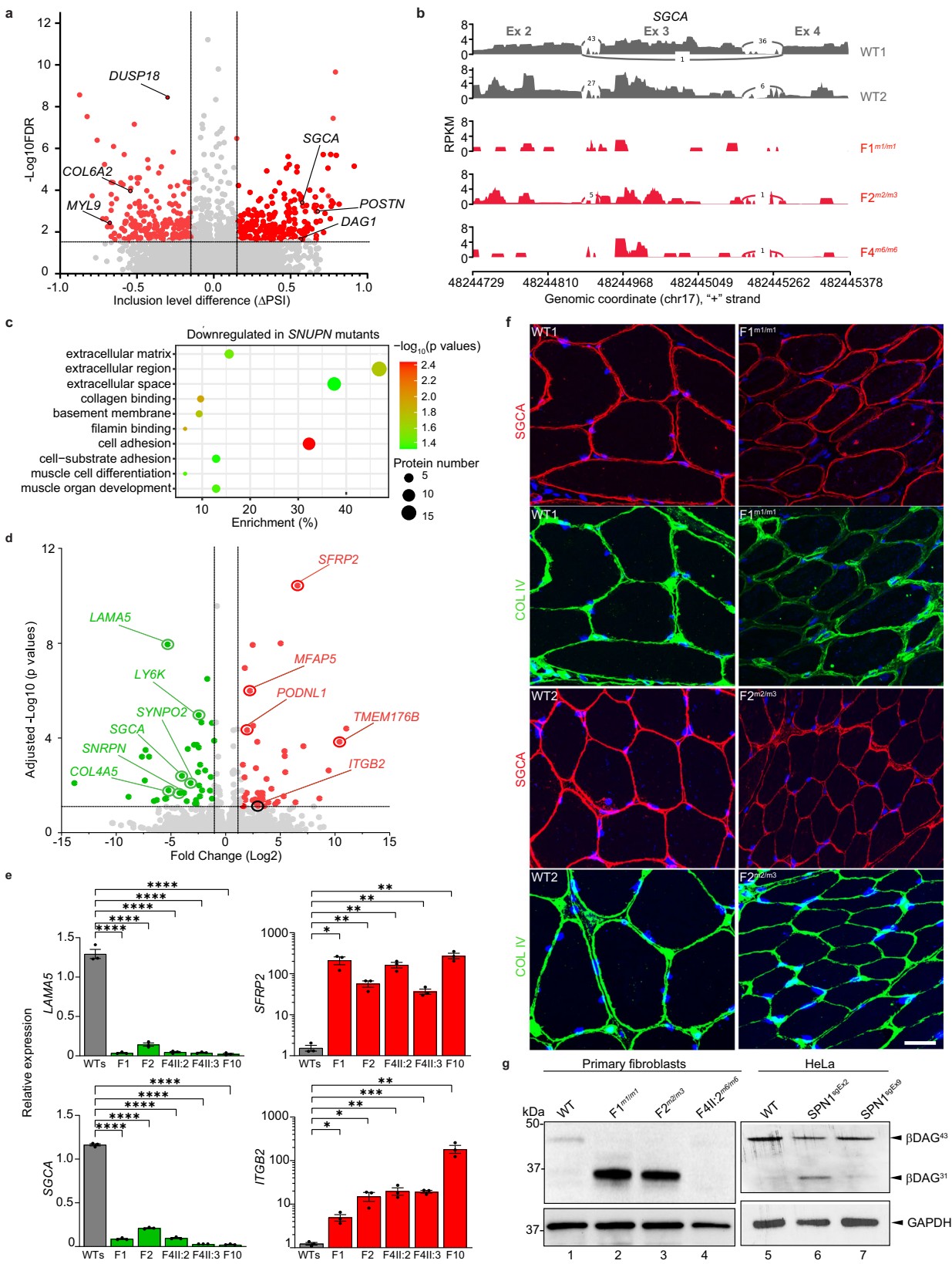

muscle fibers (Fig. 5f), suggesting a shared mechanism between fibroblasts and muscle.

To better understand the link between these ECM players, we investigated whether the decrease of SGCA observed in mutant fibroblasts and muscle tissue would impact dystroglycan (DAG), another central component of DGC. Indeed, the sarcoglycan complex

is known to safeguard the integrity of DAG by blocking the proteolytic cleavage of the transmembrane β-DAG subunit into smaller, non-functional fragments[37]. Remarkably, our western blot analysis revealed a reduced level of the β-DAG[43] normal fragment (43 kDa) in three mutants compared to WT fibroblasts (Fig. 5g). More interestingly, we observed a strong 31 kDa band on the western blot, consistent with the

**Fig. 5 | SPN1 deficiency leads to dysregulation of ECM components. a** Volcano plot showing the alternative splicing analysis comparing two WT (WT1 and WT2) and three mutants (F1$^{m1/m1}$, F2$^{m2/m3}$, and F4$^{m6/m6}$) fibroblasts. The -Log10 (FDR) (False Discovery Rate) is plotted on the y-axis, whereas the x-axis represents the exon inclusion level (ΔPSI) where PSI = Percent Splicing Inclusion. ΔPSI is the difference between PSI patient average compared to WTs. Likelihood ratio test was used **b** Sashimi plot showing significant *SGCA* exon 3 disruption in three mutant lines. **c** Bubble plot illustrating gene ontology (GO) enrichment analyses of down-regulated genes in *SNUPN* mutant fibroblasts compared to WTs. Circle size indicates the number of differentially expressed genes enriched in each pathway. Circle color represents the enrichment significance with red showing high statistical significance (Wald test). **d** Volcano plot displaying genes significantly upregulated (green) and downregulated (red) in three mutant fibroblasts compared to two WTs (Wald test). **e** RT-qPCR analysis on four dysregulated genes identified by RNA-seq.

Fold change relative to 3 WT (WTs) is plotted as mean ± SEM *n* = 3 biological replicates. *LAMA5* and *SGCA*: ****$P < 0.0001$; *SFRP2*: *$P = 0.01$; **$P = 0.005$ (F2); **$P = 0.003$ (F4-II:2, F4-II:3 and F10); *ITGB2*: *$P = 0.012$ (F1); *$P = 0.016$ (F2); **$P = 0.008$ (F4-II:2); ***$P = 0.0002$ (F4-II:3); **$P = 0.009$ (F10) with two-tailed unpaired *t*-test. **f** Immunofluorescence staining showing the amount of endogenous COL IV (green) and SGCA (red) in F1$^{m1/m1}$ and F2$^{m2/m3}$ muscle sections compared to WTs. DAPI labeled nuclei (blue). *n* = 2 independent stainings. Scale bar, 50 μm. **g** Immunoblot analysis of β-DAG (β-dystroglycan) using whole-cell lysates from fibroblasts (Lanes 1–4) and HeLa SPN1 mutant cell lines (Lanes 5–6). In fibroblasts, β-DAG$^{43}$ is observed only in the WT sample, while its cleaved fragment (β-DAG$^{31}$) is present in F1$^{m1/m1}$ and F2$^{m2/m3}$. β-DAG is completely absent in F4$^{m6/m6}$. In HeLa cells, β-DAG$^{43}$ is decreased in both mutants compared to WT, whereas the β-DAG$^{31}$ is exclusively found in SPN1$^{sgEx2}$. *n* = 2 independent experiments for each cell type. Source data are provided as a Source Data file.

non-functional β-DAG$^{31}$ fragment, only in F1$^{m1/m1}$ and F2$^{m2/m3}$ mutants (Fig. 5g, lanes 2 and 3). Neither the 43 kDa nor the 31 kDa β-DAG band was detected in F2-II:2$^{m6/m6}$ mutant fibroblast (Fig. 5g, lane 4). This result was corroborated in the whole-cell lysate of HeLa where the levels of cleaved fragments of β-DAG (β-DAG$^{31}$) were found to be increased in SPN1$^{sgEx2}$ compared to WT, while the levels of the β-DAG$^{43}$ were reduced in both SPN1$^{sgEx2}$ and SPN1$^{sgEx9}$ cell lines (Fig. 5g). Notably, histopathological reports of muscle biopsies from three patients (F2-II:1, F10-II:1, and F12-II:2) documented a reduction of DAG1, the extracellular α-DAG subunit which further substantiated our in vitro data (Supplementary Notes).

Overall, our data suggest that the alteration of SPN1 results in mis-splicing and dysregulation of RNA, highlighting the crucial role of *SNUPN* in regulating the spliceosomal machinery and protein synthesis. Using primary fibroblasts from *SNUPN* mutants, we were able to identify impairments in several key components of the DGC and basal lamina, including sarcoglycan, laminin, collagen, and integrins, which have previously been linked to muscular dystrophy disorders[14]. Thus, the change in the molecular signature observed in the *SNUPN*-deficient primary fibroblasts are relevant and meaningful in the context of altered muscular homeostasis in patients.

## Cytoskeleton integrity is compromised in SPN1 mutant fibroblasts and muscle

The DGC transmembrane complex serves as a bridge between the ECM and the cytoskeleton, playing a key role in regulating numerous cellular functions, including cytoskeletal organization and dynamics. Given that SPN1 mutant cells exhibited dysregulation in several components of the sarcolemma and basal lamina, we next examined the cytoskeletal network in patient fibroblasts and muscle tissues.

First, to compare the distribution of various cytoskeletal components between mutant and WT cells, we conducted immunofluorescence staining of F-ACTIN, β-TUBULIN (a subunit of microtubules), VIMENTIN (an intermediate filament type III), and VINCULIN (a scaffolding protein that governs cell–matrix adhesions). In all the mutant cells, F-ACTIN and β-TUBULIN network appeared significantly disorganized and even aggregated into few cytoplasmic areas compared to WT (Fig. 6a and Supplementary Fig. 6a). Additionally, we observed a marked increase of VIMENTIN and VINCULIN stainings in all the mutants compared to WT fibroblasts (Fig. 6b, c). In summary, these data strongly indicated major disruptions of the cytoskeletal structure in SPN1-deficient patient cells.

Next, we turned back to available patients' muscle sections to examine several cytoskeletal proteins by immunostaining. Remarkably, we observed abnormal intra-cytoplasmic accumulation of DESMIN, a primary intermediate filament central to the maintenance of muscle cytoarchitecture, in numerous muscle fibers from patients F1$^{m1/m1}$, F2$^{m2/m3}$, and F12$^{m7/m7}$ (Fig. 6d, e and Supplementary Notes). Moreover, focal aggregation of F-ACTIN was observed in several fibers from F2$^{m2/m3}$ patient muscle, confirming the defect seen in primary fibroblasts from

the same individual (Fig. 6e). Finally, we noted an aberrant accumulation of αβ-CRYSTALLIN, a small heat shock protein linked to DESMIN and F-ACTIN, in F2$^{m2/m3}$ patient cytoskeleton (Fig. 6f). As the functions of these three components are interlocked, any alteration in one of them would presumably result in severe disorganization to the cytoskeleton, followed by an impairment of the ability to sense ECM stiffness[38].

Healthy skeletal muscle fibers are surrounded by the endomysium, also known as basal lamina, which is composed of ECM proteins such as type IV collagen and laminin. Sarcoglycan, dystroglycan, and dystrophin are major components of the DGC complex forming the sarcolemma which links the cytoskeleton to the ECM. αB-CRYSTALLIN, DESMIN, and cytoplasmic ACTIN participate in the formation of intermediate filaments and the organization of the cytoskeleton (Fig. 6g, left). Taken together, our results support a model whereby an oligomerization defect in SPN1 mutants would impair (1) the nuclear import of snRNP complex, resulting in (2) mis-splicing and decreased mRNA levels, ultimately leading to protein dysregulation in key components of the basal lamina and finally resulting in (3) cytoskeleton aggregation and ECM alteration which may in part explain the pathogenesis of this disease (Fig. 6g, right).

## Discussion

This study presents hypomorphic biallelic *SNUPN* variants as a cause of muscular dystrophy. Our findings are based on genetic, clinical, and muscular histological analyses of 18 young children affected by this condition. Additionally, we establish a causal relationship between the loss of functional SPN1 and aberrations in mRNA splicing, ECM-cytoskeleton regulation, and muscle homeostasis using primary cell lines and muscle tissues from patients, as well as CRISPR/Cas9-mediated mutant cell lines.

All patients presented with patterns typical of muscular dystrophies. Extreme cases were characterized by generalized muscle weakness leading to a loss of independent ambulation and severe respiratory insufficiency, which eventually led to early death. The spectrum of severity, from mild to severe, followed the deleterious effect of *SNUPN* variants and their functional consequences. Patients with homozygous missense pathogenic variants in C-terminal were the least affected, whereas those with premature stop codon and C-terminus truncation displayed the most overt muscle wasting. Furthermore, neurodegeneration and cataracts were diagnosed more frequently in patients with protein-truncating variants, which is consistent with other muscular dystrophies associated with mRNA mis-splicing defects, such as DM1 and FSHD[16,39,40]. Our cohort of 18 children suggests a trend toward genotype-phenotype correlation, which is expected to become more pronounced with the addition of future cases. Notably, the sole missense mutation in the N-terminal region led to comparable clinical and histopathological muscular phenotypes, along with extramuscular features. Despite its localization within the IBB domain, our experimental data demonstrated that the ultimate pathomechanisms, encompassing the disruption of snRNP biogenesis

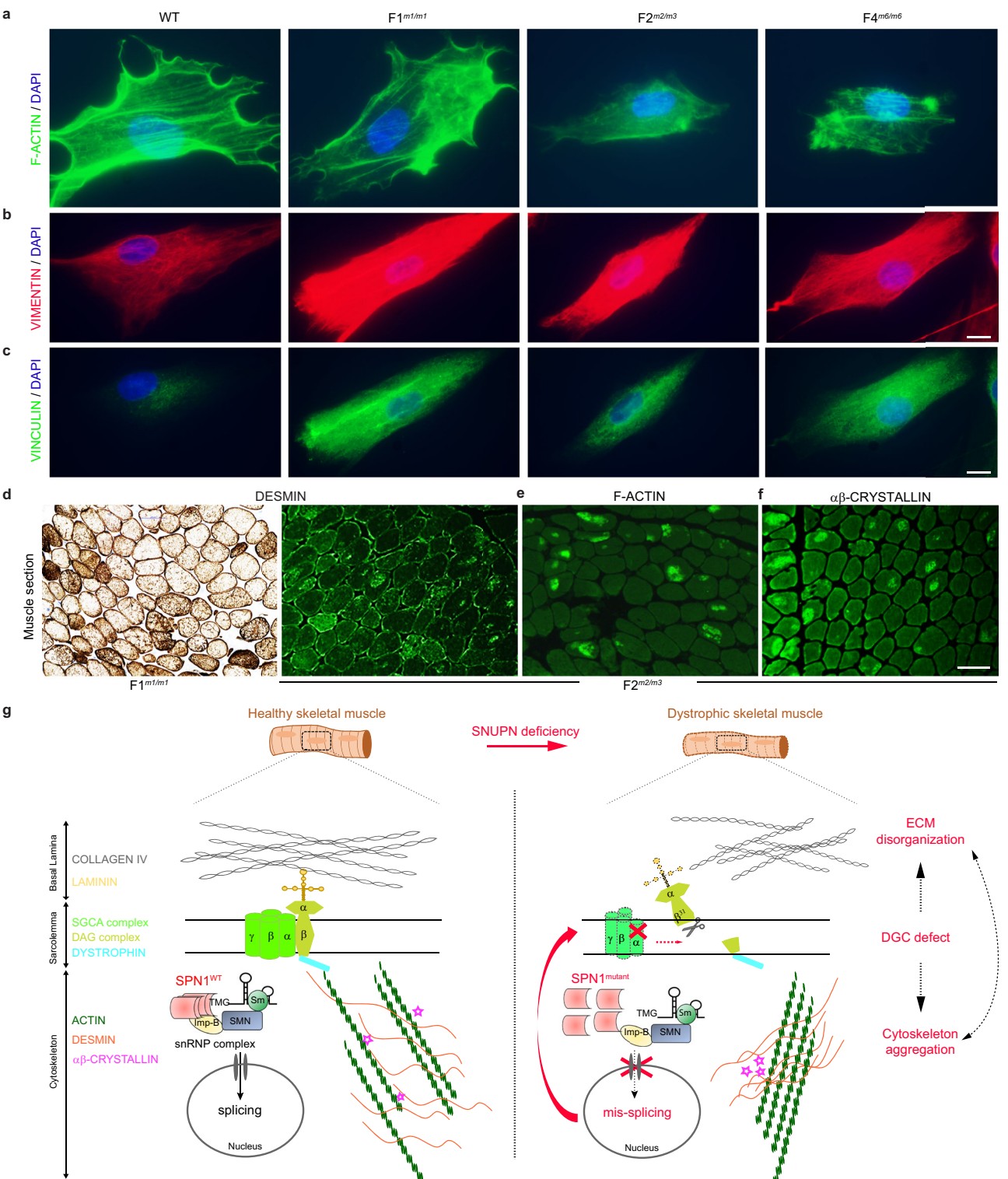

**Fig. 6 | SPN1 deficiency leads to dysregulation of ECM-associated key components. a−c** Representative images of immunofluorescence stainings from WT and patients' fibroblasts (F1$^{m1/m1}$, F2$^{m2/m3}$, and F4$^{m6/m6}$) using cytoskeleton markers. $n = 3$ independent stainings for each marker. Scale bars, 20 μm. **a** Filamentous actin (F-ACTIN) labeled with Phalloidin (green). **b** Anti-VIMENTIN (red). **c** Anti-VINCULIN (green). In the three mutants, filamentous actin (F-ACTIN) networks appeared disorganized and aggregated whereas VIMENTIN and VINCULIN stainings were more intense compared to WT cells. Nuclei are labeled with DAPI (blue). **d** Representative images of immunostaining on muscle sections from patients F1$^{m1/m1}$ and F2$^{m2/m3}$ showing aggregation of DESMIN in myofibers. $n = 1$. Scale bar, 100 μm. **e**, **f** Representative images of immunofluorescence on F2$^{m2/m3}$ muscle section

showing abnormal accumulation of F-ACTIN and αβ-CRYSTALLIN in myofibers. $n = 1$ staining. Scale bar, 100 μm. **g** Proposed schematic model of SPN1 role on ECM and cytoskeleton dynamics in muscle fibers. In healthy skeletal muscle, WT SPN1 homo-oligomers are expected to attach properly to the snRNP import complex and subsequently lead to normal splicing. In dystrophic skeletal muscle carrying hypomorphic *SNUPN* pathogenic variants, the conformational change and the inability of SPN1 mutants to self-oligomerize disorganize the snRNP import complex resulting in (1) aberrant splicing, (2) disassembly of DGC (SGCA & DAG) complexes, and finally (3) cytoskeleton aggregation (F-ACTIN, DESMIN and αβ-CRYSTALLIN) and ECM disorganization (COLLAGEN and LAMININ).

and dysregulation of key components associated with the ECM, are consistent across all *SNUPN* mutants. However, for more comprehensive investigations and a deeper understanding of the role of this region, it will be essential in the future to identify additional patients carrying recessive *SNUPN* N-terminal mutations.

Remarkably, the clinical and functional features of this unusual autosomal recessive muscular dystrophy bear a resemblance to those of LGMD type 2 such as primary dystroglycanopathy (LGMD2P, MIM:613818) or sarcoglycanopathy (LGMD2D/R3, MIM608099) which manifests with muscle weakness, respiratory abnormalities, and rare cardiomyopathy. Furthermore, muscle tissues from three *SNUPN* patients (F2$^{m2/m3}$, F10$^{m9/m9}$, and F12$^{m7/m7}$) exhibited a reduction of dystroglycan, a key component of DGC. These observations were corroborated by our functional studies on patients' fibroblasts and muscle tissues, which revealed an impairment of several components of the DGC and basal lamina, particularly those linked to the sarcoglycan and dystroglycan complexes. Consequently, the identification of *SNUPN* as a gene causing muscular dystrophy adds to the complexity of recessive inherited muscular dystrophies and delineates an additional functional subtype potentially associated with dystroglycanopathy or other LGMD types.

Of the nine variants, eight were found in the last coding exon, indicating that this DNA segment is a mutational hotspot. This region encompasses the last residues (254–300) of the TMG binding domain, and an additional 20 residues (300–320) located in a region whose structure and function are yet to be characterized. Remarkably, our study provides robust evidence that the C-terminus region contains an alpha helix necessary and sufficient for SPN1 oligomerization. The SPN1$^{Tyr301Cysfs*29}$ (*m6*) and SPN1$^{Asp300Valfs*30}$ (*m7*) mutants, which had the C-termini disrupted at the end of the TMG binding site, showed the strongest oligomerization deficiencies and were identified in patients with the most severe phenotypes. Conversely, SPN1$^{Ile309Ser}$ (*m1*), which exhibits the least disrupted SPN1 self-interaction, was found in patients with milder phenotypes. Strikingly, a recent study reported a similar link between the severity of spinal muscular atrophy (SMA) and the ability of mutated SMN to oligomerize[41].

Muscle tissues are highly susceptible to mis-splicing, which can cause pathological changes that contribute to several types of MDs[16]. To our knowledge, only a limited number of MDs have been linked to mutated spliceosome components or splicing regulatory proteins such as DM1 (MIM160900) or LGMD1 (MIM607137)[42,43]. Here, we report another type of recessive MDs and demonstrates a direct association between deficiencies in SPN1, spliceosome maturation, and aberrant splicing, highlighting mRNA mis-splicing as a pathophysiological process underlying this disease.

Beyond providing valuable insights into the pathogenesis of this singular type of MD, our data on patient's fibroblasts also hint at a potential role of myofibroblasts in the early onset of fibrosis observed in our cohort. Fibrosis impacts all types of muscles, including the diaphragm, whose progressive weakness leading to respiratory insufficiency is a major contributor to mortality in muscular dystrophies[44]. This phenomenon could explain our patients' decline and ultimately their death due to respiratory insufficiency. While fibrosis is typically considered an end-stage consequence of abnormal dystrophic muscle repair, it is also suspected to be a driver of LAMA2-related MD[45]. Therefore, gaining a more comprehensive understanding of the molecular mechanisms underlying the early stages of this process would aid in developing effective anti-fibrotic therapies applicable to a wide range of MDs.

In light of the numerous neurological abnormalities observed in the patients who underwent MRI examination, it is advisable to perform early and comprehensive assessment of the central nervous system in all individuals with biallelic *SNUPN* disease-causing variants. These findings will also provide new opportunities to investigate the as-yet uncharacterized role of SPN1 in neurological processes.

In conclusion, our study has uncovered shared underlying pathomechanisms between SPN1-deficient patients and other types of muscular dystrophies. These findings suggest that targeting SPN1 or other snRNP partners may hold promise as a therapeutic approach to correct snRNP assembly. This could potentially pave the way for the development of effective treatments for individuals with MDs.

## Methods

### Ethical approval
Written informed consent was obtained from the individuals/their guardians for the use of their clinical, genetic information and available biological samples, for derivation of primary dermal fibroblasts and for publication of the data and patients' images (Fig. 1 and Supplementary Fig. 1). Patients and their guardians have seen and read the material to be published and have provided informed consent for its publication. The data used in this project are in accordance with local ethical review boards in Turkey, Italy, Switzerland, Iraq, Iran, Macedonia, Colombia, Roumania, and Guatemala. The study protocol was also approved by Koç University Hospital Institutional Review Board (Koç University 2015.120.IRB2.047). All the experiments with human samples were in accordance with the principles set out in the WMA Declaration of Helsinki and the Department of Health and Human Services Belmont Report.

### Patient recruitment
The index case was initially diagnosed with muscular dystrophy from a non-consanguineous Kosovar family (F1) by Z.P.O., Ş.A., and H.K. at Koç University Hospital (Turkey). 17 additional muscular dystrophy patients from 14 families were recruited from different countries: (1) F2-II:1 diagnosed by L.V. at Unit of Clinical Pediatrics, State Hospital, San Marino Republic (Italy), consulted by L.B. and G.Z at Bambino Gesù Children's Hospital, IRCCS (Italy) and by A.P. at IRCCS Institute of Neurological Sciences of Bologna (Italy); (2) F3-II:4 diagnosed by A.M.C. at Division of Neurology, Nationwide Children's Hospital (USA) and by C.G.B. and S.D. at National Institute of Neurological Disorders and Stroke, National Institute of Health (USA); (3) F4-II:2 diagnosed and evaluated by K.S., A.R., and A.B. at Institute of Medical Genetics, University of Zurich (Switzerland), by M.L.G. at Division of Sleep Medicine, University Children's Hospital Zurich (Switzerland) and by G.M.S. at Neuromuscular Center and Department of Pediatric Neurology, University Children's Hospital Zurich (Switzerland); (4) F5-II:4 diagnosed by M.F.R. at Department of Neonatology, Children's and Youth Hospital Auf der Bult (Germany), by A.K.B. and S.v.H. at Department of Human Genetics, Hannover Medical School (Germany) and by Ev.B. at Institute of Diagnostic and Interventional Neuroradiology, Hannover Medical School (Germany); (5) F6-II:3 and F7-II:1 evaluated by Z.F. at Genetics Research Center, USWR (Iran) and by A.K. at Kariminejad-Najmabadi Pathology & Genetics Center (Iran); (6) F8-II:2, F9-II:1, F13-II:1, and F14-II:1 reported from CENTOGENE GmbH by M.E.R., S.K., An.P., and A.B.A. F8-II:2 evaluated by Ş.A., H.K., and Z.P.O. at Koç University (Turkey); F9-II:1 clinical evaluation performed by R.I.T. and D.A.E. at "Dr Victor Gomoiu" Children's Hospital (Romania) and by An.R. and I.S. at University of Medicine and Pharmacy of Craiova (Romania); F13-II:1 evaluated by N.G.R. at Corporación Nuevos Rumbos (Colombia); F14-II:1 diagnosed by C.B.M. at the Pediatric Department, Hospital Pablo Tobon Uribe (Colombia) and by D.L.C.R. Center of Immunology and Genetics (CIGE), SURA Ayudas Diagnosticas (Colombia); (7) F10-II:1 diagnosed by G.B., L.P., J.M.G., S.Z., S.A.B., N.A.B., and M.T. at Dr. John T. Macdonald Foundation Department of Human Genetics, School of Medicine, University of Miami (USA); (8) F11-II:1 diagnosed by K.V. at the Department of Pediatric Neurology, Ludwig-Maximilians-University (Germany) and T.B. at Institute of Human Genetics, Technical University Munich School of Medicine (Germany); (9) F12-II:2 diagnosed by M.B. at Institute of Human Genetics, School of Medicine, Technical University of Munich

(Germany) and by U.K. at Division of Child Neurology and Inherited Metabolic Diseases, Center for Pediatric and Adolescent Medicine, University Hospital Heidelberg (Germany); and (10) F15-II:1 diagnosed by I.S.S. at the Department of Neurology, University Medical Center Hamburg-Eppendorf (Germany) and D.G. at Genetikum (Germany).

## Whole exome sequencing (WES) and segregation analysis

Whole exome sequencing (WES) was performed at different research institutes according to their standard protocols. Briefly, 1 µg of high-quality genomic DNA from affected individuals (F1-F13) was captured by the Ion TargetSeq Exome and Custom Enrichment Kit (Life Technologies). Then, sequencing was performed using Ion PI™ chip (Life Technologies) in the Ion Proton™ Instrument (Life Technologies) followed by exome library preparation via Ion OneTouch™ System (Life Technologies). The Torrent Mapping Alignment Program (TMAP) from the Torrent Suite was then used to align the sequence reads to the human reference genome (GRCh37/hg19 Assembly). The variant calling was performed by the Torrent Variant Caller plugin (v.5.0.2). Finally, the annotated variants with their genes, genomic position, coverage, and quality score were analyzed by the Clinical Reporter (Fabric Genomics). Following variant annotation, they were checked and filtered to retain those variants with low allele frequency (<0.5%) in public genomic databases including Genome Aggregation Database (https://gnomad.broadinstitute.org/), the Exome Sequencing Project (http://evs.gs.washington.edu/EVS/), and the BRAVO/TOPmed database (https://bravo.sph.umich.edu/freeze8/hg38/). Besides, the potential pathogenicity of the variants was predicted using online tools PolyPhen-2[46], SIFT[47], MutationTaster[48], M-CAP[49], FATHMM-MKL[50], Mutation assessor[51], CADD[52], and DANN[53]. Finally, some of the candidate *SNUPN* variants were validated by segregation analysis using targeted Sanger sequencing with the BigDye Terminator Cycle Sequencing Kit (Applied Biosystems, catalog # 4337455). Primer sequences are shown in Supplementary Table 2.

## Runs of homozygosity (ROH) analysis

AutoMap algorithm (v1.2) was used for the detection of Runs of homozygosity (ROH) regions[54]. The ROH intervals were obtained using the patients' VCF files, the algorithm's default options, and the algorithm's common option to compute the common ROHs of multiple individuals. The default options are as follows; sliding window size: 0.75; minimal number of variants in the detected ROH: 25; minimal percentage of homozygous variants in the detected ROH: 88.

## Cell culture

Primary dermal fibroblasts were derived from skin biopsies of five affected (F1-II:1, F3-II:4, F4-II:2, F4-II:3, and F10-II:1) and three unrelated healthy individuals (WT) following standard procedures[55]. All human cell lines including primary fibroblasts, HeLa cells (ECACC catalog # 93021013, RRID:CVCL_0030), and HEK293T cells (ATCC, catalog # CRL-3216, RRID:CVCL_0063) were cultured in high glucose DMEM (Sigma, Catalog # D6429) supplemented with 10% fetal bovine serum (FBS) (Biowest, catalog # S1600-500) and 1% penicillin/streptomycin (Biowest, catalog # L0022-100) and they were maintained in a humidified 5% $CO_2$ atmosphere at 37 °C. All the cell lines were also tested negative for mycoplasma contamination using MycoAlert Mycoplasma Detection Kit (Lonza, catalog # LT07-118).

## Antibodies

Antibodies used in this work include: a rabbit anti-SPN1 polyclonal antibody (catalog #15358-1, used at 1:1000 dilution for the western blot analyses, 1/50 for immunofluorescence labeling of the cells and 1/100 for immunofluorescence labeling of the muscle tissue sections, Proteintech), a mouse anti-SNRPB monoclonal antibody (catalog # MA5-13449, clone # Y12, used at 1:100 dilution for the western blot analyses, 1/50 for immunofluorescence labeling, Invitrogen), a mouse anti-COILIN

monoclonal antibody (catalog # ab87913, clone # IH10, used at 1:500 dilution for the western blot analyses, 1/100 for immunofluorescence labeling, Abcam), a rabbit anti-SMN polyclonal antibody (catalog #11708-1-AP, used at 1/25 for immunofluorescence labeling, Proteintech), a mouse anti-α-SARCOGLYCAN monoclonal antibody (catalog # NCL-L-a-SARC, clone # AD1/20A6, used at 1/50 for immunofluorescence labeling, Leica), a mouse anti-β-DYSTROGLYCAN monoclonal antibody (catalog # NCL-b-DG, clone # 43DAG1/8D5, used at 1/1000 for the western blot analyses, Leica), a rabbit anti-β-TUBULIN monoclonal antibody (catalog # T5201, clone # TUB2.1, used at 1/100 for immunofluorescence labeling, Sigma), a rabbit anti-VIMENTIN monoclonal antibody (catalog # ab92547, clone # EPR3776, used at 1/100 for immunofluorescence labeling, Abcam), a mouse anti-VINCULIN monoclonal antibody (catalog # ab130007, clone # VIN-54, used at 1/100 for immunofluorescence labeling, Abcam), a rabbit anti-COLLAGEN IV polyclonal antibody (catalog # ab6586, used at 1:2500 dilution for the western blot analyses, 1/100 for immunofluorescence labeling of the cells and 1/400 for immunofluorescence labeling of the muscle tissue sections, Abcam), a rabbit anti-FIBRILLARIN polyclonal antibody (catalog # ab5821, used at 1/200 for immunofluorescence labeling, Abcam), a rabbit anti-H2A polyclonal antibody (catalog # 2595, used at 1:1000 dilution for the western blot analyses, Cell Signaling), a mouse anti-GAPDH monoclonal antibody (catalog # 47724, clone # 0411, used at 1:500 dilution for the western blot analyses, Santa Cruz), a mouse anti-MYC monoclonal antibody (catalog # 05-724, clone # 4A6, used at 1:1000 dilution for the western blot analyses, Sigma), a rabbit anti-FLAG polyclonal antibody (catalog # 2368, used at 1:1000 dilution for the western blot analyses, Cell Signaling), a mouse anti-DESMIN monoclonal antibody (catalog # Mab3430, used at 2.5 µg/ml dilution for the immunohistochemistry and immunofluorescence labeling, clone # DE-B-5, Millipore), a mouse anti-ACTIN monoclonal antibody (catalog # M0635, clone # HHF35, used at 1/50 dilution for the immunofluorescence labeling, Dako), a mouse anti-αβ-CRYSTALLIN monoclonal antibody (catalog # Ncl-abcrys-512, clone # G2JF, used at 1/100 dilution for the immunofluorescence labeling, Leica)

## Construct generation and transient transfection

To generate expression constructs for transient transfections, the whole ORF and the cDNA fragments containing 1–253 bp and 254–360 bp of the human *SNUPN* were amplified by iProof High-Fidelity PCR kit (Bio-Rad, catalog # 1725330) using complement primer sequences with additional BamHI and XhoI restriction enzyme sites. For Flag-tagged and Myc-tagged constructs, the tag sequences were inserted at the N-terminal after the start codon and at the C-terminal before the stop codon. Finally, the amplicons were cloned into the pCS2+ mammalian expression vector using standard restriction-digestion cloning. The *SNUPN* wild-type construct was used as a template to generate mutant constructs (*m1*, *m6*, *m7*, and *m9*) by Quick-Change II XL Site-Directed Mutagenesis kit (Agilent, catalog # 2005c22). All primer sequences used for cloning and site-directed mutagenesis are shown in Supplementary Table 1. The wild-type and mutant *SNUPN* constructs were transfected into HeLa and HEK293T cells using Opti-MEM medium (Gibco, catalog # 31985062) and lipofectamine 2000 reagent (Invitrogen, catalog # 11668019), according to the manufacturer's protocol.

## Generation of CRISPR/Cas9-mediated SPN1 mutant cell lines

*SNUPN* mutants were edited by CRISPR/Cas9 technology in HeLA cells. 2 different oligonucleotide sequences were designed as follows: 5′-CACCG(N)20-3′ and 5′AAAC(N)20C-3′ for BsmBI restriction sites. "(N) 20" corresponds to single-guide RNA (sgRNA) sequences which are shown in Supplementary Table 2. SgRNA#1 and sgRNA#2 target exon 2 and exon 9 of *SNUPN*, respectively. The sgRNA oligonucleotides were cloned into the lentiCRISPRv2 vector and validated by Sanger sequencing using the BigDye® Terminator v3.1 Cycle Sequencing Kit (Applied Biosystems, catalog # 4337455). Then, the lentiviruses were

produced by cotransfection of lentiCRISPRv2, pCMV-VSV-G, and psPAX2 plasmids into the HEK293T cells (ATCC, catalog # CRL-3216, RRID:CVCL_0063). Lentiviruses were collected for 3 consecutive days and concentrated using PEG Virus Precipitation Kit (Sigma, catalog # MAK343). The HeLa cells (ECACC catalog # 93021013, RRID:CVCL_0030) were infected with various lentivirus suspension volumes in complete media containing 5 μg/ml polybrene. Subsequently, the infected cells were selected under 0.5 μg/ml puromycin treatment, and the clones were obtained by using serial dilution from the polyclonal populations. Finally, various subclones were sequenced and two of them were utilized in this study. The first line named SPN1$^{sgEx2}$ harbors two types of mutations (c.85_89delinsACT and c.88C>A) at the beginning of the gene (Exon 2), while the second named SPN1$^{sgEx9}$ carries three types of mutations (c.896_921delCA-GACTATGCTGGGCACCAGCTCCAG; c.899_903delinsCTCCA, and c.903_915delTGCTGGGCACCAG) in the final exon (Exon 9).

## Immunofluorescence staining

Cells at the same passage were seeded on Millicell EZ 8-well chamber slides (Millipore, catalog # PEZGS0896). Next day, the semi-confluent cells were washed with PBS and fixed for 20 min in 4% paraformaldehyde at room temperature. Next, cells were permeabilized with 0.5% Triton-X100/PBS for 30 min and blocked in 1% BSA/PBS for 1 h at room temperature. Subsequently, the samples were incubated with the primary antibodies in 1% BSA/PBS overnight at 4 °C. After washing the samples with 1% BSA/PBS, they were incubated in 1:500 respective secondary antibodies conjugated to Alexa Fluor 488 (Invitrogen, catalog # A32766) or Alexa Fluor 594 (Invitrogen, catalog # A32754) for 2 h at room temperature in the dark. Nuclei staining was performed using 10 μg/ml DAPI for 15 min and the cells were mounted with Pro-Long™ Diamond Antifade Mounting (Invitrogen, catalog # P36965). For the staining of muscle tissue, the cryosections were first air-dried and then fixed in 4% paraformaldehyde at room temperature for 20 min. Subsequently, the cryosections were washed with PBS and blocked with 1% BSA/PBST at room temperature for one hour. They were then incubated with primary antibodies overnight at 4 °C. On the following day, the slides were washed with PBS and incubated with respective secondary antibodies conjugated to Alexa Fluor 488 (Invitrogen, catalog # A32766) and Alexa Fluor 594 (Invitrogen, catalog # A32754) at a dilution of 1:500. Additionally, 10 μg/ml of DAPI was included during this incubation, and the entire process was carried out at room temperature in the dark for one hour. Finally, the slides were washed with PBS and mounted with ProLong™ Diamond Antifade Mounting (Invitrogen, catalog # P36965). Fluorescence and confocal images were captured by Leica MD4 B Upright Fluorescent and Leica DMi8 SP8 Inverted Confocal Microscopes, respectively. The obtained results were confirmed using three independent experiments.

## Immunohistochemistry staining

Human muscle cryosections were processed with hematoxylin and eosin staining or immunohistochemical analysis. Slides were incubated overnight with primary antibodies, followed by incubation with secondary biotinylated polyclonal antibodies (Dako) for 1 h at room temperature. Subsequently, the sections were stained with streptavidin-biotin-peroxidase complex system (Dako) according to manufacturer's instructions. Peroxidase activity was detected with DAB substrate. All sections were counterstained with hematoxylin.

## Immunoblot

Total protein from the cells was extracted by using cold 1× RIPA lysis buffer (Millipore, catalog # 20-188) supplemented with cOmplete™, EDTA-free Protease Inhibitor Cocktail (Roche, catalog # 11873580001). Protein quantification was carried out using the Pierce™ BCA Protein Assay Kit (Thermo Fisher Scientific, catalog # 23225). For immunoblotting, the samples were reduced by Laemmli loading buffer (Bio-Rad,

catalog # 161-0747) containing dithiothreitol (DTT) and denatured for 5 min at 95 °C. Then, the Precision Plus Protein Dual Color Standard (Bio-Rad, catalog # 1610374) and equal amounts of protein samples were loaded on 4–20% Criterion™ TGX™ Precast Midi Protein Gels (Bio-Rad catalog # 5671093) and run at 80 V for 2 h. Subsequently, proteins were transferred on a Immun-Blot® Low Fluorescence PVDF Membrane (Bio-Rad, catalog # 1620261) using Trans-Blot® Turbo™ Transfer System (Bio-Rad, catalog # 1704150). Membranes were blocked with 5% milk/TBST for 1 h at room temperature and then probed with the primary antibodies diluted in 5% milk/TBST overnight at 4 °C. After 3 washes in TBST, membranes were incubated with the respective HRP conjugated anti-mouse (Abbkine, catalog # A21010) and anti-rabbit (Abbkine, catalog # A21020) secondary antibodies in 5% milk/TBST. After 3× washes in TBST, the signal was revealed by the Pierce™ ECL Western Blotting Substrate (Thermo Fisher Scientific, catalog # 32106) and the SuperSignal West Chemiluminescent Substrates (Thermo Fisher Scientific, catalog # 34580/34096/A38554). At the end, membranes were developed by ChemiDoc MP Imaging System (Bio-Rad, catalog # 17001395).

## Immunoprecipitation

An aliquot of 20 μL of slurry Anti-FLAG M2 beads (Sigma-Aldrich, catalog # A2220) was collected by centrifugation for 1.5 min at 3300 × $g$ at 4 °C after one wash with PBS and 3 washes with RIPA. They were then resuspended in the cell lysate supernatant and incubated on a rotator at 4 °C overnight. Next day, the beads were washed 6 times with RIPA and eluted by heating in 50 μL of 2× Laemmli loading buffer with 60 mM DTT for 10 min at 95 °C. Finally, the beads were removed by centrifugation and the supernatant was analyzed by immunoblotting. For visualization, HRP conjugated anti-mouse IgG light chain specific secondary antibody (Abbkine, catalog # A25012) and HRP conjugated anti-rabbit IgG light chain specific secondary antibody (Abbkine, catalog # A25022) were used.

## Subcellular fractionation

Nuclear and cytoplasmic fractions of fibroblast and HeLa cells were separated using Cell Fractionation Kit-Standard (Abcam, catalog # ab109719) according to the manufacturer's instructions. Then, the fractions were subjected to immunoblotting. GAPDH (cytoplasmic marker) and H2A (nuclear marker) were used to assess the purity of each fraction and to normalize the level of proteins.

## RNA extraction and RT-qPCR

Total RNA was isolated from the cells using NucleoSpin RNA kit (MACHEREY-NAGEL, Catalog # 740955.50) following the manufacturer's protocol, including rDNase (RNase-free) treatment. 1 μg of RNA was reverse transcribed using Iscript™ cDNA Synthesis Kit (Bio-Rad, catalog # 1708890). RT-qPCR reactions were performed using FastStart™ Universal SYBR® Green Master (Rox) (Roche, catalog # 4913914001) and gene-specific primers (Supplementary Table 1) on PikoReal 96 Real-Time PCR System (Thermo Fisher Scientific, catalog # TCR0096). GAPDH was used as the housekeeping gene to normalize gene expression levels.

## RNA sequencing analysis

Sequencing of the RNA samples was performed by Phi-Bioinformatics (Istanbul, Turkey). Illumina BaseSpace (Illumina Inc., San Diego, CA) was used for bioinformatics analysis of the raw fastq files. Briefly, raw fasta files were aligned to genome version hg19 via the STAR aligner[56] using paired-end reads option, with default settings. Obtained bam files were then employed as inputs for DESeq2[57] to obtain counts data and to find differentially expressed genes. Splicing analyses were performed with rMATS 4.1.2 turbo software, using the mentioned bam files as input[58] and enabling the "variable read length option". Sashimi plots were visualized via "rmats2sashimiplot" package. IDEP.951[59], a web-based tool for RNA-Seq analyses, was used for further visualization and statistical analyses.

## Structure modeling

The 3D structure of the protein was generated using I-TASSER (Iterative Threading ASSEmbly Refinement) web server[60–62]. The amino acid sequences of the proteins correlated with homozygous pathogenic variants were submitted to the I-TASSER server, which uses a combination of threading-based alignments and Ab Initio modeling to generate a 3D model of the protein. The top-ranked models were selected for further analysis and validation. The 3D structures of the proteins generated by I-TASSER were visualized using PyMOL software, version 2.4.1 (Schrödinger LLC). The PyMOL program was used to view and analyze the proteins' structures focusing mainly on the m3G cap-binding site, N-terminus, and C-terminus. The models were assessed by examining key structural features such as bond lengths, angles, and steric clashes using PyMOL tools. All the models were aligned with two crystallographic models (PDB 1XK5[22] and PDB 3GJX[63]) to have an accurate prediction of the models.

To assess the likelihood of multimerization between C-termini of SPN1, the protein sequence was provided as input to the AlphaFold 2 Colab notebook[64,65] following the procedures outlined by DeepMind, to generate protein structure predictions. Multiple structural models were generated for each multimer, and the highest-confidence model, determined by the lowest predicted aligned error (PAE) score, was selected for downstream analyses. Furthermore, the predicted structures generated by AlphaFold 2 Colab were subjected to further analysis using the Socket2 program[66]. This analysis focused on assessing the potential formation of coiled-coil interactions involving the C-termini with packing-cutoff at 7 Å.

## Quantification and statistical analysis

All statistical tests were carried out using Prism 9 or Excel unless otherwise stated. Information on statistical tests used for each assay and number of samples are detailed in the figure legends and in the "Methods" sections. The values are presented as mean ± SEM. $P$-value < 0.05 was considered statistically significant. All of the experiments were done in at least three biological replicates. All the samples were included, and analyses were performed without blinding.

## Web resources

The URLs for data presented herein are as follows: BLAST, https://www.blast.ncbi.nlm.nih.gov/Blast.cgi/ dbSNP146, http://www.ncbi.nlm.nih.gov/sn p/ 1000 Genomes, http://www.1000genomes.org/ Online Mendelian Inheritance in Man (OMIM), http://www.omim.org/ ANNOVAR, http://www.openbioinformatics.org/annovar/ PolyPhen-2, http://genetics.bwh.harvard.edu/pph2/ ClinVar, ftp://ftp.ncbi.nlm.nih.gov/pub/clinvar/vcf_GRCh37/ Exome Sequencing Project, https://evs.gs.washington.edu/EVS/ The Human Protein Atlas https://www.proteinatlas.org/

## Reporting summary

Further information on research design is available in the Nature Portfolio Reporting Summary linked to this article.

## Data availability

The RNA sequencing datasets generated during the current study are available in the Gene Expression Omnibus repository, accession number GSE232712, available online. Patient-related data, including genetic sequencing data not included in the manuscript or its supplements, were generated as part of clinical care and may be subject to patient confidentiality. All requests for raw and analyzed data and materials related to patients presented in this article will be reviewed by the respective institution to verify if the request is subject to any intellectual property or confidentiality obligations. Data requests for anonymized data are typically shared with qualified investigators after a material transfer agreement; such requests should be directed to corresponding author nbeillard@ku.edu.tr. Due to the nature of coordinating data access across multiple institutions and countries along with their specific regulations, the timeframe for responding to requests may vary. They are no specific restrictions on data-use via data-use agreements. All other data generated in this study are available within this article and its Supplementary Information and Source Data files. Source data are provided with this paper.

## Code availability

No custom code was used for data analysis. All software and packages used are listed in the "Methods" section.

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

## Acknowledgements

We are grateful to all the individuals and their families who participated in this research. Special thanks to Prof. Devrim Gozuacik, and Dr. Yunus Akkoç for their invaluable advice and generous sharing of material and consumables. We are grateful to Prof. Tugba Bagci-Onder and his research group for their kind assistance in generating CRISPR/Cas9 mutant cell lines and sharing material. We express our gratitude to Dr. Madhuri Hegde, the team at the Center for Mendelian Genomics, Broad Institute of MIT and Harvard, and CureCMD for their help. We extend our thanks to Prof. Dek Woolfson and Dr. Rokas Petrenas, University of Bristol, for their assistance in running and providing Socket2 analysis. We are grateful to all members of Department of Medical Genetics, Koç University School of Medicine (KUSoM) for their support and constructive feedback. The authors gratefully acknowledge the use of the services and facilities of the Koç University Research Center for Translational Medicine (KUTTAM), funded by the Presidency of Turkey, Head of Strategy and Budget. N.E.B. is funded by a 2232 International Fellowship for Outstanding Researchers Program of Scientific and Technological Research Council of Turkey (TÜBİTAK) (Project No: 118C318). Work in C.G.B. is supported by intramural funds from the NIH National Institute of Neurological Disorders and Stroke. G.Z. is a member of the E.B.European Reference Network for Rare Neurological Diseases. A.R. received funds from the University of Zurich Research Priority Program ITINERARE. M.T. and S.Z. are funded by NIH Common Fund, through the Office of Strategic Coordination/Office of the NIH Director under award number 1U01HG010230.

## Author contributions

N.E.B. directed the project, designed, analyzed, and processed all the experiments. N.E.B., M.N. and E.B. coordinated the clinical part of the study. N.N., H.P.S., M.N. and B.S. with the help of S.E.U and Z.B.E. performed and analyzed all the biochemical, immunofluorescence, culture experiments, and quantification. M.N. generated the predicted structural SPN1 mutant analyses. M.N. and E.B. performed the splicing and expression transcriptomics analyses. N.N. and E.Y. generated the CRISPR-cas9 mutant cell line. Em.B. supervised cloning design and statistical analysis. H.P.S, C.K.K., M.L.V., and L.V.S. performed immunostaining on human muscle sections. Z.P.O., Ş.A., H.K., L.V., A.P., G.Z., L.B., A.M.C., C.G.B., S.D., K.S., A.R., A.B.A, M.E.R, S.K, M. L. G., G.M.S., M.F.R., A.K.B., Ev.B., S.v.H., Z.F., A.K., R.I.T., An.R., I.S., D.A.E., G.B., L.P., J.M.G., S.Z., S.A.B., M.T., K.V., T.B., M.B., U.K., N.G.R., C.B.M., D.L.C.R., I.S.S., A.B., M.L.G. and D.G. conducted the clinical and genetic diagnosis of the patients, the collection of human biological samples and solo/trio exome sequencing on the affected and their parents. N.E.B., N.N. and M.N. wrote the manuscript with the input of all the co-authors.

## Competing interests

M.E.R., S.K., A.P. and A.B.A. are employed by and receive a salary from Centogene AG. The remaining authors declare no competing interests.

## Additional information

Marwan Nashabat[1,42], Nasrinsadat Nabavizadeh [1,42], Hilal Pırıl Saraçoğlu [1,42], Burak Sarıbaş [1], Şahin Avcı[2], Esra Börklü [2], Emmanuel Beillard [3], Elanur Yılmaz [1], Seyide Ecesu Uygur[1], Cavit Kerem Kayhan [4,5], Luca Bosco[6,7], Zeynep Bengi Eren[1], Katharina Steindl[8], Manuela Friederike Richter [9], Guney Bademci[10], Anita Rauch [8,11,12], Zohreh Fattahi[13,14], Maria Lucia Valentino[15,16], Anne M. Connolly[17], Angela Bahr[8], Laura Viola [18], Anke Katharina Bergmann [19], Maria Eugenia Rocha[20], LeShon Peart [10], Derly Liseth Castro-Rojas[21], Eva Bültmann[22], Suliman Khan[20], Miriam Liliana Giarrana[23], Raluca Ioana Teleanu[24,25], Joanna Michelle Gonzalez[10], Antonella Pini[26], Ines Sophie Schädlich[27], Katharina Vill[28,29], Melanie Brugger [29], Stephan Zuchner[10,30], Andreia Pinto[20], Sandra Donkervoort[31], Stephanie Ann Bivona[10], Anca Riza[32,33], Undiagnosed Diseases Network*, Ioana Streata[32,33], Dieter Gläser[34], Carolina Baquero-Montoya[35], Natalia Garcia-Restrepo[36], Urania Kotzaeridou[37], Theresa Brunet [28,29], Diana Anamaria Epure[24], Aida Bertoli-Avella [20], Ariana Kariminejad[14], Mustafa Tekin [10,30], Sandra von Hardenberg[19], Carsten G. Bönnemann [31], Georg M. Stettner [38], Ginevra Zanni [6], Hülya Kayserili [2,39], Zehra Piraye Oflazer [40] & Nathalie Escande-Beillard [1,41] ✉

[1]Laboratory of Functional Genomics, Department of Medical Genetics, Koç University, School of Medicine (KUSoM), Istanbul, Turkey. [2]Diagnostic Center for Genetic Diseases, Department of Medical Genetics, Koç University Hospital, Istanbul, Turkey. [3]Department of Biopathology, Centre Léon Bérard, Lyon, France. [4]Pathology Laboratory, Acıbadem Maslak Hospital, Istanbul, Turkey. [5]Department of Biotechnology, Nişantaşı University, Istanbul, Turkey. [6]Unit of Muscular

and Neurodegenerative Disorders and Developmental Neurology, Bambino Gesù Children's Hospital, IRCCS, Rome, Italy. [7]Department of Science, University "Roma Tre", Rome, Italy. [8]Institute of Medical Genetics, University of Zurich, Schlieren-Zurich, Switzerland. [9]Department of Neonatology, Children's and Youth Hospital Auf der Bult, Hannover, Germany. [10]Dr. John T. Macdonald Foundation Department of Human Genetics, University of Miami Miller School of Medicine, Miami, FL, USA. [11]Research Priority Program (URPP) ITINERARE: Innovative Therapies in Rare Diseases, University of Zurich, Zurich, Switzerland. [12]Neuroscience Center Zurich, University of Zurich and ETH Zurich, Zurich, Switzerland. [13]Genetics Research Center, University of Social Welfare and Rehabilitation Sciences, Tehran, Iran. [14]Kariminejad-Najmabadi Pathology & Genetics Centre, Tehran, Iran. [15]IRCCS Institute of Neurological Sciences of Bologna, Bologna, Italy. [16]Department of Biomedical and Neuromotor Sciences, University of Bologna, Bologna, Italy. [17]Division of Neurology, Nationwide Children's Hospital, The Ohio State University College of Medicine, Columbus, OH, USA. [18]Unit of Clinical Pediatrics, State Hospital, San Marino Republic, Italy. [19]Department of Human Genetics, Hannover Medical School, Hannover, Germany. [20]CENTOGENE GmbH, Rostock, Germany. [21]Genomics Laboratory, Center of Immunology and Genetics (CIGE), SURA Ayudas Diagnosticas, Medellín, Colombia. [22]Institute of Diagnostic and Interventional Neuroradiology, Hannover Medical School, Hannover, Germany. [23]Division of Sleep Medicine, University Children's Hospital Zurich, Zurich, Switzerland. [24]Dr Victor Gomoiu Children's Hospital, Bucharest, Romania. [25]Carol Davila University of Medicine and Pharmacy, Bucharest, Romania. [26]Neuromuscular Pediatric Unit, IRCCS Institute of Neurological Sciences of Bologna, Bologna, Italy. [27]Department of Neurology, University Medical Center Hamburg-Eppendorf, Hamburg-Eppendorf, Germany. [28]Department of Pediatric Neurology and Developmental Medicine and LMU Center for Children with Medical Complexity, Dr. von Hauner Children's Hospital, LMU Hospital, Ludwig-Maximilians-University, Munich, Germany. [29]Department of Human Genetics, Technical University of Munich, School of Medicine, Munich, Germany. [30]John P. Hussmann Institute for Human Genomics, University of Miami Miller School of Medicine, Miami, FL, USA. [31]Neuromuscular and Neurogenetic Disorders of Childhood Section, National Institute of Neurological Disorders and Stroke, National Institutes of Health, Bethesda, MD, USA. [32]Human Genomics Laboratory, University of Medicine and Pharmacy, Craiova, Romania. [33]Regional Centre of Medical Genetics Dolj, County Clinical Emergency Hospital, Craiova, Romania. [34]Genetikum, Neu-Ulm, Germany. [35]Pediatric department, Hospital Pablo Tobon Uribe, SURA Ayudas Diagnosticas, Medellín, Colombia. [36]Universidad de Manizales, Manizales, Caldas, Colombia. [37]Division of Child Neurology and Inherited Metabolic Diseases, Center for Pediatric and Adolescent Medicine, University Hospital Heidelberg, Heidelberg, Germany. [38]Neuromuscular Center Zurich and Department of Pediatric Neurology, University Children's Hospital Zurich, University of Zurich, Zurich, Switzerland. [39]Department of Medical Genetics, Koç University School of Medicine (KUSoM), Istanbul, Turkey. [40]Department of Neurology, Koç University Hospital Muscle Center, Istanbul, Turkey. [41]Research Center for Translational Medicine (KUTTAM), Koç University School of Medicine (KUSoM), Istanbul, Turkey. [42]These authors contributed equally: Marwan Nashabat, Nasrinsadat Nabavizadeh, Hilal Pırıl Saraçoğlu. *A list of authors and their affiliations appears at the end of the paper. ✉e-mail: nbeillard@ku.edu.tr

## Undiagnosed Diseases Network

Guney Bademci[10], Stephanie Ann Bivona[10], Stephan Zuchner[30] & Mustafa Tekin[30]

