## [Peer Review File · Nature Communications]

SNUPN deficiency causes a recessive muscular dystrophy due to RNA mis-splicing and ECM dysregulationReviewer #1 (Remarks to the Author):

Review of Nashabat et al. Nat Comms ms. # NCOMMS-23-18913-T

In this manuscript, the authors identify a cohort of patients presenting with a novel form of muscular dystrophy (MD), with recessive mutations in the human SNUPN gene that encodes the ribonucleoprotein import factor Snurportin-1 (SPN1). SPN1 recognizes the mature 5'-trimethylguanosine cap on the spliceosomal small nuclear RNAs and functions as an adaptor to bring these essential snRNPs into the nucleus following assembly of the Sm core RNP. Thus null mutations in SNUPN are expected to be inviable. The authors have identified what appear to be nine new alleles of SNUPN that cause a significant MD phenotype and carry out a nice set of follow up experiments to get at the underlying mechanism.

I feel that after suitable revisions have been completed, this work would represent an important contribution and should be published. As it stands now, there are improvements needed (mainly in Figs 3 and 4) most of which involve interpretation or dry-lab experimentation. I hope that the other referees are better able to judge the experiments presented in Figs 5 and 6, as I did not have enough time to delve deeply into those data to offer any substantive criticism.

Respectfully signed, Greg Matera

Major comments:

Overall, the manuscript is well written but several key points need to be addressed.

1. First of all, the Abstract states that: "Additionally, the nuclei exhibited defective spliceosomal maturation, decreased SMN levels, and breakdown of Cajal bodies." Claims made in the Abstract should be supported by the main figures of the paper.

(a) The data showing SMN levels go down in the mutant cells are not in the main figures. Where are these data and why are they not in the main?

(b) Do the authors suggest that "defective spliceosomal maturation" is shown by the data presented in Fig 4c? The antibody for detecting SNRNPB is not described. Is this mAb Y12? If so, that antibody recognizes three different Sm proteins, most prominently SmB. But I digress. I don't think that the authors can make this claim with the current dataset, although I don't doubt that there is likely a problem in snRNP maturation.

(c) Line 293 mentions that the product of the SNRNPB gene (SmB) undergoes nucleocytoplasmic shuttling. Loose terminology here. SmB is NOT known to shuttle, per se, but it DOES undergo nuclear import (nucleocytoplasmic transport not same as shuttling). Anyway, the anti-Sm staining in Fig 4b looks good. Can the fluorescent intensity in the nucleoplasm of these WT vs mutants be quantified and statistically compared across multiple nuclei?

(d) Figure 4 is trying to do too many things at once and it does none of them particularly well. It has been demonstrated multiple times in the literature that mammalian fibroblasts do not display Cajal bodies (CBs). The frequency per cell is very low in that cell type. So the claim that coilin foci (CBs) are disrupted in the mutant cells is not substantiated by Fig 4d. HOWEVER, the coilin phenotype displayed in 4d is almost certainly a relocalization of coilin to the nucleolus. This is good because RNAi targeting SPN1 was shown to cause exactly this phenotype (first shown by Shpargel 2005, PMID 16301532, c.f. Fig 3C). This is good evidence that the patient mutation is causing a disruption in snRNP biogenesis and defects in this process cause coilin to relocalize. Perhaps this phenotype could be quantified as well? If so, those two new bits of information (critique points 1c and 1d) would greatly bolster the idea

that SPN1 mutations are causing disruptions to snRNP biogenesis. Co-staining with anti-Fibrillarin or other nucleolar marker would be helpful.

(e) I don't think the data in Fig 4e are informative because the WT fibroblast cells do not display coilin foci in the first place (the small dots they show in panel 4d are not CBs). So they cannot be 'broken' if they are not there to begin with. In addition to scoring the nucleolar relocation of coilin, a potential solution to the foci problem would be to use HeLa cells. In Fig 3b, the authors transfect HeLa cells with several Myc-tagged SPN1 constructs. Why not use them for immunofluorescence with anti-Myc and anti-coilin? Do the mutants with the frameshifted C-termini behave differently in the presence or absence of leptomycin B (LMB)? See Ospina 2005 (PMID: 16030253) for details. Anyway, even if the myc-tagged mutants don't behave differently in an LMB assay, the data in panel 4e should be removed and re-done as described in critique 1d.

2. The idea that SPN1 homodimerizes (or multimerizes) is intriguing and fairly well supported by the data in Fig 2. But I think the authors missed an opportunity here with their molecular modeling. Out of curiosity, I just pulled up two different AlphaFold2 structures of SPN1 from H.sapiens and P.troglodytes. And those models predict a beautiful alpha-helix located around aa position 295-330. Notably, this is precisely where the two frameshift mutations occur, as well as the Gln308 truncation. I think the molecular modeling should be redone to incorporate the idea that a coiled-coil (Leu zipper or other type of interaction) interaction might be important for SPN1 dimerization.

Mutation of key residues within that presumptive coiled coil (e.g. Ile309Ser) could disrupt potential intramolecular interactions. My quick perusal of the AlphaFold structure suggests that there may be contacts between Ile309 and a region near the N-terminal (aa 25-30) IBB domain. There could be some sort of intra-molecular interaction between the N- and C-terminal regions of SPN1 that is affected by dimerization (as the authors mentioned was predicted by data in Ospina, 2005, PMID 16030253).

3. The data in Fig 2b strongly suggest that the C-terminal region of SPN1 is necessary for homodimer formation. What does the subcellular distribution of these mutant proteins look like? As mentioned above in point 1e, the prediction would be that these frameshift (and also the C-term truncation) alleles would be unable to form heterodimers with the WT protein. This can be explicitly tested by western blotting with Flag-WT and myc-mutant for example.

Moreover, one could easily test for sufficiency. That is, the C-term region seems to be necessary for oligomer formation. Is it also sufficient? One could easily make GST-fusions or some other type of construct containing only the C-termini of the WT and one or more of the mutants and see if they multimerize. I'm betting that the putative C-term (alpha helical?) domain is sufficient.

Minor points

A. The order of figure panels 2e and 2d should be flipped. Show the data first and then the quantification of those data.

B. Line 220, first use. Not a fan of the term "mutain." Consider removing this term throughout the text.

C. Line 233, I don't think that Ile309 is "deeply buried" in either the I-TASSER structure shown in Fig 2a or in the AlphaFold structures I referred to above. Again, aa 309 might tuck up near the IBB domain. A phylogenetic comparison of the putative helix region (295-330) could be analyzed with a helical wheel diagram. A charged residue on the wrong side of the helix could disrupt dimer formation. We used a helical wheel when thinking about oligomerization of the SMN C-terminal region recently,

and it was really helpful (See Gupta 2021, PMID: 34181727).

D. Line 271, label the figures clearly as to which gels were run in the presence of reducing agent (DTT) and which were not. It might be better to show a gel like the one in the supplement where same gel is +/- reducing agent to better illustrate this important point in the main figures.

Reviewer #2 (Remarks to the Author):

Major comments:

Although the genetic and clinical evidence supporting the pathogenicity of most variants are pretty convincing, I am not sure about p.Arg55Gln because it was found in a small nuclear family with only in-silico data and unlike other variants located in N-terminus presenting with a phenotype a bit different from others accompanied with microcephaly and ID, so you should either provide more data to support its causality and if you can not then discuss it clearly.

Have you checked the effect of SNUPN(ENST00000308588.10):c.164G>A(p.Arg55Gln) on splicing as it is within the splicing region and it might affect splicing.

It would be good to functionally test another non-pathogenic variant in C-terminus of the protein like SNUPN(ENST00000308588.10):c.971T>A(p.Leu324His), as it has high AF so it is most likely not pathogenic, and use it as a control to compare with your missense variant p.Ile309Ser.

Do you have muscle MRI from the patients? It would be good to collect and review the images to see if any pattern would emerge and include them in the figure.

Minor comments:

Remove Exac database as it doesn't exist anymore. You should check big databases such as UKBB, gnomAD v2&3, TOPMed and Centogene internal database.

For consistency, change mutation to pathogenic variants throughout the manuscript.

Line 12, What is "mode of inheritance analysis"?

Line 16, To date, pathogenic variants in more than 60 genes have been associated with

Line 149, died due to respiratory failure before the age of 15, years or months?

Line 152, disease-causing variants in SNUPN, to potentially disease-causing variants.

Line 166, What is "severe frameshift" variants? How can a frameshift be severe or not severe?

Line 460, change "severe variants" to protein-truncating variants

would be good to do haplotype analysis for 3 recurrent variants to see if they are founder or recurrent variants?

It would be nicer to put families with same variants next to each other.

Please add a statement about number of males/females and age range of the patients in the cohort in the table, intellectual disability for a boy who is only 3 years old? ID should be used for children above 5 years old.

I don't think you need to show the conservation across species for LOF variants.

Family 6: the patient is only 3 years old so how it was determined that he has intellectual disability?

You should change to developmental delay or motor delay.

What is the level of intellectual disability in family 10 and is microcephaly congenital or progressive and how severe it is, z score?

For families with very uncommon features such as intellectual disabilities or hearing loss it would be good to check the sequencing data which could possibly explain these features as they might have blended phenotype.

Reviewer #3 (Remarks to the Author):

The authors identified biallelic mutations in SNUPN in 18 cases from 15 families affected by complex form of muscular dystrophy with early onset and severe progression, and associated with extra muscular involvement, including central nervous system involvement.

Most mutations were stopgain, frameshift, splicing although missense variants were also identified, and nearly all clustered at the c-terminus of the gene

Authors performed a series of functional studies on three patients derived fibroblast lines and one muscle biopsy. Key results of the functional studies were:

- SNUPN expression was unchanged at mRNA and protein level, although a truncated isoform was detected in a patients with a splicing variant. In fibroblasts SNUPN protein seems increased in the perinuclear region.
- CO-IP experiments of in vitro supported an effect of two mutations located in the C-terminus on SNUPN protein oligomerization.
- Altered snRNP complex formation
- Altered splicing and gene expression, including genes involved in the extracellular matrix
- Alteration of cytoskeleton and extracellular matrix

The genetic data are convincing. However the evidence of the disease causing mechanisms is limited by the low number of samples included.

At least three control lines should be used, particularly when high-throughput experiments like RNAseq. Most patients underwent a muscle biopsy but only 1 muscle was used. qPCR quantification seems highly variable on fig 2-d. Western blotting was not quantified in main figure but by eye it seems band in patients is less intense then in control Notably, figure 4f there seems to be a reduced staining for SNUPN (SPN1) in the patient biopsy compared to control.

Gene expression (qPCR), western blot for SNUPN quantification and IHC should be repeated on more samples and possibly on affected tissue.

If the stopgain/frameshift variants do not lead to nonsense mediated decay one would expect to observe a truncated SNUPN protein, while these were not observed in two patients with C term truncating variants. Full gels should be provided and if possible more cases and controls included. It seems the anti SNUPN/SPN1 antibody used correctly binds to the N terminus, worth mentioning it

Also retention of truncated isoforms escaping NMD in proteins which acts as part of dimers/oligomers/complexes as SNUPN would be expected to lead to a dominant-negative effect, while the disease is recessive and parents unaffected. This should be considered, discussed and probably reflected in CO-IP experiments (wt/mut co transfection should be carried out, not only wt/wt, mut/mut). It may also be useful to consider transfecting a shorted SNUPN isoform lacking the C terminus to further support the role of C terminus on oligomerization

Why only 2 fibroblasts were used in figure 3d?

AT least 3 controls should be used in RNAseq experiments before drawing more conclusions from differently expressed genes and exons. The clustering of patients and controls should be showed as PCA plots. Do authors see in SNUPN transcripts the presence of variants escaping NMD?

Additional comments:

Clinical data could be more clearly presented.

Case II-1 from family 10 carrying biallelic missense variant has early onset and severe phenotype, which contrast the genotype-phenotype correlation

The spelling of single mutation including nt and AA change throughout the text instead of m.1 ..m2 etc would make the reading easier.

Reviewer #4 (Remarks to the Author):

In this manuscript, the authors investigated the link between snurportin-1 (SPN1) deficiency and muscular dystrophy. Using primary fibroblasts and muscle tissues from patients, their major findings are: 1) in the large majority of patients, mutations in the *spn1* gene are mapped to the C-terminal domain, 2) mutations in the C-termini of SPN1 mutants prevent the formation of oligomeric complexes, 3) SPN1 mutants alter the assembly of the snRNP complex and impair its transport to the nucleus, 4) disruption of the maturation process of the spliceosome in SPN1 fibroblasts mutants, 5) SPN1 deficiency alters key components of the extracellular matrix, 6) the structural integrity of the cytoskeletal component is compromised in cells deficient in SPN1. Based on these observations, the authors conclude that SPN1 is critical in maintaining the structural integrity of the extracellular matrix, DGC, and cytoskeleton network and that mutations in the c-termini of SPN1 are associated with muscular dystrophy.

The overall manuscript is well-written, and the results appear interesting. However, there are several issues with this manuscript. The main concerns of this reader are that the results are overinterpreted and that there are several discrepancies between the results and the drawn conclusions.

1. The data showing the accumulation of SPN1 (Fig 2 f) around the nuclei of the patients' fibroblasts are very sketchy. The authors should provide the entire field of wild-type and mutant cultured fibroblasts (not one cell) with quantifications. Additionally, the claim that Fig. 2g supports the perinuclear accumulation of SPN1 mutant observed in mutant fibroblasts is weak as the SPN1 staining is uniformly distributed over the entire sarcolemma and upregulated in the sarcoplasm. The authors should provide high-resolution images to make their case. Also, the authors should explain why the expression levels of the SPN1 mutant have increased in muscle cells but not in fibroblasts when compared to the wild type and the significance of such results. Finally, the expression levels of SPN1 mutants in fibroblasts may not give an idea about the pathogenicity of SNUPN variants. This should be done on muscle biopsies.

2. The authors claimed in the abstract of this manuscript that mutant SPN1 failed to homodimerize, and yet in Figure 3 d, they showed in mutant fibroblasts the existence of a higher band around 100 kDa, which presumably due to the dimerization of SPN1 monomers. The authors should clarify this issue.

3. The crux of this paper is that mutations in the c-termini of SPN1 alter the assembly of the snRNP complex and impair its transport to the nucleus. This reader has an issue with the following conflict results: In Figure 2 f and g the authors claimed that in F1m1/m1 mutant fibroblasts and muscles, SPN1 accumulates around the nucleus (failed to translocate into the nucleus). However, in Figure 4C, they provided data showing that SPN1 is highly localized in the nuclear fraction. The authors should explain this discrepancy and provide immunostaining data of SNRPB and coilin/SMN in F1m1/m1 mutant fibroblasts.

4. In Figure 2 g, the authors provided immunohistochemistry images of muscle sections showing a more intense level of SPN1 in F1 m1/m1 mutant subsarcolemmal. Figure 4 f provided images showing the complete absence/diffuse of SPN1 staining in F1 m1/m1 mutant subsarcolemmal. These conflicting results are concerning.

5. Concerning the dysregulation of ECM components and DGC complex in SPN1 mutants, the authors should label muscle sections with antibodies against DGC complex and ECM proteins to determine which molecules are affected by the SPN1 mutations.

Point-by-point Response to Reviewers' Comments

Reviewer #1:

In this manuscript, the authors identify a cohort of patients presenting with a novel form of muscular dystrophy (MD), with recessive mutations in the human SNUPN gene that encodes the ribonucleo protein import factor Snurportin-1 (SPN1). SPN1 recognizes the mature 5'-trimethylguanosine cap on the spliceosomal small nuclear RNAs and functions as an adaptor to bring these essential snRNPs into the nucleus following assembly of the Sm core RNP. Thus null mutations in SNUPN are expected to be inviable. The authors have identified what appear to be nine new alleles of SNUPN that cause a significant MD phenotype and carry out a nice set of follow up experiments to get at the underlying mechanism.

I feel that after suitable revisions have been completed, this work would represent an important contribution and should be published. As it stands now, there are improvements needed (mainly in Fig.3 and 4) most of which involve interpretation or dry-lab experimentation. I hope that the other referees are better able to judge the experiments presented in Figs 5 and 6, as I did not have enough time to delve deeply into those data to offer any substantive criticism.

Respectfully signed, Greg Matera

Dear Greg,

Thank you for your expert, positive and fruitful review.

Major comments:

Overall, the manuscript is well written but several key points need to be addressed.

1. First of all, the Abstract states that: "Additionally, the nuclei exhibited defective spliceosomal maturation, decreased SMN levels, and breakdown of Cajal bodies." Claims made in the Abstract should be supported by the main figures of the paper.

(a) The data showing SMN levels go down in the mutant cells are not in the main figures. Where are these data and why are they not in the main?

Thank you for bringing this to our attention. You are absolutely correct, it is a mistake to have pointed this preliminary result in the abstract. Initially, we believed that SMN levels were reduced in the nuclei. Despite conducting further investigations in various models, including primary fibroblasts, muscle tissues from patients, and HeLa *SNUPN* mutants, we observed inconsistent results that we are currently unable to comprehensively explain. Regrettably, we do not have sufficient confidence to draw conclusions regarding the relationship between SMN levels and *SNUPN* mutants at this time. Therefore, since we consider this result non-essential for the current manuscript, we have opted to postpone our investigation into SMN for this paper and delve deeper into it for future studies.

(b) Do the authors suggest that “defective spliceosomal maturation” is shown by the data presented in Fig 4c? The antibody for detecting SNRPB is not described. Is this mAb Y12? If so, that antibody recognizes three different Sm proteins, most prominently SmB. But I digress. I don't think that the authors can make this claim with the current dataset, although I don't doubt that there is likely a problem in snRNP maturation.

Thank you for highlighting this important concern. The antibody we used is the SNRPB monoclonal Antibody (Y12) obtained from Invitrogen (#MA5-13449). Reference is listed in the Supplementary Table 3. In our effort to gain a deeper understanding, we contacted the company to obtain the exact immunogenic sequence used for antibody generation and thoroughly investigate potential targets. Regrettably, the company was unable to provide any additional information regarding the epitope as it remains undetermined, and there is no supplementary epitope or immunogen data available.

Next, we conducted an extensive literature review. In some sources, mAb Y12 has been referred to as anti-SmB/B' (Ajiro et al, 2021). Hsieh et al. in 2019 identified the Y12 antibody as anti-Sm and provided evidence of its recognition of both SmB (25 kDa) and SmD3 (15 kDa) through western blot analysis. For immunofluorescence images, they designated it as SmB/D3 (Hsieh et al, 2019), as depicted in **Reviewer Figure 1**.

Reviewer Figure 1.

Western blot and immunofluorescence showing that Y12 antibody (Invitrogen, Cat # MA5-13449) recognizes SmB and SmD3 proteins (Hsieh et al, 2019).

In our western blot analysis using the Y12 antibody (Invitrogen, Cat # MA5-13449), we observed 3 different bands under high-exposure conditions. In addition to SmB (25 kDa) and SmD3 (15 kDa), we also identified a faint third band at approximately 18 kDa (**Reviewer Figure 2**). Furthermore, we acknowledge that these isoforms are not distinguishable in immunofluorescence staining. Indeed, as you correctly pointed out, it is more accurate to label it as an “anti-Sm” antibody. Therefore, we have updated the labeling of this antibody to “Sm” proteins throughout the manuscript.

Reviewer Figure 2. Western blot analysis showing that Y12 antibody (Invitrogen, Cat # MA5-13449) recognizes several Sm proteins on fibroblast cells.

(c) Line 293 mentions that the product of the SNRPB gene (SmB) undergoes nucleocytoplasmic shuttling. Loose terminology here. SmB is NOT known to shuttle, per se, but it DOES undergo nuclear import (nucleocytoplasmic transport not same as shuttling). Anyway, the anti-Sm staining in Fig 4b looks good. Can the fluorescent

intensity in the nucleoplasm of these WT vs mutants be quantified and statistically compared across multiple nuclei?

We have replaced “nucleocytoplasmic shuttling” with “nuclear import” in the manuscript.

The quantification and statistical comparison of anti-Sm staining in the nucleoplasm of both wild-type (WT) and mutants have already been conducted and were presented in Supplementary Fig. 4b. These data have now been incorporated into the main figure (**New Fig. 4c**). We also demonstrate by western blot a significant increase of Sm proteins in the cytoplasm of 5 mutants (**New Fig. 4d**).

In addition, we have also confirmed that Sm decreased in nuclei of *SNUPN* mutant HeLa lines that we generated via CRISPR-cas9 (**New Supplementary Fig. 4a**). The first line named SPN1^{sgEx2} harbors two types of mutations (c.85_89delinsACT and c.88C>A) at the beginning of the gene (Exon 2), while the second named SPN1^{sgEx9} carries three types of mutations (c.896_921delICAGACTATGCTGGGCACCAGCTCCAG; c.899_903delinsCTCCA and c.903_915delITGCTGGGCACCAG) in the final exon (Exon 9), very similar to our patient variants notably *m6* and *m7* (**New Supplementary Fig. 2f-g**).

(d) Figure 4 is trying to do too many things at once and it does none of them particularly well. It has been demonstrated multiple times in the literature that mammalian fibroblasts do not display Cajal bodies (CBs). The frequency per cell is very low in that cell type. So the claim that coilin foci (CBs) are disrupted in the mutant cells is not substantiated by Fig 4d. HOWEVER, the coilin phenotype displayed in 4d is almost certainly a relocalization of coilin to the **nucleolus**. This is good because RNAi targeting SPN1 was shown to cause exactly this phenotype (first shown by Shpargel 2005, PMID 16301532, c.f. Fig3C). This is good evidence that the patient mutation is causing a disruption in snRNP biogenesis and defects in this process cause coilin to relocalize. Perhaps this phenotype could be quantified as well? If so, those two new bits of information (critique points 1c and 1d) would greatly bolster the idea that SPN1 mutations are causing disruptions to snRNP biogenesis. Co-staining with anti-Fibrillarin or other nucleolar markers would be helpful.

Thank you for guiding us to strengthen these data. As you suggested, we performed co-staining of COILIN and FIBRILLARIN, an established marker of nucleolus. Remarkably, we have successfully validated a significant relocalization of COILIN to the nucleolus in all available SPN1 mutants when compared to WT fibroblasts (**New Fig. 4e-f and Supplementary Fig. 4b**).

Furthermore, we have conducted the same experiment using *SNUPN* mutant HeLa lines that we have generated via CRISPR-cas9. Interestingly, the results in both HeLa mutant cell lines corroborate those observed in mutant fibroblasts (**New Supplementary Fig. 4c**).

As you pointed out, these new pieces of data align with the existing literature (Shpargel and Matera 2005) and provide compelling confirmation that defective SPN1 directly contributes to disruptions in snRNP biogenesis.

(e) I don't think the data in Fig 4e are informative because the WT fibroblast cells do not display coilin foci in the first place (the small dots they show in panel 4d are not CBs). So they cannot be 'broken' if they are not there to begin with. In addition to scoring the nucleolar relocalization of coilin, a potential solution to the foci problem would be to use HeLa cells. In Fig 3b, the authors transfect HeLa cells with several Myc-tagged SPN1 constructs. Why not use them for immunofluorescence with anti-Myc and anti-coilin? Do the mutants with the frameshifted C-termini behave differently in the presence or absence of leptomycin B (LMB)? See Ospina 2005 (PMID: 16030253) for details. Anyway, even if the myc-tagged mutants don't behave differently in an LMB assay, the data in panel 4e should be removed and re-done as described in critique 1d.

Thank you for sharing your expertise on Cajal bodies (CBs). While we were aware that fibroblasts might not be the ideal cells for observing CBs based on the literature, we did not have a clear understanding that fibroblasts typically do not form CBs.

However, following your suggestion, we have removed the data related to fibroblasts and instead focused on examining HeLa *SNUPN* mutant cells. In the new panels **Fig. 4g and h**, we clearly demonstrate a significant increase of COILIN foci in two HeLa *SNUPN* mutant cell lines.

Finally, we believe that these data are sufficient to establish that endogenous deficiency of SPN1 indeed leads to CBs breakdown. As a result, we did not perform the LMB assay.

2. The idea that SPN1 homodimerizes (or multimerizes) is intriguing and fairly well supported by the data in Fig 2. But I think the authors missed an opportunity here with their molecular modeling. Out of curiosity, I just pulled up two different Alphafold2 structures of SPN1 from *H.sapiens* and *P.troglodytes*. And those models predict a beautiful **alpha-helix located around aa position 295-330**. Notably, this is precisely where the two frameshift mutations occur, as well as the Gln308 truncation. I think the molecular modeling should be redone to incorporate the idea that a coiled-coil (Leu zipper or other type of interaction) interaction might be important for SPN1 dimerization.

Mutation of key residues within that presumptive coiled coil (e.g. Ile309Ser) could disrupt potential intramolecular interactions. My quick perusal of the Alphafold structure suggests that there may be contacts between Ile309 and a region near the N-terminal (aa 25-30) IBB domain. There could be some sort of intra-molecular interaction between the N- and C-terminal regions of SPN1 that is affected by dimerization (as the authors mentioned was predicted by data in Ospina, 2005, PMID 16030253).

Thank you once again for your valuable input. As suggested, we have generated new SPN1 models using I-TASSER. Indeed in the WT model, we observe a well-defined alpha Helix between Gly303 and Glu322 which appears to be partially or entirely disrupted in all the mutant models (**New Fig. 3a**).

The best Predicted Aligned Error (PAE) score was observed for the Ct-Ct dimer ranging roughly from 10-15, followed by the Ct-Ct tetramer around 15, where two parallel dimers interact with each other in an antiparallel orientation. Furthermore, the possibility of Nt-Nt and Nt-Ct multimerization appears to be valid, especially between amino acids (11-63) and (295-330), although the PAE scores were slightly higher (15-20). For this manuscript, we have chosen to focus on Ct-Ct oligomerization. We will further investigate the role of Nt through prediction modeling and biochemical confirmation in our future work. We also have added coiled-coil predictions of Ct tetramer formation (**New Fig. 3e**).

3. The data in Fig 2b strongly suggest that the C-terminal region of SPN1 is necessary for homodimer formation. What does the subcellular distribution of these mutant proteins look like? As mentioned above in point 1e, the prediction would be that these frameshift (and also the C-term truncation) alleles would be unable to form heterodimers with the WT protein. This can be explicitly tested by western blotting with Flag-WT and myc-mutant for example.

We have followed your suggestion and confirmed by immunoprecipitation that Myc-SPN1 frameshift mutants (*m6* and *m7*) are unable to form heterodimers with Flag-SPN1^{WT} unlike Myc-SPN1 WT or *m1* (**New Supplementary Fig. 3b**). This suggests that an intact C-terminal region is indeed necessary for oligomerization.

Moreover, one could easily test for sufficiency. That is, the C-term region seems to be necessary for oligomer formation. Is it also sufficient? One could easily make GST-fusions or some other type of construct containing only the C-termini of the WT and one or more of the mutants and see if they multimerize. I'm betting that the putative C-term (alpha helical?) domain is sufficient.

Thank you for your suggestion. First, we generated several constructs, both wildtype and mutant, with the C-terminal region spanning all our mutations and tagged with either Myc or Flag (Myc-SPN1²⁵⁴⁻³⁶⁰ and Flag-SPN1²⁵⁴⁻³⁶⁰). As expected, we noted that *m6* and *m7* fragments were highly unstable compared to WT and *m1*. Firstly, we observed that WT SPN1²⁵⁴⁻³⁶⁰ can only interact with WT and *m1* SPN1²⁵⁴⁻³⁶⁰ fragments but not with *m6* and *m7* SPN1²⁵⁴⁻³²⁸ constructs (**New Fig. 3f and Supplementary Fig. 3d**). Secondly, quite interestingly, the binding of *m1* SPN1²⁵⁴⁻³⁶⁰ fragments with itself was significantly less efficient than the binding to the WT C-terminal fragments confirming the likely involvement of the Ile309 residue in oligomerization.

Minor points

- A. The order of figure panels 2e and 2d should be flipped. Show the data first and then the quantification of those data.

It appears there may have been a misunderstanding. Panel 2d did not represent the quantification of panel 2e. Panel 2d showed the quantification of *SNUPN* RNA by qPCR, whereas panel 2e displayed the SPN1 protein level by Western blot.

However, in the revised version, Figure 2 has been modified in response to several requests from other reviewers. The qPCR data are now shown in the **New Supplementary Fig. 2c** and western blot data and their quantification can be found in the **new Fig. 2d and e**.

- B. Line 220, first use. Not a fan of the term “mutein.” Consider removing this term throughout the text.

We have replaced “mutein” with “mutant” throughout the text.

- C. Line 233, I don't think that Ile309 is "deeply buried" in either the I-TASSER structure shown in Fig 2a or in the AlphaFold structures I referred to above. Again, aa 309 might tuck up near the IBB domain. A phylogenetic comparison of the putative helix region (295-330) could be analyzed with a helical wheel diagram. A charged residue on the wrong side of the helix could disrupt dimer formation. We used a helical wheel when thinking about oligomerization of the SMN C-terminal region recently, and it was really helpful (See Gupta 2021, PMID: 34181727).

We followed your advice and generated helical wheel diagrams, both parallel and anti-parallel, for the Ct-Ct regions based on the prediction of coiled-coil oligomeric state (LOGICOIL) (**Reviewer Figure. 3**). Remarkably, this region exhibits a higher probability of forming a tetramer rather than a dimer and spans amino acids Gly303 to Leu339. Moreover, in these predictions Ile309, located in position “a”, plays a central role in the hydrophobic interactions. We have included a model of tetramer formation based on socket-2 coiled-coil prediction with this C-terminal region. The model highlights residues including Ile309 that are predicted to form hydrophobic interactions (**New Fig. 3e and Supplementary Fig. 3e**). Thus, we speculate that the substitution to Ser309, a hydrophilic residue, is likely to contribute to the alteration in the stability of the SPN1 oligomer. As described earlier in reply to comments 3, we have explored further by performing coimmunoprecipitation of C-terminal fragments of WT and mutants and demonstrated that m1-m1 interaction is less efficient than wt-m1 C-terminal fragments (**New Fig.3f, lanes 1 and 4**). As a result, we modified our explanation in the manuscript accordingly.

MARCOIL predicted region: 2
 Sequence: GHQLQQIMEHKKSQKEGMKEKLTHKASENGHYELEHL
 Register: abcdefgabcdefgabcdefgabcdefgabcdefgab
 Result of prediction:
 Most probable state is TETRAMER
 Second most probable state TRIMER

 ANTI PARA TRIM TETRA
 Raw score is 0.94 1.05 1.07 1.36

Reviewer Figure.3: Prediction SPN1 WT and I309S mutant coiled-coil oligomeric state using LOGICOIL. Helical wheels prediction with DrawCoil show charged or polar amino acids in orange. Negatively charged Glutamic acid (E) is in red whereas positively charged lysine (K) is in blue.

D. Line 271, label the figures clearly as to which gels were run in the presence of reducing agent (DTT) and which were not. It might be better to show a gel like the one in the supplement where the same gel is +/- reducing agent to better illustrate this important point in the main figures.

As per your suggestion, we have moved Supplementary Fig. 3a to the main figure (**New Fig. 3c**). Furthermore, we have included information regarding the presence or absence of DTT in all the relevant western blot (**Fig. 3c-d and Supplementary Fig. 3c**).

Reviewer #2 (Remarks to the Author):

Major comments:

Although the genetic and clinical evidence supporting the pathogenicity of most variants are pretty convincing, I am not sure about p.Arg55Gln because it was found in a small nuclear family with only in-silico data and unlike other variants located in N-terminus presenting with a phenotype a bit different from others accompanied with microcephaly and ID, so you should either provide more data to support its causality and if you can not then discuss it clearly.

Thank you for your positive feedback on our study. We agree that pathogenicity of p.Arg55Gln was open to discussion at the point of our initial submission. However, there are now several evidences that support its potential pathogenic role:

- 1- All the predicted tools that we employed (Polyphen, SIFT, DANN, MutationTaster, MCAP, FATHMM-MKL) consistently provided a "possibly damaging" score, as shown in Supplementary Table 1.
- 2- We generated primary fibroblasts from the patient carrying this mutation and conducted various assays. Whereas RNA and protein levels, and oligomerization of this mutant appear unaffected (**New Fig. 2d-e and Supplementary Fig. 2c and 3a**), SPN1 immunofluorescence staining revealed a similar aberrant pattern as observed in other mutant fibroblasts (**Supplementary Fig. 2d**).
- 3- Most significantly, we observed in this patient's fibroblasts a disruption in snRNP biogenesis, primarily illustrated by coilin redistribution to the nucleolus (**New Fig. 4e and Supplementary Fig. 4b**) and variations in *LAMA5*, *SGCA*, *SFR2* and *ITGB2* genes (**New Fig. 5e**) mirroring observations made in other mutant fibroblasts.

We are firmly convinced that this variant is pathogenic, likely operating through additional distinct pathogenic mechanisms due to its particular localization in the IBB domain. Interestingly, despite p.Arg55Gln seemingly unaffected oligomerization, all these mutants share a common defect in snRNP biogenesis. In the future, it will be interesting to delve further into the uniqueness of this variant and investigate as to why the variant displays a peculiar phenotype including microcephaly and intellectual disability (ID). We also have mentioned it in the discussion.

Have you checked the effect of SNUPN(ENST00000308588.10):c.164G>A(p.Arg55Gln) on splicing as it is within the splicing region and it might affect splicing.

We have examined the potential effect of *SNUPN* (ENST00000308588.10):c.164G>A on splicing by RT-PCR. Our finding revealed no discernible changes in the size of *SNUPN* amplicons (exon2-exon9 and exon2-exon4) between WT and patient cells, suggesting no

splicing defect (**Reviewer Figure 4**). Furthermore, we did not observe any change in the size of the endogenous or overexpressed mutant protein (**New Fig. 2d and Supplementary Fig. 3a-b**). To further support our results, we investigated a potential effect of the *SNUPN* c.164G>A variant on the pre-mRNA splicing. Using three different algorithms (NNSPLICE, MaxEntScan and GeneSplicer), the *in silico* analysis did not predict any splicing alteration, possibly because the variant is too far away from the exon boundary (5 nt).

Reviewer Figure 4. a) Schematic representation of *SNUPN* mRNA, *SNUPN* CDS, *m9* mutation (c.164G>A) and RT-PCR primers. **b)** Sequence of RT-PCR forward and reverse primers and their product length. **c)** RT-PCR results show no detectable differences in the size *SNUPN* amplicons (exon2-exon9 and exon2-exon4) in the F10_{*m9/m9*} compared to WTs.

It would be good to functionally test another non-pathogenic variant in C-terminus of the protein like *SNUPN*(ENST00000308588.10):c.971T>A(p.Leu324His), as it has high AF so it is most likely not pathogenic, and use it as a control to compare with your missense variant p.Ile309Ser.

Thank you for your suggestion. However, while we appreciate the idea of testing non-pathogenic variants in the C-terminus, we cannot be certain that selecting a random variant with a high heterozygous allele frequency (AF) would guarantee that the same variant would not be pathogenic in a homozygous state. This is exemplified by our families, where healthy parents carry the heterozygous *SNUPN* variant, while the affected patients carry the same variant in both alleles.

For instance, *SNUPN* (ENST00000308588.10): c.971T>A (p.Leu324His) is indeed found at a high AF in a heterozygous state but has never been reported at the homozygous state. We were unsure whether using this variant or other missense variants with high heterozygous AF would serve as suitable negative controls. Given that this C-terminal region is a hotspot for *SNUPN* pathogenic variants and that we could not identify any missense variant with a high AF in the homozygous state, we chose not to take the risk of generating potentially confusing data. Furthermore, our clinical and functional data for p.Ile309Ser in primary fibroblasts and muscle tissues provide compelling evidence of its pathogenicity.

Do you have muscle MRI from the patients? It would be good to collect and review the images to see if any pattern would emerge and include them in the figure.

We asked all the clinicians if they had conducted muscle MRI. Unfortunately, none of the patients in this cohort had undergone muscle MRI scans previously. Most clinicians expressed reluctance to perform these tests, primarily because patients' conditions are now quite severe. Especially for those who present severe contractures and scoliosis, it would be challenging to position them correctly in the scanner.

However, clinicians offered to perform muscle ultrasounds for patients F12-II:2 and F15-II-1 as the value of this exploration in muscle screening and muscle disease diagnosis has been well-established for more than 20 years (Albayda & van Alfen, 2020; Vill et al, 2020).

These results have been included in **New Supplementary Fig.1d**. Both patients showed dystrophic involvement with homogeneously increased echogenicity of the quadriceps, ankle flexors, anterior tibial, abdominal wall and biceps brachii muscles. Triceps brachii, autochthonous back musculature, deltoid, hamstrings, gracilis and adductor longus were examined only in patient F15 and also showed a dystrophic pattern. Thoracic wall (serratus anterior) and forearm flexors were examined only in the patient from family 15 and also showed severe atrophy and increased muscle echogenicity. Muscular ultrasound of the patient from family 15 revealed relative sparing of masseter and genioglossus which is in line with preserved bulbar function. The exact classifications according to the modified Heckmatt scale are shown in the figures.

Minor comments:

Thank you for your thorough review.

Remove Exac database as it doesn't exist anymore. You should check big databases such as UKBB, gnomAD v2&3, TOPMed and Centogene internal database.

ExAc database has been removed from the manuscript. The frequency of each variant has been included using gnomAD v2&3, TOPMed, Exome variant server and Centogene internal databases (**Supplementary Table 1**).

For consistency, change mutation to pathogenic variants throughout the manuscript.

Even if the term “mutation” is extensively used in the field, we agree that it is a broader term that encompasses all types of genetic change not necessarily causing disease. Therefore, we did replace “mutation” with “pathogenic variants” throughout the manuscript.

Line 12, What is “mode of inheritance analysis”?

This is an error. We removed “analysis”.

Line 16, To date, pathogenic variants in more than 60 genes have been associated with

This sentence has been corrected.

Line 149, died due to respiratory failure before the age of 15, years or months? Text has been updated with “15 years”.

Line 152, disease-causing variants in SNUPN, to potentially disease-causing variants. Your suggestion has been applied.

Line 166, What is “severe frameshift” variants? How can a frameshift be severe or not severe?

Thank you for bringing this to our attention. We have now removed the term “severe”.

Line 460, change “severe variants” to protein-truncating variants

“Severe variants” is now replaced with “protein-truncating variants”.

would be good to do haplotype analysis for 3 recurrent variants to see if they are founder or recurrent variants?

Thank you for your suggestion. We performed Runs of Homozygosity (ROH) analysis for the pathogenic variants *m1*, *m6*, and *m7*, which unveiled founder haplotypes within the families harboring identical *SNUPN* mutations, specifically encompassing *SNUPN*: chr15:75,890,424-75,918,810 (**Supplementary Fig. 1e** and **Reviewer Table 1**).

SNUPN Pathogenic Variant	Family	ROH Genomic Coordinates
m1	F1, F9	Chr15:74,467,855-78,310,306
m6	F13, F14	Chr15: 74,219,582-79,057,949
m7	F6, F7, F12	Chr15: 60,257,987-88,669,382

Reviewer Table 1. Runs of homozygosity (ROH) analysis results for *m1*, *m6* and *m7* pathogenic variants.

It would be nicer to put families with same variants next to each other.

You are correct, and we did consider this option. However, given the presence of patients with heterozygous mutations, it introduced complexity. As a result, we chose to retain the original order in which we collected the families and implemented a numbering and color-coding system for each variant instead.

Please add a statement about number of males/females and age range of the patients in the cohort.

This sentence has been included in the second paragraph of the clinical description: “The patient cohort in this study comprises a total of 10 males and 8 females with onset ages ranging from birth to 10 years old.”

in the table, intellectual disability for a boy who is only 3 years old? ID should be used for children above 5 years old.

We removed ID from the table and changed it to developmental delay.

I don't think you need to show the conservation across species for LOF variants.

As per your suggestion, we highlight the conservation only for the missense and splice residues.

Family 6: the patient is only 3 years old so how it was determined that he has intellectual disability? You should change to developmental delay or motor delay.

Thank you for pointing this out. Clinicians agreed to change to developmental delay.

What is the level of intellectual disability in family 10 and is microcephaly congenital or progressive and how severe it is, z score?

Patient has persistent microcephaly with a head circumference of 49 cm ($z = -3.60$). We have updated the clinical information in the supplementary note (See below).

F10 Clinical Summary (Supplementary note).

The proband is a 13-year-old female with short stature, microcephaly, developmental delay, and progressive neuromuscular disease.

She was born full-term without complications but came to medical attention due to speech and motor developmental delays. She was first noted at age 3 to have severe microcephaly with a head circumference of 42.7 cm ($z=-3.69$), and brain MRI demonstrated a bilateral opercular migrational disorder. Muscle biopsy performed due to her persistent weakness and fatigue was suggestive of dystroglycanopathy. Her CK measurements ranged from 480 to 6601. She also received growth hormone for short stature but did not respond to the treatment.

At age 7, she underwent serial casting for Achilles tendon contractures, which then led to a regression in her mobility. She later exhibited ataxia and intermittent torticollis, as well as worsening scoliosis for which she underwent a posterior lumbar fusion. She was subsequently hospitalized for pneumonia and found to have severe obstructive sleep apnea requiring BiPAP. By age 11, her myopathic weakness and contractures progressed such that she was no longer ambulatory.

The proband has a current weight of 27.8 kg ($z = -3.40$), height of 119 cm ($z = -5.58$), and persistent microcephaly with a head circumference of 49 cm ($z = -3.60$). She has been diagnosed with mild-to-moderate intellectual disability. In addition, she has developed chronic lung disease complicated by recurrent pneumonia and is G-tube dependent secondary to persistent dysphagia.

She is the first liveborn child of a nonconsanguineous Guatemalan couple. Her family history is negative for any similarly affected individual.

For families with very uncommon features such as intellectual disabilities or hearing loss it would be good to check the sequencing data which could possibly explain these features as they might have blended phenotype.

We have carefully reviewed single nucleotide variants, copy number variants and structural variants in our sequencing data. We did not identify any relevant variant that could explain the intellectual disability or hearing loss. However, we cannot rule out entirely the possibility that variants of unknown significance may still contribute to these particular phenotypes.

Reviewer #3 (Remarks to the Author):

The authors identified biallelic mutations in SNUPN in 18 cases from 15 families affected by complex form of muscular dystrophy with early onset and severe progression, and associated with extra muscular involvement, including central nervous system involvement.

Most mutations were stop gain, frameshift, splicing although missense variants were also identified, and nearly all clustered at the c-terminus of the gene

Authors performed a series of functional studies on three patients derived fibroblast lines and one muscle biopsy. Key results of the functional studies were:

- SNUPN expression was unchanged at mRNA and protein level, although a truncated isoform was detected in a patient with a splicing variant. In fibroblasts SNUPN protein seems increased in the perinuclear region.
- CO-IP experiments of in vitro supported an effect of two mutations located in the C-terminus on SNUPN protein oligomerization.
- Altered snRNP complex formation
- Altered splicing and gene expression, including genes involved in the extracellular matrix - Alteration of cytoskeleton and extracellular matrix

The genetic data are convincing. However the evidence of the disease causing mechanisms is limited by the low number of samples included.

Thank you for your positive feedback regarding the genetic data. We trust that you will recognize our dedicated efforts to include as many patients, cell samples and assays as possible in order to improve the robustness of our functional analysis.

At least three control lines should be used, particularly when high-throughput experiments like RNAseq.

You are absolutely correct. In fact, initially, we conducted RNA-seq on three control and three patient samples. However, after clustering the data with Principal Component Analysis (PCA), we unfortunately identified one of our control samples (WT3) as an outlier (**Reviewer Figure 5a**). Consequently, we chose to exclude WT3 sample to strengthen further our analysis. However, this analysis allowed us to identify several major dysregulated targets that are involved in the pathophysiology of this disease.

Reviewer Figure 5. PCA plots for RNA-seq data using 3 controls (**panel a**) and 2 controls (**panel b**).

We then proceeded to validate the dysregulation of few of these genes at both RNA and protein levels using independent samples from various models (**Old Fig. 5e-f**). Initially, we performed qPCR on biological replicates (3 mutant and 3 control primary fibroblasts), with each sample tested in three technical replicates (**Old Fig. 5e**). For these revisions, we were able to incorporate into our analysis two more primary fibroblasts from two new patient samples (F4-II:3 and F10-II:1). Remarkably, we observed the same pattern of dysregulation in the genes and proteins tested within these new samples (**New Fig. 5e**).

However, we acknowledge that relying solely on this RNA-seq analysis for all our conclusions would not have been appropriate. We used it as a foundational step and conducted extensive investigations beyond that. We performed tests on many samples from different models, including primary fibroblasts and muscles from patients, as well as CRISPR-mutant cell lines (**New Fig. 5e-f and Supplementary Fig. 5h-g**). These investigations include not only RNA but also protein levels. We are confident that this comprehensive approach is sufficient to establish the significance and robustness of our analysis.

Most patients underwent a muscle biopsy but only 1 muscle was used.

Initially, we only had access to the muscle biopsy from our first patient, F1^{m1/m1} for performing staining and we were very cautious in its usage due to the limited amount of remaining tissue. We agree that multiple muscle biopsies were conducted in this cohort and we have made extensive efforts to obtain additional tissues or at the very least, images. We could only retrieve a few muscle staining images of patients F2^{m2/m3} (**Old Fig. 1d**). Unfortunately, the majority of these biopsies were collected by various clinicians worldwide a considerable time ago, rendering the samples or images either irretrievable or no longer accessible.

Lately, we successfully obtained a few tissue muscle sections from patients F2_{m2/m3}. Subsequently, in our pursuit of obtaining more consistent results, we repeated all the immunofluorescence staining and imaging processes on F1_{m1/m1}, F2_{m2/m3} and two control samples simultaneously (**New Fig. 2g and Fig. 5f**).

qPCR quantification seems highly variable on fig 2-d.

Thank you for raising your concern. The observed variation in RNA *SNUPN* levels within the same sample may be indicative of the use of primary cell lines, even though we cultured them together under identical conditions and extracted samples from sets taken at the same passage. Furthermore, we have noted variability in SPN1 protein levels across different passages.

To enhance the robustness of our results, we conducted tests on two new patient samples (F4II:3_{m6/m6} and F10_{m9/m9}). We consistently tested three technical replicates and three biological replicates from different passages for each sample. In total we have tested 5 *SNUPN* mutant cells and 3 independent control samples (**New Supplementary Fig. 2c**).

Western blotting was not quantified in main figure but by eye it seems band in patients is less intense than in control. Notably, figure 4f there seems to be a reduced staining for SNUPN (SPN1) in the patient biopsy compared to control.

Thank you for your suggestion. We have performed multiple western blotting and included two new patient samples (F4-I:3 and F10-II:1). SPN1 protein levels were quantified across 4 to 5 blots for each mutant compared to 2 to 3 controls in each run (**New Fig. 2e**). Except for F10_{m9/m9}, quantification revealed a significant decrease in SPN1 protein level in all patients compared to wildtypes (**New Fig. 2e**).

However, over the course of our experiments and passages, we observed substantial variability in the SPN1 protein levels, particularly among the mutants. Although we cannot pinpoint the exact cause, we suspect that stability of SPN1 mutant proteins with C-terminal alterations may be particularly sensitive in solution, despite our stringent measures to prevent degradation. Our observation aligns with prior findings of SPN1 degradation in solution compared to its higher stability when it is in a complex state (Strasser *et al*, 2005).

Gene expression (qPCR), western blot for SNUPN quantification and IHC should be repeated on more samples and possibly on affected tissue.

As already mentioned in replies above, RT-qPCR and western blot were performed multiple times and two new patient fibroblast samples were included (F4II:3^{m6/m6} and F10^{m9/m9}) (**new Fig. 2d-e and Supplementary Fig. 2c**).

While we made every effort to collect additional muscle biopsies, due to invasiveness of muscle sampling, it was unfortunately impossible to include more in this study. However, we performed additional staining on F1^{m1/m1}, F2^{m2/m3} and two control samples simultaneously (**New Fig. 2g and Fig. 5f**).

If the stop gain/frameshift variants do not lead to nonsense mediated decay one would expect to observe a truncated SNUPN protein, while these were not observed in two patients with C term truncating variants. Full gels should be provided and if possible more cases and controls included.

The full-length SPN1 protein has a molecular weight of 41 kDa whereas the truncated SPN1 *m6* (c.902-903delAT. p.Tyr301Cysfs*29) and *m7* (c.899-900delAC, p.Asp300Valfs*30) proteins are expected to be around 37 kDa with only 32 amino acids lacking (**Reviewer Figure 5a-b**). Thus the 4kDa difference between wildtype (WT) and *m6* or *m7* mutant SPN1 proteins is challenging to detect. Even by increasing the gel percentage to 10%, it was nearly impossible to observe this difference with endogenous SPN1. However, in HEK293T transfected cells with Myc-SPN1^{WT}, Myc-SPN1^{m1}, Myc-SPN1^{m6} or Myc-SPN1^{m7} constructs, we were able to identify slightly shorter bands for Myc-SPN1^{m6} and Myc-SPN1^{m7} compared to Myc-SPN1^{WT} and Myc-SPN1^{m1} (**Reviewer Figure 5c**).

a)

Wildtype SPN1	MEELSQUALASSFSVSQDLNSTAAPHRLSQYKSKYSSLEQSERRRRLLELQKSKRLDYVN	60
m6 SPN1	MEELSQUALASSFSVSQDLNSTAAPHRLSQYKSKYSSLEQSERRRRLLELQKSKRLDYVN	60
m7 SPN1	MEELSQUALASSFSVSQDLNSTAAPHRLSQYKSKYSSLEQSERRRRLLELQKSKRLDYVN	60

Wildtype SPN1	HARRLAEDDWTGMESEEEENKKDDEEMDIDTVKKLPKHANQLMLSEWLIDVPSDLGQEWI	120
m6 SPN1	HARRLAEDDWTGMESEEEENKKDDEEMDIDTVKKLPKHANQLMLSEWLIDVPSDLGQEWI	120
m7 SPN1	HARRLAEDDWTGMESEEEENKKDDEEMDIDTVKKLPKHANQLMLSEWLIDVPSDLGQEWI	120

Wildtype SPN1	VVVCVPGKRALIVASRGSTSAYTKSGYCVNRFSSLLPGGNRRNSTAKDYTILDICIYNEVN	180
m6 SPN1	VVVCVPGKRALIVASRGSTSAYTKSGYCVNRFSSLLPGGNRRNSTAKDYTILDICIYNEVN	180
m7 SPN1	VVVCVPGKRALIVASRGSTSAYTKSGYCVNRFSSLLPGGNRRNSTAKDYTILDICIYNEVN	180

Wildtype SPN1	QTYYYLDVMCWRGHPFYDCQTDFRFYWMHSHKLPPEEGLGEKTKLNPFKFVGLKNFPCPTPE	240
m6 SPN1	QTYYYLDVMCWRGHPFYDCQTDFRFYWMHSHKLPPEEGLGEKTKLNPFKFVGLKNFPCPTPE	240
m7 SPN1	QTYYYLDVMCWRGHPFYDCQTDFRFYWMHSHKLPPEEGLGEKTKLNPFKFVGLKNFPCPTPE	240

Wildtype SPN1	SLCDVLSMDFPFVEVDGLLFYHKQTHYSPGSTPLVGWLRPYMVSDVLGVAVPAGPLTTKPD	300
m6 SPN1	SLCDVLSMDFPFVEVDGLLFYHKQTHYSPGSTPLVGWLRPYMVSDVLGVAVPAGPLTTKPD	300
m7 SPN1	SLCDVLSMDFPFVEVDGLLFYHKQTHYSPGSTPLVGWLRPYMVSDVLGVAVPAGPLTTKPV	300

Wildtype SPN1	YAGHQLQQIMEHKKSQKEGMKEKLTHKASENGHYELEHLSTPKLKGSSHSPDHPGCLMEN	360
m6 SPN1	CWAPAPADYGAQE--EPEGRHEGETHTQGL-----	328
m7 SPN1	CWAPAPADYGAQE--EPEGRHEGETHTQGL-----	328
. : :: : ** : * ** .		

b)

SPN1 Protein	Size (kDa)
Wildtype	41.15
m6	37.36
m7	37.36

c)

Reviewer Figure 6. a) Protein sequence encoded by wildtype (Lane 1) and mutant *m6* and *m7* transcripts (Lanes 2 and 3). Amino acid changes in mutant proteins are highlighted in red. **b)** Predicted protein size for SPN1 wildtype and mutants **c)** Western blot analysis of HEK293T cells transfected with wildtype or mutant Myc-tagged SPN1 constructs. Truncated Myc-SPN1^{m6} and Myc-SPN1^{m7} appear at a lower size compared to Myc-SPN1^{WT} and Myc-SPN1^{m1}.

It seems the anti SNUPN/SPN1 antibody used correctly binds to the N terminus, worth mentioning it

In this study, we employed the SPN1 polyclonal antibody (Proteintech, Catalog # 15358-1-AP) referred in **supplementary Table 3**. As we reviewed the manufacturer's product information, the immunogen used to generate this antibody consists of a fusion protein

containing the complete SPN1 sequence. Consequently, we speculate that this antibody is a combination of multiple epitopes targeting various segments of the protein. As it is common, we have described it in the text as a “pan” anti-Snurportin-1 antibody.

Also retention of truncated isoforms escaping NMD in proteins which acts as part of dimers/oligomers/complexes as SNUPN would be expected to lead to a dominant-negative effect, while the disease is recessive and parents unaffected. This should be considered, discussed and probably reflected in CO-IP experiments (wt/wt co transfection should be carried out, not only wt/wt, mut/mut).

As you recommended, we co-transfected the HEK293T cells with FLAG-SPN1^{WT} and various WT and mutant Myc-SPN1 constructs. As reviewer 1 and us suspected, we demonstrated by immunoprecipitation that Myc-SPN1 frameshift mutants (*m6* and *m7*) are unable to form heterodimers with Flag-SPN1^{WT} (**New Supplementary Fig.3b**), suggesting that intact C-terminal region is indeed necessary for homodimer formation.

We hypothesize that haplosufficiency is at play in the heterozygous carrier, where having one normal copy of a functional gene is sufficient to maintain its function. The intact proteins produced by the wildtype allele in the parents can still form SPN1 oligomers, ensuring the continuation of normal function. It is worth noting that physiopathology of SMA, a primarily autosomal recessive disorder, has also been directly associated with defects in SMN self oligomerization (Lorson *et al*, 1998).

It may also be useful to consider transfecting a shorted SNUPN isoform lacking the C terminus to further support the role of C terminus on oligomerization.

Thank you for your suggestion that was also recommended by Reviewer 1 (comment #3). First, we generated several constructs, both wildtype and mutant, with the C-terminal region spanning all our mutations and tagged with either Myc or Flag (Myc-SPN1²⁵⁴⁻³⁶⁰ and Flag-SPN1²⁵⁴⁻³⁶⁰). As expected, we noted that *m6* and *m7* fragments were highly unstable compared to WT and *m1*. Firstly, we observed that WT SPN1²⁵⁴⁻³⁶⁰ can only interact with WT and *m1* SPN1²⁵⁴⁻³⁶⁰ fragments but not with *m6* and *m7* SPN1²⁵⁴⁻³²⁸ constructs. Secondly, very interestingly, the binding of *m1* SPN1²⁵⁴⁻³⁶⁰ fragments with itself was much less efficient than the one with the WT C-terminal fragments confirming that Ile309 residue is likely to be involved in the oligomerization (**New Fig. 3f and Supplementary Fig. 3d**).

Why only 2 fibroblasts were used in figure 3d?

We have updated the figure by adding a new western blot showing fibroblasts cytoplasmic fractions of $F4^{m6/m6}$ and $F10^{m9/m9}$. Whereas the SPN1 band at 150kD disappeared in the sample from $F4^{m6/m6}$, it was still present in $F10^{m9/m9}$ sample (**New Fig. 3d**). These results are consistent with the pull-down experiment showing that SPN1^{Arg55Gln} which carries a modification in the N-terminus does not affect the SPN1 oligomerization (**New Supplementary Fig. 3a**).

At least 3 controls should be used in RNAseq experiments before drawing more conclusions from differently expressed genes and exons.

You are absolutely correct. As explained in response to your first comment, we initiated the study by performing RNA-seq on three control and three patient samples. However, due to the identification of one of the control samples as an outlier, we excluded it from our analysis. We next ensure the accuracy of our results by validating them by different means such as qPCR, western-blot and immunofluorescence on various samples including primary fibroblast, muscle tissues and CRISPR-mutant cell lines (**New Fig. 5e-g and Supplementary Fig. 5g-h**).

The clustering of patients and controls should be showed as

PCAplots. PCA plot is shown in **new Supplementary Fig.5a**.

Do authors see in SNUPN transcripts the presence of variants escaping NMD?

Interestingly, the activation of Nonsense-Mediated Decay (NMD) depends on the specific location of nonsense mutations. Mutations situated at least 50 nucleotides upstream of an exon junction lead to the degradation of the affected mRNA at an accelerated rate. In contrast, the less common nonsense mutations occurring within the last exon do not trigger NMD. Instead, they yield a stable mRNA that leads to the production of truncated proteins (Neu-Yilik *et al*, 2011; Nagy & Maquat, 1998). Notably, all our *SNUPN*-truncating variants are located in the last exon and their RNA and truncated protein were all detected (**New Fig. 2d and supplementary Fig. 2c**).

Additional comments:

Clinical data could be more clearly presented.

We apologize if this section appeared unclear but we would have appreciated more input on how we could improve it. All the clinicians who contributed to this study provided valuable insights and comments during the writing of the clinical sections. We followed a logical sequence by initially presenting our first case and subsequently compiling the most

notable clinical phenotypes for the entire cohort. Furthermore, we have summarized all available clinical information in a comprehensive clinical table (Table 1), and detailed descriptions of each case are provided in the supplementary notes.

Case II-1 from family 10 carrying biallelic missense variant has early onset and severe phenotype, which contrast the genotype-phenotype correlation.

We agree, the early and severe phenotype observed in patient F10^{m9/m9}, who carries a missense mutation in the N-terminal region of SPN1, was unexpected.

Since we had the opportunity to obtain primary fibroblast from this patient recently, we further investigated its function. Although this mutant did not appear to display oligomerization defects, SPN1 immunofluorescence staining revealed an aberrant pattern similar to that observed in other mutant fibroblasts (**New Supplementary Fig. 2b**). Most notably, we observed a disruption in snRNP biogenesis in this patient's fibroblasts, as evidenced by the redistribution of COILIN to the nucleolus (**New Supplementary Fig. 4b**) and dysregulations in *LAMA5*, *SGCA*, *SFR2* and *ITGB2* genes (**New Fig. 5e**).

We have added in the discussion that this variant due to its particular localization in the IBB domain, is likely pathogenic through additional and more severe mechanisms. In the future, it will be interesting to delve deeper into understanding how this missense mutation can exert such a profound impact on the protein's functionality.

The spelling of single mutation including nt and AA change throughout the text instead of m.1 ..m2 etc would make the reading easier.

Thank you for your comment. The use of abbreviations and symbols, such as "m" for mutations, is a common practice in human genetic papers to simplify the text and avoid repetitive use of the full mutation nomenclature (Moreno Traspas *et al*, 2022). Therefore, we would appreciate maintaining the current labeling.

Reviewer #4 (Remarks to the Author):

In this manuscript, the authors investigated the link between snurportin-1 (SPN1) deficiency and muscular dystrophy. Using primary fibroblasts and muscle tissues from patients, their major findings are: 1) in the large majority of patients, mutations in the *spn1* gene are mapped to the C-terminal domain, 2) mutations in the C-termini of SPN1 mutants prevent the formation of oligomeric complexes, 3) SPN1 mutants alter the assembly of the snRNP complex and impair its transport to the nucleus, 4) disruption of the maturation process of the spliceosome in SPN1 fibroblasts mutants, 5) SPN1 deficiency alters key components of the extracellular matrix, 6) the structural integrity of the cytoskeletal component is compromised in cells deficient in SPN1. Based on these observations, the authors conclude that SPN1 is critical in maintaining the structural integrity of the extracellular matrix, DGC, and cytoskeleton network and that mutations in the c-termini of SPN1 are associated with muscular dystrophy.

The overall manuscript is well-written, and the results appear interesting. However, there are several issues with this manuscript. The main concerns of this reader are that the results are over interpreted and that there are several discrepancies between the results and the drawn conclusions.

Thank you for your positive feedback. We have incorporated additional experiments and provided further explanations to thoughtfully address the discrepancies that you have rightly pointed out.

1-The data showing the accumulation of SPN1 (Fig 2 f) around the nuclei of the patients' fibroblasts are very sketchy. The authors should provide the entire field of wild-type and mutant cultured fibroblasts (not one cell) with quantifications.

Thank you for your suggestion. We have included high resolution immunofluorescence images showing SPN1 immunofluorescence on a group of cells for a total of 2 wild-type and 5 mutant fibroblasts (**New Fig. 2d** and **New Supplementary Fig. 2d**). Majority of the mutant cells show an accumulation of SPN1 in the cytoplasm particularly around the nuclei. Quantification confirmed an increased trend in the fluorescence intensity in the perinuclear region of mutants, which appeared to be significant in four mutants F1^{m1/m1}, F2^{m2/m3}, F4II:2^{m6/m6} and F10^{m9/m9} compared to WT's fibroblasts (**New Supplementary Fig. 2e**).

Additionally, the claim that Fig. 2g supports the perinuclear accumulation of SPN1 mutant observed in mutant fibroblasts is weak as the SPN1 staining is uniformly distributed over the entire sarcolemma and upregulated in the sarcoplasm. The authors should provide high-resolution images to make their case.

In order to address this comment accurately, we performed immunofluorescence staining on muscle cryosections from F1^{m1/m1}, F2^{m2/m3} and two controls simultaneously. Muscle biopsies were compared to controls obtained from the same type of muscle, and DAPI

was used to label the nuclei. To account for potential variations in muscle section thickness and their potential influence on the results, we utilized high magnification and confocal microscopy to generate Z-stack images. Our new results clearly demonstrate the accumulation of SPN1 signal particularly along the sarcolemma, around nuclei and within the sarcoplasm in both mutants compared to their respective control (**New Fig. 2g**). To prevent redundancy in our results, we have substituted the previous SPN1 immunohistochemistry (IHC) staining with new immunofluorescence images. Both sets of images are presented side by side in **Reviewer Figure 7** for your reference.

Reviewer Figure 7. SPN1 immunofluorescence staining (top panel) and SPN1 immunohistochemistry staining (bottom panel) images using wildtype and patient F1^{m1/m1} muscle cryosections.

Also, the authors should explain why the expression levels of the SPN1 mutant have increased in muscle cells but not in fibroblasts when compared to the wild type and the significance of such results.

To strengthen our findings, we included two additional patient fibroblast samples (F4II:2^{m6/m6} and F10^{m9/m9}) and employed *SNUPN* HeLa mutant cells generated using CRISPR/Cas9. We meticulously repeated all immunofluorescence staining and conducted multiple technical replicates for western blots, accompanied by corresponding quantification.

Notably, our patient fibroblasts carrying C-terminal mutations exhibited lower SPN1 protein levels, as quantified in five separate western blot analyses. However, intriguingly, the immunofluorescent images depicted a higher SPN1 signal in the patient's fibroblast cytoplasm (**New Fig. 2d-f and new Supplementary Fig. 2d-e**).

To further investigate this controversy, we have generated two SPN1 HeLa mutant cell lines. The first line named SPN1^{sgEx2} harbor two types of mutations (c.85_89delinsACT and c.88C>A) at the beginning of the gene (Exon 2), while the second named SPN1^{sgEx9} carry three types of mutations (c.896_921delCAGACTATGCTGGGCACCAGCTCCAG; c.899_903delinsCTCCA and c.903_915delTGCTGGGCACCAG) in the final exon (Exon 9), very similar to our patient variants notably *m6* and *m7*. Firstly, we observed by western blot significant reduction in the total SPN1 endogenous protein level in SPN1^{sgEx2} HeLa mutant cells, as expected given the frameshift variant c.85_89delinsACT. Secondly, we also observed a reduction of SPN1 total protein in SPN1^{sgEx9} (**New Supplementary Fig.2f**). However, we observed a remarkable increase in SPN1 cytoplasm signal in SPN1^{sgEx9} but not in SPN1^{sgEx2} by immunofluorescence (**Supplementary Fig.2g**). Together these data corroborate our results with patients' cells carrying mutations in C-terminal.

Our hypothesis to explain the observed discrepancy between western blotting and immunofluorescence staining for SPN1 lies in the instability of SPN1 mutant proteins compared to the wildtype protein. Although we cannot pinpoint the exact cause of the observed discrepancy, we suspect that SPN1 mutant proteins with C-terminal alterations may be particularly unstable in solution, despite our stringent measures to prevent degradation. Our observation aligns with prior findings of SPN1 degradation in solution compared to its higher stability when it is in a complex state (Strasser *et al*, 2005). While in the immunofluorescence process, immediate fixation or freezing of mutant SPN1 within their native cellular environment may better prevent their degradation and provide a more accurate representation of their distribution.

Finally, the immunofluorescence results obtained with SPN1 on fibroblast and muscle patients are consistent since both show an increase in the cytoplasm with particular aggregation around the nuclei.

Finally, the expression levels of SPN1 mutants in fibroblasts may not give an idea about the pathogenicity of SNUPN variants. This should be done on muscle biopsies.

We agree that fibroblasts are not the most suitable cells for studying muscle-related mechanisms. However, through the utilization of these cells, we have identified a rational pathogenic mechanism that we have in part validated in distinct models such as muscle tissues and two distinct HeLa mutant cell lines. (**New Fig. 5e-g and Supplementary Fig. 5g-h**).

For this study, we were only able to obtain muscle cryosections from two patients, and regrettably, no additional muscle biopsies were available for further analysis. In our forthcoming research endeavors, we aspire to reinforce the exploration of this mechanism by identifying additional cases and collecting more muscle samples.

2. The authors claimed in the abstract of this manuscript that mutant SPN1 failed to homodimerize, and yet in Figure 3 d, they showed in mutant fibroblasts the existence of a higher band around 100 kDa, which presumably due to the dimerization of SPN1 monomers. The authors should clarify this issue.

Thank you for bringing up this issue. To address this matter, we also employed the cytoplasmic fraction of both HeLa wildtype and *SNUPN* mutant cells (SPN1^{sgEx2} and SPN1^{sgEx9}).

Western blot data demonstrate a substantial decrease in SPN1 protein levels in HeLa mutant cells compared to wildtype in the lower exposure images whereas the loading of GAPDH was equal for all samples (**Reviewer Figure 8**).

As expected, in the cytoplasmic fraction of WT cells, we observed bands at 100 kDa and 150 kDa, corresponding to SPN1 dimers and oligomers, respectively. Interestingly, we detected a decrease in both SPN1 oligomers and dimers in SPN1^{sgEx2} cells, reflecting the downregulation of SPN1 in these cells (**Reviewer Figure 8 lane 3**). This highlights that the 100 kDa band corresponds to SPN1 dimer.

Notably, the oligomer band is completely absent in SPN1^{sgEx9} cells, which carry a mutation in the C-terminal region, confirming the critical role of the C-terminus in oligomerization. However, the dimer band appears to be slightly increased compared to the total SPN1 level in SPN1^{sgEx9} cells which could correspond to a compensatory response to stabilize SPN1 proteins that cannot form oligomers (**Reviewer Figure 8, lane 2**). In fact, we hypothesize that SPN1 C-terminus, which is essential for oligomerization, might be partially or not involved in its dimerization. This could explain why the 100 bp is not disrupted in our patients samples. For further evidence, please refer to replies to Reviewer 1 (Section 2 and 3 from major comments and Section c from minor comments).

Indeed, SPN1 interactions appear to be more complex than initially anticipated. In this current manuscript, we have chosen to emphasize our findings on the C-terminal region for which we provide strong evidence of its involvement in oligomerization (**New Fig. 3e-f And Supplementary Fig. 3d**). The potential role of other SNP1 domains in its self-interaction will be further explored in our future studies.

Reviewer Figure 8. Western blot analysis of SPN1 in the cytoplasmic fraction of HeLa wildtype, SPN1^{sgEx2} and SPN1^{sgEx9} mutant cells.

3. The crux of this paper is that mutations in the c-termini of SPN1 alter the assembly of the snRNP complex and impair its transport to the nucleus. This reader has an issue with the following conflict

results: In Figure 2 f and g the authors claimed that in F1m1/m1 mutant fibroblasts and muscles, SPN1 accumulates around the nucleus (failed to translocate into the nucleus). However, in Figure 4C, they provided data showing that SPN1 is highly localized in the nuclear fraction. The authors should explain this discrepancy and provide immunostaining data of SNRNPB and coilin/SMN in F1m1/m1 mutant fibroblasts.

We believe that this particular missense mutation does not prevent SPN1 from being transported into the nucleus but does affect the formation and importation of the snRNP complex. Furthermore, impairment of SPN1's transport to the nucleus is not a prerequisite for altering the biogenesis of the SnRNP complex. This was demonstrated by Ospina et al, 2005 where they argued that cargo binding was not a requirement for SPN1 nuclear import (Ospina et al, 2005).

Due to the variability and decrease in SPN1 protein levels observed in the mutants through western blot (**New Fig. 2d-e**), we were no longer confident in asserting defects in SPN1 mutant transport into the nuclei based on our fractionation results. Consequently, we decided to remove this panel and instead concentrate on the implications of SPN1 mutations on the snRNP complex.

Firstly, the quantification and statistical comparison of anti-Sm staining in the nucleoplasm of both wild-type (WT) and mutants have already been conducted and were presented in old Supplementary Fig. 4b. These data have now been incorporated into the main figure (**New Fig. 4c**). Furthermore there is an abnormal increase of Sm in the cytoplasm as evidenced in all the mutant cytoplasmic fractions (**New Fig. 4d**). We also have confirmed by immunofluorescence a reduction of Sm proteins in the two *SNUPN* HeLa mutant cells line (**New Supplementary Fig. 4a**).

Secondly, as suggested from Reviewer 1 we performed co-staining of COILIN and FIBRILLARIN. Remarkably, we have successfully validated a significant relocalization of COILIN to the nucleolus in all available SPN1 mutants when compared to WT fibroblasts (**New Fig. 4e- f and Supplementary Fig. 4b**). In parallel, we have conducted the same experiment using *SNUPN* mutant HeLa lines. Interestingly, the results in both HeLa mutant cell lines corroborate those observed in mutant fibroblasts (**New Supplementary Fig. 4c**). These new pieces of data align with the existing literature (Shpargel & Matera, 2005) and provide compelling confirmation that SPN1 defects lead to abnormal relocalization of COILIN to the nucleolus.

Thirdly, in response to the advice of Reviewer 1 not to use fibroblasts to study CBs formation, we have removed the data showing COILIN/SMN in fibroblasts. Instead, we have shifted our focus to examining HeLa *SNUPN* mutant cells in which CBs are clearly observable. In the new panels, **Fig. 4g and h**, we clearly demonstrate a significant increase in COILIN foci in two HeLa *SNUPN* mutant cell lines, indicative of CBs breakdown.

Taken together these data provide strong evidence that endogenous deficiency of SPN1 directly contributes to disruptions in snRNP biogenesis.

4. In Figure 2 g, the authors provided immunohistochemistry images of muscle sections showing a more intense level of SPN1 in F1 m1/m1 mutant subsarcolemmal. Figure 4 f provided images showing the complete absence/diffuse of SPN1 staining in F1 m1/m1 mutant subsarcolemmal. These conflicting results are concerning.

Thank you for pointing out these concerns. As explained in comment 1, part 2, we have repeated the immunofluorescence staining and imaging on two patients and two control muscle samples simultaneously. This new set of data confirmed clearly an aggregation of SPN1 along the sarcolemma, around the nuclei and within the sarcoplasm in mutant muscle sections compared to healthy controls. This finding aligns with the results of our previous immunohistochemistry analysis (**Old Fig. 2g**). Thus we replaced this panel with

the high resolution immunofluorescence images from both F1 and F2 muscle section (**New Fig. 2g**).

5. Concerning the dysregulation of ECM components and DGC complex in SPN1 mutants, the authors should label muscle sections with antibodies against DGC complex and ECM proteins to determine which molecules are affected by the SPN1 mutations.

As you suggested, DGC and ECM markers were explored by immunofluorescence. We performed SGCA and Collagen IV stainings using the two available patient's $F1^{m1/m1}$ and $F2^{m2/m3}$ muscle cryosections. DAPI was used to label the nuclei and all the images were captured by confocal microscopy. SGCA and COL IV were consistently found to be decreased in both patient's muscle sections (**New Fig. 5f**). Remarkably, These stainings corroborate the results found in fibroblasts and HeLa mutant cells (**New Supplementary Fig. 5g-h**).

Furthermore, in our manuscript, at the end of the section titled "SPN1 deficiency leads to dysregulation of ECM associated key components", we had included the following sentence: "*Remarkably, histopathological reports of muscle biopsies from three patients (F2-II:1, F10-II:1 and F12-II:2) documented a reduction of DAG1, the extracellular α DAG subunit which further substantiated our in vitro data (Supplementary notes).*"

Unfortunately, we were unable to retrieve the images but we have retested α DAG in the $F2\text{-II:1}^{m2/m3}$ muscle sections. We clearly confirmed the decrease of α DAG signal in the sarcolemma of this patient muscle compared to the control muscle (**Reviewer Fig. 9**).

Reviewer Figure 9. Immunofluorescence showing reduction of α DAG (red) in muscle sections from $F2\text{-II:1}^{m2/m3}$ patient compared to wildtype (WT2). DAPI was used to label the nuclei.

References

- Ajiro M, Awaya T, Kim YJ, Iida K, Denawa M, Tanaka N, Kurosawa R, Matsushima S, Shibata S, Sakamoto T, et al (2021) Author Correction: Therapeutic manipulation of IKBKAP mis-splicing with a small molecule to cure familial dysautonomia. *Nat Commun* 12: 6039
- Albayda J & van Alfen N (2020) Diagnostic Value of Muscle Ultrasound for Myopathies and Myositis. *Curr Rheumatol Rep* 22: 82
- Hsieh Y-C, Guo C, Yalamanchili HK, Abreha M, Al-Ouran R, Li Y, Dammer EB, Lah JJ, Levey AI, Bennett DA, et al (2019) Tau-Mediated Disruption of the Spliceosome Triggers Cryptic RNA Splicing and Neurodegeneration in Alzheimer's Disease. *Cell Rep* 29: 301–316.e10
- Lorson CL, Strasswimmer J, Yao JM, Baleja JD, Hahnen E, Wirth B, Le T, Burghes AH & Androphy EJ (1998) SMN oligomerization defect correlates with spinal muscular atrophy severity. *Nat Genet* 19: 63–66
- Moreno Traspas R, Teoh TS, Wong P-M, Maier M, Chia CY, Lay K, Ali NA, Larson A, Al Mutairi F, Al-Sannaa NA, et al (2022) Loss of FOCAD, operating via the SKI messenger RNA surveillance pathway, causes a pediatric syndrome with liver cirrhosis. *Nat Genet* 54: 1214–1226
- Nagy E & Maquat LE (1998) A rule for termination-codon position within intron-containing genes: when nonsense affects RNA abundance. *Trends Biochem Sci* 23: 198–199
- Neu-Yilik G, Amthor B, Gehring NH, Bahri S, Paidassi H, Hentze MW & Kulozik AE (2011) Mechanism of escape from nonsense-mediated mRNA decay of human beta-globin transcripts with nonsense mutations in the first exon. *RNA* 17: 843–854
- Ospina JK, Gonsalvez GB, Bednenko J, Darzynkiewicz E, Gerace L & Matera AG (2005) Cross-talk between snurportin1 subdomains. *Mol Biol Cell* 16: 4660–4671
- Shpargel KB & Matera AG (2005) Gemin proteins are required for efficient assembly of Sm-class ribonucleoproteins. *Proc Natl Acad Sci U S A* 102: 17372–17377
- Strasser A, Dickmanns A, Lührmann R & Ficner R (2005) Structural basis for m3G-cap-mediated nuclear import of spliceosomal UsnRNPs by snurportin1. *EMBO J* 24: 2235–2243
- Vill K, Sehri M, Müller C, Hannibal I, Huf V, Idriess M, Gerstl L, Bonfert MV, Tacke M, Schroeder AS, et al (2020) Qualitative and quantitative muscle ultrasound in patients with Duchenne muscular dystrophy: Where do sonographic changes begin? *Eur J Paediatr Neurol* 28: 142–150

Reviewer #1 (Remarks to the Author):

The revised manuscript is greatly improved. The authors did a fair amount of work and should be commended for it.

Signed, Greg Matera

There are only a few minor comments I have regarding the author response to previous reviews document.

1. I think I can help shed some light on the mAb Y12 issue here, as one of my postdoctoral advisors developed this antibody. The reason that none of the websites or companies list an epitope for this antibody is because this monoclonal antibody was derived from an autoimmune mouse. Despite the fact that it is an mAb, it was not raised on purpose from a specific antigen. HOWEVER, the epitope has been determined to be composed of RG dipeptide repeats (present on C-terminal tails of certain Sm proteins) that contain sDMA (PMID: 10747894).

2. Regarding the new subcellular localization data, it might be worth pointing out (to the authors and perhaps also to the readership) that Fibrillarin is a marker for the dens fibrillar component of the nucleolus. Thus it does not stain the granular component. So in the SNUPN mutant cell lines, the coilin and fibrillarin staining may not perfectly overlap. Upon SNUPN RNAi, coilin relocates to both parts of the nucleolus (granular + fibrillar) - often forming foci inside of the GC. Depending on how the DAPI staining is done, you can often see the entire nucleolus more clearly in that channel.

3. On page 17, the authors talk about the molecular modeling shown in Fig. 3A. In my review, I was the one who pointed out the presence of this alpha helix in the C-terminus. There was no mention of it in the original manuscript. And by comparing the original ms submission to the current revised one, there is suddenly a well defined helix in the new WT panel 3a. Yet the description of how they did the modeling remains the same as in the first draft. Did the authors change any settings or input data in I-TASSER? What changed? Did they use alpha-fold as I had done?

Excerpt from response to review:

"Thank you once again for your valuable input. As suggested, we have generated new SPN1 models using I-TASSER. Indeed in the WT model, we observe a well-defined alpha Helix between Gly303 and Glu322 which appears to be partially or entirely disrupted in all the mutant models (New Fig. 3a)."

There is no mention of them doing anything different, either in the revised manuscript itself or in the response to reviews. Seems a bit strange. How did the C-term go from looking like spaghetti in the first draft to an alpha helix now?

4. I don't understand this sentence on pg 18: "...were unable to immunoprecipitate their Myc homologous..."

Do the authors mean to say: "...were unable to immunoprecipitate their Myc-tagged counterparts"?

5. Also on pg 18 and Fig. 3a-b, I would change this conclusion from: "These data strongly suggested that an intact C-terminal region is indeed necessary for oligomerization." to - These data demonstrate that an intact C-terminal region is required for SPN1 dimerization.

These experiments do more than just 'suggest.' They demonstrate that this region is necessary. But they also do not really tell us anything about the oligomerization status. Panels 3d-f start to get at the question of oligomer formation. So the argument in the text should be built up in a stepwise fashion.

6. I'm not sure I agree with the interpretation on pg 19 that the C-term (CT) is more important for the

oligomer. Where are the data with a WTΔCT mutant? If that construct is unable to bind to itself or to the full length protein, then the CT is required for dimer formation. I'm not sure what the 100kD band is. Are the authors certain?

I do agree that the data regarding the Ile309Ser mutation is an interesting result suggestive of a role for this residue in higher order multimerization.

Reviewer #2 (Remarks to the Author):

The revised manuscript has improved a lot, but I still have some major concern and few minor comments which should be addressed.

Major comments:

I am still not totally convinced and confident about causality of (p.Arg55Gln). as there are not enough robust data supporting its pathogenicity and it's clearly an outlier in this study and dataset which should be discussed in the discussion because with more data in future it might turns out that this variant was not pathogenic at all. Also you should not make association between this variant and a more severe phenotype. You should discuss the evidence supporting its pathogenicity as well as the ones against its causality such as 1- it was found in only one single case 2- limited segregation support 3- relatively higher AF in comparison to other variants in different databases and other changes of same residues, Arg55Leu, Arg55Trp, 4- location of the variant in protein 5- a bit different phenotype with additional features (e.g. microcephaly and ID) which might or might not be associated with this variant, 6- functional data 7- limitation of testing this specific variant. Also please move the pedigree and the data to the end of the clinical table and the figure if its possible.

There are 2 individual who are homozygous for (p.Leu324His) which is in c-terminus close to Ile309Ser in gnomAD V4.0.0 and could be a good control for testing your assay to see if and how it can distinguish these two variants. This could be a nice way of showing how robust and sensitive your assay is for testing the missense variants in c-terminus of the protein. In fact its interesting that this is the only homozygous variant in c-terminus of the protein in control genetic databases.

Minor comments:

In abstract why this is a syndromic MD, what is syndromic about this condition apart from some uncommon variable features and cataract in 5-6 in few cases specially family 5 which is still not totally confirmed, and we don't know I these additional features are due to NSUN deficiency or second mutations. Please remove syndromic.

Please add AF from new gnomAD V4.0.0 as well.

Mode of inheritance, still it doesn't make sense, so I assume you just simply mean genetic studies. with atypical muscular dystrophy and variable neurological defects

Reviewer #3 (Remarks to the Author):

The authors have addressed my comments

Reviewer #4 (Remarks to the Author):

The authors have satisfactorily addressed the reviewers' comments, and as a result, I have no objections to endorsing the publication of this paper.

Reviewer #1 (Remarks to the Author):

Point-by-point Response to the Editor's and Reviewers' Comments

Nashabat et al., (2023)

Manuscript number: NCOMMS-23-18913-T

12th October 2023

The revised manuscript is greatly improved. The authors did a fair amount of work and should be commended for it.

Signed, Greg Matera

Dear Greg,

Thank you for acknowledging our earnest efforts to improve the manuscript.

There are only a few minor comments I have regarding the author response to previous reviews document.

1. I think I can help shed some light on the mAb Y12 issue here, as one of my postdoctoral advisors developed this antibody. The reason that none of the websites or companies list an epitope for this antibody is because this monoclonal antibody was derived from an autoimmune mouse. Despite the fact that it is an mAb, it was not raised on purpose from a specific antigen. HOWEVER, the epitope has been determined to be composed of RG dipeptide repeats (present on C-terminal tails of certain Sm proteins) that contain sDMA (PMID: 10747894)

Thank you for the clarification.

2. Regarding the new subcellular localization data, it might be worth pointing out (to the authors and perhaps also to the readership) that Fibrillarin is a marker for the dense fibrillar component of the nucleolus. Thus it does not stain the granular component. So in the SNUPN mutant cell lines, the coilin and fibrillarin staining may not perfectly overlap. Upon SNUPN RNAi, coilin relocalizes to both parts of the nucleolus (granular + fibrillar) - often forming foci inside of the GC. Depending on how the DAPI staining is done, you can often see the entire nucleolus more clearly in that channel.

Thank you for your insightful comment. Indeed, it is interesting to consider the distinction between the dense fibrillar and granular components of the nucleolus in relation to FIBRILLARIN and COILIN staining. We did incorporate this information on p.22 when we first introduced Fibrillarin as a nucleolus marker.

3. On page 17, the authors talk about the molecular modeling shown in Fig. 3A. In my review, I was the one who pointed out the presence of this alpha helix in the C-terminus. There was no mention of it in the original manuscript. And by comparing the original ms submission to the current revised one, there is suddenly a well defined helix in the new WT panel 3a. Yet the description of how they did the modeling remains the same as in the first draft. Did the authors change any settings or input data in I-TASSER? What changed? Did they use alpha-fold as I had done?

Excerpt from response to review:

"Thank you once again for your valuable input. As suggested, we have generated new SPN1 models using I-TASSER. Indeed in the WT model, we observe a well-defined alpha Helix between Gly303 and Glu322 which appears to be partially or entirely disrupted in all the mutant models (New Fig. 3a)."

There is no mention of them doing anything different, either in the revised manuscript itself or in the response to reviews. Seems a bit strange. How did the C-term go from looking like spaghetti in the first draft to an alpha helix now?

Thank you for bringing your concern to our attention. In fact, we did not make any changes on our end, except for the timing of uploading the FASTA sequences to I-TASSER for structure prediction. The initial SPN1 protein structure, without a predicted alpha-helix in C-terminal, was generated in 2021, while the updated version is from 2023. It's worth noting that I-TASSER employs deep machine learning algorithms, which result in improved predictions over time. In hindsight, we should have considered this and updated our protein prediction before the initial submission. We appreciate your diligence once more in pointing this out and helping us enhance the accuracy of our results.

4. I don't understand this sentence on pg 18: "...were unable to immunoprecipitate their Myc homologous..."

Do the authors mean to say: "...were unable to immunoprecipitate their Myc-tagged counterparts"?

As per your suggestion, the sentence was corrected in the text.

5. Also on pg 18 and Fig. 3a-b, I would change this conclusion from: "These data strongly suggested that an intact C-terminal region is indeed necessary for oligomerization." to - These data demonstrate that an intact C-terminal region is required for SPN1 dimerization.

As per your suggestion, we have revised the sentence. We also opted to use “SPN1 self-interaction” as a broader concept instead of employing specific terms such as “oligomerization or dimerization” for this sentence.

These experiments do more than just 'suggest.' They demonstrate that this region is necessary. But they also do not really tell us anything about the oligomerization status. Panels 3d-f start to get at the question of oligomer formation. So the argument in the text should be built up in a stepwise fashion.

You are absolutely correct. We have removed the term “oligomer” before panel Fig. 3c, where we began to demonstrate SPN1 oligomerization.

6. I'm not sure I agree with the interpretation on pg 19 that the C-term (CT) is more important for the oligomer. Where are the data with a WT Δ CT mutant? If that construct is unable to bind to itself or to the full length protein, then the CT is required for dimer formation. I'm not sure what the 100kD band is. Are the authors certain?

We perfectly understand your concern. But all the evidence shown in the manuscript and below has prompted us to this conclusion.

1-In additional set of experiments, we discovered that Δ CT truncated WT SPN1¹⁻²⁵³ is capable of binding itself and SPN1^{Ile309Ser(m1)} (Rev. Fig. 1A, lanes 1 and 2). Interestingly, this interaction is disrupted with *m6* and *m9* SPN1 mutants (Reviewer Fig. 1A, lanes 3 and 4).

Reviewer Figure 1: Co-immunoprecipitation performed in HEK293T cells co-transfected with Myc and Flag-tagged SPN1 constructs. Input and eluate samples blotted with anti-Myc (top) or anti-Flag (bottom) antibodies reveal interaction between Myc-SPN1^{WT(1-253)} and Flag-SPN1^{WT(1-253)} and with Flag-SPN1^{m1} (Lanes 1 and 2) but not with Flag-SPN1^{m6} and Flag-SPN1^{m9} (Lanes 3 and 4).

2- In the previous rebuttal, we addressed a similar concern about the 100 kDa band raised by Reviewer 4. Seeking a deeper understanding, we used the cytoplasmic fraction without DTT treatment from both HeLa wildtype and *SNUPN* mutant cells (SPN1^{sgEx2} and SPN1^{sgEx9}).

As a reminder, SPN1^{sgEx2} harbor two types of mutations (c.85_89delinsACT and c.88C>A) at the beginning of the gene (Exon 2), while SPN1^{sgEx9} carry three types of mutations (c.896_921delCAGACTATGCTGGGCACCAGCTCCAG; c.899_903delinsCTCCA and c.903_915delTGCTGGGCACCAG) in the final exon (Exon 9), very similar to our patient variants notably *m6* and *m7*.

- As expected, in the cytoplasmic fraction of WT cells, we observed bands at 100 kDa and 150 kDa, corresponding to SPN1 dimers and oligomers, respectively (**Reviewer Figure 2**)
- Interestingly, we detected a decrease in both SPN1 oligomers and dimers in SPN1^{sgEx2} cells, reflecting the downregulation of SPN1 in these cells (**Reviewer Figure 2, lane 3**). This indicates that the 100 kDa band corresponds to SPN1 dimer.
- Notably, the oligomer band is completely absent in SPN1^{sgEx9} cells, which carry a mutation in the C-terminal region, confirming the critical role of the C-terminus in oligomerization. However, the dimer band appears to be slightly increased compared to the total SPN1 level in SPN1^{sgEx9} cells which could correspond to a compensatory response to stabilize SPN1 proteins that cannot form oligomers (**Reviewer Figure 2, lane 2**).

Reviewer Figure 2. Western blot analysis of SPN1 in the cytoplasmic fraction of HeLa wildtype, SPN1^{sgEx2} and SPN1^{sgEx9} mutant cells.

Altogether, these results have led us to hypothesize that SPN1 C-terminus, essential for oligomerization, might be partially or not involved in its dimerization. This could explain why the 100 bp is not significantly disrupted in our patients' samples.

We hope that you will agree that SPN1 self-interactions appear to be more complex than initially anticipated. Therefore, for more clarity in this current manuscript, we have chosen to emphasize our findings on the C-terminal region, for which we provide strong evidence of its involvement in oligomerization (New Fig. 3e-f And Supplementary Fig. 3d). In our future studies, we will further explore and support the potential role of N-terminal SPN1 in its self-interaction.

I do agree that the data regarding the Ile309Ser mutation is an interesting result suggestive of a role for this residue in higher order multimerization.

Thank you for your positive feedback.

Reviewer #2 (Remarks to the Author):

The revised manuscript has improved a lot, but I still have some major concerns and few minor comments which should be addressed.

Thank you for your positive and objective review. We have attempted to clarify most, if not all, of the concerns that you raised.

Major comments:

I am still not totally convinced and confident about causality of (p.Arg55Gln). as there are not enough robust data supporting its pathogenicity and it's clearly an outlier in this study and dataset which should be discussed in the discussion because with more data in future it might turn out that this variant was not pathogenic at all.

While we understood your doubt in the first review, we now differ with the notion that there is not enough solid data supporting the pathogenicity of p.Arg55Gln. Furthermore, we would like to mention that three other reviewers did not raise any doubt regarding the pathogenicity of this variant. In the previous revised manuscript, we had already provided a substantial amount of evidence including residue conservation (Fig. 2c), pathogenicity predictions (Supplementary Table 1), structural disruption prediction (Fig. 3a), and functional defects in the patient's primary cells and several assays (Supplementary Fig. 2d-e, Fig. 4d, Supplementary Fig. 4b and Fig. 5e), which are usually sufficient to establish variant pathogenicity.

Histopathological reports from this patient revealed abnormal muscle fibers, particularly a predominance of type I fibers and endomysial fibrosis. These findings were consistent and observed in 8 additional skeletal muscle biopsies from other affected individuals. Additionally, it was reported a reduction of DAG1, the extracellular α -DAG subunit, further supporting our *in vitro* data (Table 1 and Supplementary notes).

We observed a disruption in snRNP biogenesis and dysregulation of ECM associated key components in this patient's fibroblasts. Notably, these data mirrored observations made in C-terminal mutant fibroblasts and in our 2 mutant cell lines, SPN1^{sgEx2} and SPN1^{sgEx9} (Fig. 4 and 5). As a reminder, SPN1^{sgEx2} harbors two types of mutations (c.85_89delinsACT and c.88C>A) at the beginning of the gene (Exon 2). Additionally, our data are completely in line with the findings of Ospina *et al* in which they demonstrated that mutation in particular residues of region 11-72 abolished the snRNP import (Ospina et al. 2005).

Recently, we have obtained one more piece of evidence. Indeed, we discovered that while truncated WT SPN1¹⁻²⁵³ is capable of binding itself (Reviewer Figure 1, lane 1) and F1-SPN1^{m1} mutant (Lane 2), this interaction is completely abolished with F4-SPN1^{m6} and F10-SPN1^{Arg55Gln(m9)} mutants (Lanes 3 and 4). This data clearly confirms that the N-terminal of *m9* is disrupted.

Taken together, there is no doubt that the muscular clinical phenotypes, muscle histopathology observations, and functional data collectively support the establishment of a causative link between the p.Arg55Gln variant and the muscular disorder diagnosed in this patient. As we suggested in the first rebuttal and our revised discussion, this mutant is likely operating through additional mechanisms due to its particular localization in the IBB domain. Since it is outside the scope of this manuscript, in our future studies, we will further explore the essential role of the SPN1 N-terminal region in muscular dystrophy.

Also you should not make association between this variant and a more severe phenotype.

We removed “more severe” as per your suggestion although we are not entirely convinced that it is not the case.

You should discuss the evidence supporting its pathogenicity as well as the ones against its causality such as 1- it was found in only one single case 2- limited segregation support 3- relatively higher AF in comparison to other variants in different databases and other changes of same residues, Arg55Leu, Arg55Trp, 4- location of the variant in protein 5- a bit different phenotype with additional features (e.g. microcephaly and ID) which might or might not be associated with this variant, 6- functional data 7- limitation of testing this specific variant.

We agree to discuss, as you suggested, evidence against its pathogenicity. However, we feel that all the following points are not totally valid against its pathogenicity for the following reasons:

1- While it is true that only one variant has been identified in this region for now, we anticipate that this is just the initial discovery. We believe that the findings from our work will likely contribute to the identification of additional variants in the future. This is another reason why we consider it essential to report this particular variant.

2- Segregation may be limited, but it still supports the autosomal recessive inheritance of the variant given that the parents are heterozygous.

3- We do not follow your point in mentioning that: “this variant has a relatively higher AF in comparison to other variants.” For instance, in gnomAD V4, the AF for p.Ile309Ser (m1)

and p.Arg55Gln (m9) are comparable at 5.47×10^{-6} and 9.34×10^{-6} , respectively. Furthermore, there are no homozygous variants reported.

The fact that this residue is mutated into several other residues (Arg55Leu, Arg55Trp) still with a low AF rate does not go against its pathogenicity; it might rather highlight that this region is a hotspot for mutations. However, no homozygous variant has been reported yet.

4- We have already discussed earlier the reasons why we think that the location of this variant in the IBB domain is not incompatible with its pathogenicity.

5- It is true that this patient displays additional clinical features (microcephaly and ID). However, she is not this only one since few other patients also displayed additional phenotypes, particularly neurodevelopmental delay, low global brain volume reduction and sensorineural hearing loss in F14, F15 and F12, respectively. Because of these symptoms, we have previously mentioned in our discussion the following points *“In light of the numerous neurological abnormalities observed in the patients who underwent MRI examination, it is advisable to perform an early and comprehensive assessment of the central nervous system in all individuals with biallelic SNUPN disease-causing variants. These findings will also provide new opportunities to investigate the as-yet uncharacterized role of SPN1 in neurological processes”*.

6 & 7: As mentioned above, for the revision we did not feel limited for testing this variant as we managed to get primary fibroblasts and generated the DNA construct of the variant. We performed multiple assays and clearly demonstrated that its pathogenicity was consistent with the other mutants in the C-terminal domain. However, we do not rule out the possibility that this variant is involved in other functions of SPN1.

Nevertheless, we have expanded and moderated our discussion about this variant to be on the cautious side, as you suggested.

Also please move the pedigree and the data to the end of the clinical table and the figure if its possible.

As we explained in the first rebuttal, we opted to maintain the original order in which we collected the families and implemented a numbering and color-coding system for each variant instead. We believe that placing this family last will not make a major difference to the reader, as it already appears among the last families and its variant number is the last in the list (m9). Furthermore, changing the family number at this stage would impact all our family coding and follow-up.

There are 2 individual who are homozygous for (p.Leu324His) which is in c-terminus close to Ile309Ser in gnomAD V4.0.0 and could be a good control for testing your assay to see if and how it can distinguish these two variants. This could be a nice way of showing how

robust and sensitive your assay is for testing the missense variants in c-terminus of the protein. In fact its interesting that this is the only homozygous variant in c-terminus of the protein in control genetic databases.

It is true that since the release of gnomAD V4 in early November 2023, there are 2 homozygous variants p.Leu324His that appeared. However, this number remains very low and it can not be totally certain that they were detected in healthy samples as explicitly stated by gnomAD (see below in yellow). Furthermore, with 62 heterozygous and 2 homozygous alleles out of 1.6 million, the AF of $3.84 \cdot 10^{-5}$ from gnomAD V4 can not be considered high. Therefore, we are currently not convinced this is the most optimal and reliable negative control, especially considering that we know that this region is a hotspot for *SNUPN* pathogenic variants. Nevertheless, we will try to make every effort to obtain more information about these two individuals to verify they are not affected by a similar disorder.

Statement From gnomAD V4 *"While we are provided high level study phenotype and case/control status for some samples, we do not have comprehensive phenotype metadata for gnomAD samples, and many samples are now derived from large biobanks, which can include individuals with disease. As such, we cannot ensure that samples in a non-disease subset do not have the specified disease..."*

Minor comments:

In abstract why this is a syndromic MD, what is syndromic about this condition apart from some uncommon variable features and cataract in 5-6 in few cases specially family 5 which is still not totally confirmed, and we don't know I these additional features are due to NSUN deficiency or second mutations. Please remove syndromic.

As per your suggestion we removed "syndromic" from the abstract.

Please add AF from new gnomAD V4.0.0 as well.

We have incorporated the data from gnomAD V4, which was released in early November 2023, into Supplementary Table 1. Unfortunately, we were unable to update our table with this new version for the previous rebuttal, as we had already resubmitted our manuscript before the release date.

Mode of inheritance, still it doesn't make sense, so I assume you just simply mean genetic studies.

with atypical muscular dystrophy and variable neurological defects

We use “Mode of Inheritance” to mention the manner in which the disorder is passed from one generation to the next. In our case, autosomal recessive.

Reviewer #3 (Remarks to the Author):

The authors have addressed my comments

Reviewer #4 (Remarks to the Author):

The authors have satisfactorily addressed the reviewers' comments, and as a result, I have no objections to endorsing the publication of this paper.